# How many samples are needed to leverage smoothness?

**Vivien Cabannes**
Meta AI

**Stefano Vigogna**
University of Rome Tor Vergata

## Abstract

A core principle in statistical learning is that smoothness of target functions allows to break the curse of dimensionality. However, learning a smooth function seems to require enough samples close to one another to get meaningful estimate of high-order derivatives, which would be hard in machine learning problems where the ratio between number of data and input dimension is relatively small. By deriving new lower bounds on the generalization error, this paper formalizes such an intuition, before investigating the role of constants and transitory regimes which are usually not depicted beyond classical learning theory statements while they play a dominant role in practice.

## 1 Introduction

The current practice of machine learning consists in feeding a machine with many samples for it to infer a useful rule. In supervised learning, the samples are input/output (I/O) pairs, and the rule is a relationship to predict outputs from inputs. Once learned, the I/O mapping can be deployed in the wild to infer useful information from new inputs. Because it was not engineered by hand, one can question the existence of unwanted behaviors. Classical guarantees regarding the mapping's correctness are offered by statistics: assuming that the training samples are independent and identically distributed according to the future use cases, it is possible to derive theorems akin to the central limit theorem.

While many statistical learning principles offer practical insights, theoretical results often appear somewhat obscure, and forming intuition about them is often challenging, which limits their impact. In this paper, we focus on one simple principle: "smoothness allows us to break the curse of dimensionality". The curse of dimensionality is a generic term referring to a set of high-dimensional phenomena with significant practical consequences. In supervised learning, it manifests as follows: without a good prior on the I/O mapping to be learned, one can only get good estimates of the mapping close to the observed examples; as a consequence, to obtain a good global estimate, one needs to collect enough data points to finely cover the input space, which implies that the number of data points should scale exponentially with the dimension of the input space. Yet, when provided with the information that the mapping has some structure, one might need significantly less examples to learn from. This is notably the case when the mapping is assumed to be smooth. The goal of this paper is to better understand how and when we can expect to get much better convergence guarantees when the target function is known to be smooth.

**Related work.** Nonparametric local estimators were introduced as soon as the field of learning began to form in the 50's [11], and their consistency was studied extensively in the second half of the twentieth century (see Stone [29], Devroye et al. [8] and references therein). Introduced for scatter plots [7], local polynomials were the first estimators to leverage smoothness to improve regression [12]. They were later replaced by kernel methods, which are a powerful way to adapt to smoothness without much fine-tuning, and were widely regarded as state-of-the-art before the deep learning era [25]. Convergence results for kernel methods can be understood through the size of their associated functional spaces [13, 32], how those sizes relate to generalization guarantees [31], and how those spaces adhere to $L^2$ [22, 26]. For least-squares regression, relations between the size of those spaces

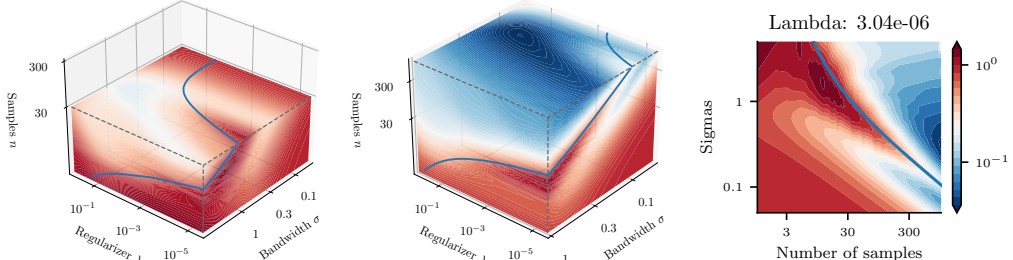

**Figure 1:** Log-log-log-log plots of excess risk (in color) with respect to the number of samples ($z$-axis), the regularizer $\lambda$ ($x$-axis) and the bandwidth $\sigma$ ($y$-axis) when $f^*(x) = \text{sign}(\langle x, e_1 \rangle)$ and $\rho_{\mathcal{X}}$ is uniform on $[-1, 1]^2$. Asymptotically, there exists some $(\lambda_n, \sigma_n)$ to ensure that the excess risk decreases in $O(n^{-\gamma})$ for a $\gamma$ predicted by theory. However, for finite values of $n$, the excess risk can present different power law decays. The dark blue line on each plot indicates the transition between low- and high-sample regime: it corresponds to the graph $\{(\sigma, \lambda, n) \mid n = \mathcal{N}_1(\sigma, \lambda)\}$ (see (10) for the definition of $\mathcal{N}_1$). The figure illustrates a double descent phenomenon where excess risk peaks are reached when $n = \mathcal{N}_1$ before a second descent takes place. Those peaks can be avoided in practice by computing the effective dimension $\mathcal{N}$ and tuning hyperparameters to ensure it to be smaller than $n$.

and generalization guarantees are usually derived through operator concentration [27, 6]. More recently, transitory regime behaviors were described in Mei et al. [16], Mei and Montanari [15], and high-dimensional phenomena that lift the need to have more data than dimensions were studied in [23, 14], with Bach [3] relating the latter analyses with the former ones.

**Contributions.** This work focuses on the role of smoothness in breaking the curse of dimensionality for typical supervised learning problems. At a high-level, our main contribution is to showcase the importance of *transitory regimes* where one does not have enough samples to leverage high-order smoothness. In such transitory regimes, the behavior of the excess risk can be quite different than its asymptotic "stationary" behavior. Usually not well described by theory, they might be the dominating regimes in applied machine learning, where the number of samples is often relatively small compared to the input dimension (see Figure 1 for an illustration). This arguably explains the poor performance of kernel methods without strong kernel engineering in the deep learning era: they try to leverage smoothness, but usually do not access enough samples to meaningfully estimate high-order derivatives. More precisely, our contributions are twofold:

- We provide two generic "minimax" lower bounds that illustrate how the curse of dimensionality can not be fully beaten under smoothness assumptions alone.
- We delve more specifically into guarantees offered by algorithms that are built to leverage smoothness assumptions, and get a fine-grained picture of some of the transitory regimes where learning takes place in practice.

All results are illustrated by numerical experiments, some of them to be found in Appendix C.

## 2 The Significance of Constants

This section reviews classical results in learning theory, before providing generic lower bounds when relying solely on smoothness assumptions.

### 2.1 Established Upper Bounds

Supervised learning is concerned with learning a function from an input space $\mathcal{X}$ to an output space $\mathcal{Y}$ from a dataset $\mathcal{D}_n = (X_i, Y_i)_{i \in [n]}$ of $n \in \mathbb{N}$ examples.[1] For simplicity, we will assume $\mathcal{Y} = \mathbb{R}$ and $\mathcal{X} = \mathbb{R}^d$, or $\mathcal{X} = \mathbb{T}^d$ being the torus. A learning rule, or learning algorithm, is a mapping $\mathcal{A} : \mathcal{D}_n \mapsto f_n$ that builds a function $f_n : \mathcal{X} \to \mathcal{Y}$ based on the dataset $\mathcal{D}_n$, with the goal of capturing

---

[1]We use $[n] = \{1, \ldots, n\}$ to denote the set of integer from one to $n$.

the underlying I/O relation. To discuss generalization to unseen examples, it is standard to model both the already collected and the future examples as independent realizations of a random pair $(X, Y)$.

**Assumption 1.** *There exists a distribution $\rho \in \Delta_{\mathcal{X} \times \mathcal{Y}}$ that has generated $n$ independent training samples $\mathcal{D}_n = (X_i, Y_i)_{i \in [n]} \sim \rho^{\otimes n}$, and will generate future test samples independently.*

Under Assumption 1, the quality of a mapping $f : \mathcal{X} \to \mathcal{Y}$ is measured through the excess risk

$$\mathcal{E}(f) = \mathcal{R}(f) - \mathcal{R}(f^*) = \|f - f^*\|_{L^2(\rho_{\mathcal{X}})}^2, \tag{1}$$

where $\rho_{\mathcal{X}}$ is the marginal distribution of $\rho$ on $\mathcal{X}$ and, assuming that $(Y \mid X = x)$ has a second order moment for every $x \in \mathcal{X}$,

$$f^*(x) = \mathbb{E}\left[Y \mid X = x\right], \qquad \text{which minimizes} \qquad \mathcal{R}(f) = \mathbb{E}[|f(X) - Y|^2]. \tag{2}$$

The risk $\mathcal{R}(f)$ represents the average error of guessing $f(X)$ in place of $Y$ when the error is measured through the least-squares loss. From a statistical viewpoint, it is useful to model $f_n = \mathcal{A}(\mathcal{D}_n)$ as a random function (inheriting its randomness from the samples), so to study the expectation or the upper tail of the excess risk $\mathcal{E}(f_n)$. Provided that $f^*$ is measurable, it is possible to find methods such that $\mathcal{E}(f_n)$ converges to zero in probability. Without additional assumptions on $f^*$, it is not possible to give any guarantee on the speed of this convergence [12, Theorem 3.1]. However, when $f^*$ is assumed to be smooth, the picture improves consequently.

**Theorem 1** (Breaking the curse of dimensionality [12, 4])**.** *Under Assumption 1, when $f^*$ is $\alpha$-smooth for $\alpha > 0$ in the sense that it admits $\lfloor \alpha \rfloor$ derivatives that are regular, more precisely if $f^* \in C^\alpha$ (i.e. $f$ is $\alpha$-Hölder regular), or $f^* \in H^\alpha = W^{\alpha,2}$ (i.e. $f$ is $\alpha$-Sobolev regular), there exists a learning rule $f_n = \mathcal{A}(\mathcal{D}_n)$ that guarantees*

$$\mathbb{E}_{\mathcal{D}_n}[\mathcal{E}(f_n)] \leq cn^{-2\alpha/(2\alpha+d)}, \tag{3}$$

*where $c$ is a constant independent of $n$. Moreover, the bound (3) is minimax optimal, in the sense that for any rule $f_n = \mathcal{A}(\mathcal{D}_n)$ there exists a distribution $\rho$ such that $f^* \in \mathcal{C}^\alpha$, or $f^* \in H^\alpha$, and the upper bound (3) holds as a lower bound with a different constant $c$.*

**Why do constants matter?** At first glance, when two algorithms $\mathcal{A}_1$ and $\mathcal{A}_2$ guarantee two different upper bounds $O(n^{-\gamma_1})$ and $O(n^{-\gamma_2})$ on the expected excess risk, $\mathcal{A}_1$ will be deemed superior to $\mathcal{A}_2$ if $\gamma_1 \geq \gamma_2$, since after a certain number of samples, we will have that $\mathbb{E}_{\mathcal{D}_n}[\mathcal{A}_1(\mathcal{D}_n)] \leq \mathbb{E}_{\mathcal{D}_n}[\mathcal{A}_2(\mathcal{D}_n)]$. However, the constants hidden in the front of the big $O$s might lead to a different picture when given a small number of samples: $\mathcal{A}_1$ might actually be a so-called "galactic algorithm", similarly to Strassen's algorithm for matrix multiplication, that might not be worth using without an indecently large number of samples.

## 2.2 Minimax Lower Bounds

The following lower bounds show how classical algorithms that reach the minimax optimal convergence rates based on smoothness assumptions (3) necessarily present constants that are growing fast with respect to the input dimension. Our analysis holds in noisy settings.[2]

**Assumption 2** (Homoscedasticity)**.** *The noise in the label $(Y \mid X = x)$ is assumed to be independent of $x \in \mathcal{X}$ with $\mathrm{var}\,(Y \mid X = x) = \varepsilon^2$.*

**Theorem 2.** *Under Assumptions 2, for any algorithm $\mathcal{A}$, there exists a target function $f^*$ such that $f^{*(\alpha+1)} = 0$, and*

$$\mathbb{E}_{\mathcal{D}_n}[\mathcal{E}(\mathcal{A}(\mathcal{D}_n))] \geq \frac{\varepsilon^2}{n} \binom{d + \alpha}{d}. \tag{4}$$

**Theorem 3.** *Under Assumption 2, for any algorithm $\mathcal{A}$, there exists a target function $f^*$ such that its Fourier transform is compactly supported on the $\ell^\infty$-disk of radius $\omega$, i.e., $\widehat{f^*}(\omega') = 0$ for all $\|\omega'\|_\infty > \omega$, and*

$$\mathbb{E}_{\mathcal{D}_n}[\mathcal{E}(\mathcal{A}(\mathcal{D}_n))] \geq \frac{\varepsilon^2(2\omega + 1)^d}{n}. \tag{5}$$

---

[2]This contrasts with interpolation regimes where other phenomena might appear, requiring different analysis tools.

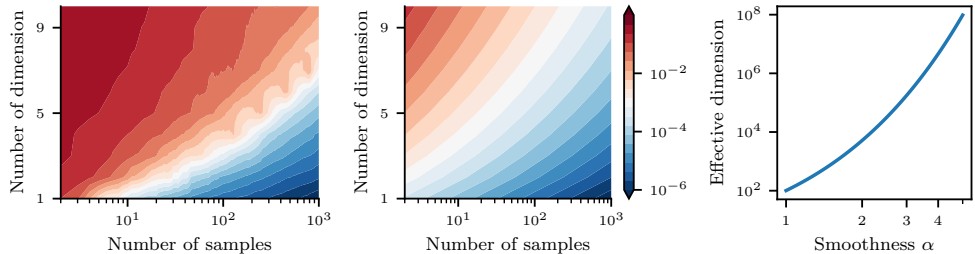

**Figure 2:** (Left) Convergence rates as a function of the input dimension $d$ and the number of samples $n$ when the target function is $f^*(x) = x_1^5$ and $k(x, y) \propto (1 + x^\top y)^5$. We observe that convergence rates depend heavily on the dimension, which is mainly due to "changing constants". (Middle) Theoretical lower bound. As the number of samples increases and we enter the high-sample regime, the lower bound resembles the real convergence rates. (Right) Illustration of $\alpha \mapsto \mathcal{N}(\alpha) = \binom{d+\alpha}{\alpha}$ for $d = 100$, which corresponds to the dimension of the space of polynomials with $d$ variables of degree at most $\alpha$. Given a number of samples, Taylor expansion can only be estimated meaningfully up to the order $\alpha$ such that $\mathcal{N}(\alpha) \simeq n$. In particular, it shows that in dimension one-hundred, one millions samples ($n = 10^6$) only allow to leverage no more than fourth-order smoothness ($\alpha = 4$). See Appendix C.2 for details.

*Proof Sketch (detailed in Appendix).* The proofs of those *two new theorems* consist in retaking the standard lower bound in $\varepsilon^2 D/n$ when performing linear regression with $D$ orthogonal features. In Theorem 2, $D$ corresponds to the number of polynomials of degree less than $\alpha$ with $d$ variables; in Theorem 3, $D$ is the number of integer vector $m \in \mathbb{Z}^d$ whose norm is smaller than $\omega$. $\square$

Theorems 2 and 3 illustrate how the usage of strong smoothness assumptions on $f^*$ can not guarantee an excess risk lower than $\varepsilon^2$ without accessing an indecently large number of samples with respect to the input dimension (e.g., $n \geq \binom{d+\alpha}{d} \geq (1+\alpha/d)^d$ or $n \geq 2^d \omega^d$ respectively). Empirical validations are offered by Figure 2. In their inner-workings, those theorems capture how the rates derived through Theorem 1 are deceptive when one does not have enough samples compared to the size of the hypothesis space that $f^*$ is assumed to belong to; and that the size of smooth functions spaces grows quite fast with respect to the dimension of the input space. Similar lower bound theorems can be proven with rates in $n^{-2\alpha/(2\alpha+d)}$ under more detailed assumptions, as illustrated with Theorem 7 in Appendix. Efficient learning beyond the limits imposed by those theorems can only take place when leveraging other priors: for example, sparsity priors would reduce the minimax rates from $\varepsilon^2 D/n$ to $\varepsilon^2 s \log(D)/n$ where $s$ is the sparsity index of $f^*$ [9].

## 3 Crisp Picture in RKHS Settings

To provide a fine-grained analysis of the phenomena at stake, this section presents stylized settings where constants and transitory regimes can be studied precisely, allowing to get a better picture of convergence rates in practical machine learning setups.

### 3.1 Backbone Analysis

In the following, we shall consider a feature map $\varphi : \mathcal{X} \to \mathcal{H}$, with $\mathcal{H}$ a Hilbert space and $\varphi \in L^2(\rho_\mathcal{X})$. The map $\varphi$ linearly parameterizes the space of functions

$$\mathcal{F} = \{f_\theta : x \mapsto \langle \varphi(x), \theta \rangle_\mathcal{H} \,|\, \theta \in \mathcal{H}\} \subset L^2(\rho_\mathcal{X}). \tag{6}$$

For example, $\mathcal{H}$ could be $\mathbb{R}^k$, $\varphi$ seen as defining $k$ features $\varphi_i(x)$ on inputs $x \in \mathcal{X}$. This model can be used to estimate $f^*$ through the empirical risk minimizer

$$f_{n,0} \in \arg\min_{f \in \mathcal{F}} \sum_{i \in [n]} |f(X_i) - Y_i|^2. \tag{7}$$

In order to ensure that $\mathcal{F}$ can learn any function $f^*$, the features can be enriched by concatenating an infinite countable number of features together. In this setting, it is more convenient to describe the geometry induced by $\mathcal{F}$ through the (reproducing) kernel $k : \mathcal{X} \times \mathcal{X} \to \mathbb{R}$ defined as $k(x, x') =$

$\langle \varphi(x), \varphi(x') \rangle$. When $\mathcal{F}$ can fit too many functions, the estimate (7) needs to be refined to avoid overfitting. This paper will focus on Tikhonov (also known as ridge) regularization[3]

$$f_n \in \arg\min_{f \in \mathcal{F}} \sum_{i \in [n]} |f(X_i) - Y_i|^2 + n \|f\|_{\mathcal{F}}^2, \tag{8}$$

where the norm is defined from (6) as $\|f\|_{\mathcal{F}} = \inf \{\|\theta\| \mid f = f_\theta\}$, but can also be expressed with the sole usage of $k$ through the integral operator $K : L^2(\rho_{\mathcal{X}}) \to L^2(\rho_{\mathcal{X}})$,

$$Kf(x) = \int_{\mathcal{X}} k(x, x') f(x') \rho_{\mathcal{X}}(\mathrm{d}x') = \mathbb{E}_X[k(x, X) f(X)], \tag{9}$$

as $\|f\|_{\mathcal{F}} = \|K^{-1/2} f\|_{L^2(\rho_{\mathcal{X}})}$, with the convention $K^{-1}(\ker K) = \{+\infty\}$.

The statistical quality of $f_n$ (8) depends on two central quantities, defined as

$$\mathcal{N}_a(K) = \mathrm{Tr}\left(K^a (K+1)^{-a}\right) \qquad \text{and} \qquad \mathcal{S}(K) = \left\|(K+1)^{-1} f^*\right\|_{L^2(\rho_{\mathcal{X}})}^2, \tag{10}$$

where $a = 2$. The first term, known as the effective dimension, quantifies the size of the space $\mathcal{F}$ in which $f^*$ is searched for. It relates to the variance of the estimator as a function of the dataset $\mathcal{D}_n$. It will capture the estimation error, the error due to the finite number of accessed samples, related to the risk of overfitting. The second term quantifies the adherence of $f^*$ to $\mathcal{F}$. It can be understood as the proximal distance between $f^*$ on $\mathcal{F}$ since $(K+1)^{-1} = I - K(K+1)^{-1}$ is a proximal projector. It will capture the approximation error, the error due to the fact that our model does not exactly fit the target function, related to the risk of underfitting.

**Theorem 4** (High-sample regime learning behavior). *Under Assumptions 1 and 2, as well as two mild technical Assumptions 3 and 4, when $f^*$ is in the closure of $\mathcal{F}$ in $L^2(\rho_{\mathcal{X}})$, there exists a constant $c$ such that the estimate* (8) *verifies*

$$\left| \mathbb{E}_{\mathcal{D}_n}\left[ \mathcal{E}(f_n) \right] - \frac{\varepsilon^2 \mathcal{N}_2(K)}{n} - \mathcal{S}(K) \right| \leq c\mathcal{N}_1(K) \left( a_n \cdot \frac{\varepsilon^2 \mathcal{N}_1(K)}{n} + a_n^{1/2} \mathcal{S}(K) \right) \tag{11}$$

*where $a_n = \mathcal{N}_+(K)/n$, and $\mathcal{N}_+(K) = \mathrm{ess\,sup}_{x \sim \rho_{\mathcal{X}}} \mathrm{Tr}(K_x(K+1)^{-1})$ with $K_x$ the rank-one operator on $L^2(\rho_{\mathcal{X}})$ that maps $f$ to the constant function equal to $\mathbb{E}[f] k(x, x)$. The different notions of search space size are always related by $\mathcal{N}_2 \leq \mathcal{N}_1 \leq \mathcal{N}_+$.[4] Moreover, under the interpolation property $K^p(L^2(\rho_{\mathcal{X}})) \hookrightarrow L^\infty(\rho_{\mathcal{X}})$, i.e. $\|K^p f\|_\infty \leq \|f\|_{L^2(\rho_{\mathcal{X}})}$, it holds that $\mathcal{N}_+(\lambda^{-1} K) = O(\lambda^{-2p})$; while under the source condition $f^* \in K^r(L^2(\rho_{\mathcal{X}}))$, it holds that $\mathcal{S}(\lambda^{-1} K) = O(\lambda^{2r})$.*

While the excess risk upper bound deriving from Theorem 4 is somewhat standard, the *lower bound is new*. This theorem states that the generalization error $\mathbb{E}_{\mathcal{D}_n}[\mathcal{E}(f_n)]$ behaves as $A(n, K) := \varepsilon^2 \mathcal{N}_2(K)/n + \mathcal{S}(K)$ up to higher order terms specified in the right-hand side. Theorem 4 also holds for ridge-less regression (7) with $\mathcal{N}_1(K, 0) = \mathcal{N}_2(K, 0) = \dim \mathcal{F}$, $\mathcal{N}_+(K, 0) = \|K^{-1}\|^{-1} \|\varphi\|_\infty$, and $\mathcal{S}(K, 0) = \|f^* - \pi_{\mathcal{F}} f^*\|^2$, where $\pi_{\mathcal{F}}$ is the $L^2(\rho_{\mathcal{X}})$-orthogonal projection onto $\mathcal{F}$. In this setting, $\mathbb{E}_{\mathcal{D}_n}[\mathcal{E}(f_{n,0})] = A(n, K, 0)(1 + O(n^{-1}))$. More in general, we conjecture the right-hand side of Theorem 4 to be improvable with the removal of $\mathcal{N}_1(K)$ in front of the rates (which is due to our usage of concentration inequalities on operators rather than on scalar values), the change of the second $\mathcal{N}_1$ into $\mathcal{N}_2$, and the substitution of $a_n^{1/2}$ by $a_n$. This would show that $\mathbb{E}_{\mathcal{D}_n}[\mathcal{E}(f_n)]$ behaves as $A(n, K)(1 + O(a_n))$. In the following, we will call very high-sample regimes situations where $a_n \leq 1$, and *high-sample regimes* situations where $\mathcal{N}_2(K) \leq n$.

## 3.2 Approach Generality

Despite their apparent simplicity, reproducing kernels $k$ describe rich spaces of functions $\mathcal{F}$, known as reproducing kernel Hilbert space (RKHS), namely any Hilbert space of functions with continuous pointwise evaluations [2]. Classical examples are provided by subspaces of analytical functions $C^\omega$

---

[3]In practice, it is usual to add a regularization parameter $\lambda$ in front of $\|f\|_{\mathcal{F}}^2$ in (8). This parameter $\lambda$ can be incorporated as a specific hyperparameter of the kernel by replacing $k$ by $\lambda^{-1} k$. This is useful to unify the study of the different hyperparameters that might define a kernel.

[4]When $\rho_{\mathcal{X}}$ does not present heavy tails behaviors, those three quantities actually behaves similarly.

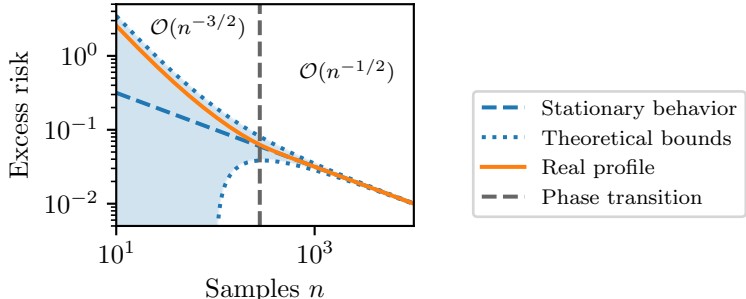

**Figure 3:** Illustration of transitory regimes. In essence, Theorem 4 states that $\mathcal{E}_n := \mathbb{E}_{\mathcal{D}_n}[\mathcal{E}(f_n)] = A(n,K)(1 + h(n,K))$ for $h = O(\mathcal{N}_+(K)/n)$. We illustrate our upper-lower bound when $A(n,K) = n^{-1/2}$ and $nh(n,K)$ is known to be in $[-10^2, 10^2]$. The upper-lower bound forces $\mathcal{E}_n$ to behave in $n^{-1/2}$ when $n$ goes to infinity, yet when $n$ is small, it can showcase quite different "transitory" behaviors.

through the Gaussian kernels $k(x,x') = \exp(-\|x - x'\|^2/\sigma^2)$, and by the Sobolev space $H^{(d+1)/2}$ through the exponential kernel $k(x,x') = \exp(-\|x - x'\|/\sigma)$. Reproducing kernels encompass several approaches and algorithms that have been suggested to leverage smoothness of the target function $f^*$.

The first approach consists in estimating local Taylor expansion through local polynomials, defined through

$$\varphi(x) = \lambda^{-1/2}(\mathbf{1}_{x \in A} x^i)_{i \leq \beta, A \in \mathcal{P}}, \tag{12}$$

for $\mathcal{P}$ a partition on $\mathcal{X}$, $\beta \in \mathbb{N}$ a degree, $\lambda > 0$ a regularization parameter. Intuitively in high-dimension problems, where $d = \dim(\mathcal{X})$ is big, leveraging local properties may not be very reasonable, since the covering of $\mathcal{X}$ with local neighborhoods grows exponentially with the dimension $d$ (when $\mathcal{X}$ has unit volume, and neighborhoods have a fixed radius), meaning that if one wants to have enough samples per neighborhood, $n$ should scale exponentially with $d$.

Rather than local properties, the second approach consists in leveraging global smoothness properties, through the estimation of Fourier coefficients. Such estimators can be built implicitly from translation-invariant kernels, defined as

$$k(x,x') = \lambda^{-1}q((x-x')/\sigma), \tag{13}$$

for $q : \mathbb{R}^d \to \mathbb{R}$ a basic function, $\sigma$ a bandwidth parameter, and $\lambda$ a regularization parameter. However, the number of frequencies smaller than a cut-off frequency, i.e., the number of trigonometric monomials $x \mapsto e^{im^\top x}$ with $m \in \mathbb{Z}^d$ such that $\|m\| \leq \omega$, grows exponentially with the dimension, and this approach will not escape from the curse of dimensionality.

Other approaches, such as windowed Fourier estimation, or wavelets expansion estimation (aiming to reconstruct both fine local details together with coarse large-scale behaviors), could be thought of and described through the lens of RKHS. Yet, whatsoever the definition of smoothness considered (i.e., Hölder, Sobolev, Besov), all those methods will hit an inherent limit: the number of "smooth" functions increases really fast as the dimension increases. Indeed, since their proofs simply consist in finding $D$ linearly independent function, lower bound theorems akin to Theorems 2 and 3 could be derived without difficulties for other notions of smoothness.

All the previously described methods are usually endowed with a few hyperparameters that modify the integral operator $K$ and the norm $\|\cdot\|_{\mathcal{F}}$ defining the estimator (8). Geometrically, a change of the front regularization parameter $\lambda$ leads to an isotropic rescaling of the ball $\|\cdot\|_{\mathcal{F}}^{-1}\{1\}$ inside $L^2(\rho_{\mathcal{X}})$, while a change of other hyperparameters could favor certain directions, or even remove some functions in $\mathcal{F}$. In practice, fitting hyperparameters through cross-validation can be understood as implicitly searching to balance and minimize $\mathcal{N}(K)$ and $\mathcal{S}(K)$. This fitting allows all those methods to reach the performance of Theorem 1 in $O(n^{-2\alpha/(2\alpha+d)})$ as we explain in Appendix.

| Kernel | $\mathcal{F}$ | $\mathcal{N}(\sigma, \lambda)$ | $\mathcal{S}(\sigma, \lambda; H^\alpha)$ |
|---|---|---|---|
| Gaussian | $\mathcal{F} \subset C^\omega$ | $\sigma^{-d} \log(\lambda^{-1}\sigma^d)^{d/2}$ | $\sigma^{2\alpha} \log(\lambda^{-1}\sigma^d)^{-\alpha}$ |
| Matérn | $H^\beta$ | $\sigma^{-d(2\beta-d)/2\beta}\lambda^{-d/2\beta}$ | $\sigma^{(2\beta-d)\alpha/\beta}\lambda^{\alpha/\beta}$ |
| Exponential | $H^{(d+1)/2}$ | $(\sigma\lambda)^{-d/(d+1)}$ | $(\sigma\lambda)^{2\alpha/(d+1)}$ |

**Table 1:** Example of translation-invariant kernels, their associated function classes, upper bounds (up to multiplicative constants) on their sizes as a function of the bandwidth $\sigma$ and regularization parameter $\lambda$, as well as on the bias when approximating a function in the Sobolev space $H^\alpha$. Here $C^\omega$ stands for the set of analytical functions. Proofs and details are to be found in Appendix B.

### 3.3 Exploration of Transitory Regimes

Given $n$ samples, Theorem 4 suggests to tune $K$ so as to minimize $\varepsilon^2 \mathcal{N}(K)/n + \mathcal{S}(K)$. Interestingly, while theory tends to focus on deriving convergence rates in $O(n^{-2\alpha/(2\alpha+d)})$ that maximize the coefficient $\alpha$,[5] Theorem 4 can also be leveraged to characterize tightly the expected decay of the generalization error when accessing a small number of samples. To ground the discussion, we will focus on a stylized setting where $\mathcal{N}$ and $\mathcal{S}$ can be studied in detail as a function of hyperparameters.

**Proposition 1** (Capacity and bias bounds). *When $\rho_\mathcal{X}$ is uniform on the torus $\mathbb{T}^d = \mathbb{R}^d/\mathbb{Z}^d$, and $k$ is a translation-invariant kernel $k(x, y) = \lambda^{-1}q((x-y)/\sigma)$, the capacity of the space defined through $k$ with regularization $\lambda$ and bandwidth $\sigma$ verifies, for $a \in \{1, 2\}$,*

$$\mathcal{N}_a(\sigma, \lambda) = \int_{\mathbb{R}^d} \left( \frac{\widehat{q}(\sigma\omega)}{\widehat{q}(\sigma\omega) + \lambda\sigma^{-d}} \right)^a \#(\mathrm{d}\omega), \tag{14}$$

*where $\#$ is the counting measure on $\mathbb{Z}^d \subset \mathbb{R}^d$, and $\widehat{q}$ is the (discrete) Fourier transform of $q$. Similarly, the biases quantifying the adherence of $f^*$ in $\mathcal{F}$ verify*

$$\mathcal{S}(\sigma, \lambda) = \int_{\mathbb{R}^d} \frac{|\widehat{f^*}(\omega)|^2}{(\sigma^d\widehat{q}(\sigma\omega) + \lambda)^2} \#(\mathrm{d}\omega). \tag{15}$$

*Moreover on $\mathcal{X} = \mathbb{R}^d$, if $\rho_\mathcal{X}$ has a density bounded above by $\rho_\infty < +\infty$, then (14) and (15) become upper bounds for $\mu$ the Lebesgue measure and $\widehat{f}$ the continuous Fourier transform, at the cost of extra constants in front of their right-hand sides (respectively $\rho_\infty$ and $\max(\rho_\infty, 1)$ for $\mathcal{N}_a$ and $\mathcal{S}$).*

*Proof Sketch.* This relatively standard fact follows from the assumption on $\rho_\mathcal{X}$ which implies that $K$ is diagonal in the Fourier domain. $\square$

Proposition 1 unlocks a precise sense of the effective dimension for the Gaussian kernel, defined with $q(x) = \exp(-\|x\|^2)$, the exponential kernel, with $q(x) = \exp(-\|x\|)$, and the Sobolev kernel, with $\widehat{q}(\omega) = (1 + \|\omega\|^2)^{-\beta}$, as well as the bias term $\mathcal{S}$ when approximating a function $f \in H^\alpha$ with those kernels. This is reported in Table 1 and proved in Appendix B.

**High-sample regimes in harmonics settings.** Proposition 1 is useful to describe formally different convergence rates profiles that one may expect in practice. In particular, the linearity of the bias characterization (15) is theoretically useful to decorrelate the estimation of different power laws appearing in the Fourier transform of $f$. More precisely, if

$$\left|\widehat{f^*}(\omega)\right|^2 = \int_\alpha^\infty c_\gamma (1 + \|\omega\|^2)^{-\gamma} \mu(\mathrm{d}\gamma),$$

with $c_\gamma$ being the inverse of the constant in front of the characterization of $\mathcal{S}(\sigma, \lambda; H^\gamma)$ in Table 1, and $\mu$ some measure with a profile that ensures the good definition of $f^* \in H^\alpha$, we have, taking for example $k$ as the Gaussian kernel,

$$\mathcal{S}(\sigma, \lambda; f^*, k) \simeq \int_\alpha^\infty (\sigma^2 \log(\lambda^{-1}\sigma^d)^{-1})^\gamma \mu(\mathrm{d}\gamma).$$

---

[5]For example, when $\mathcal{F} = \mathrm{im}\, K^{1/2} = H^\beta$ and $f^* \in H^\alpha$, it typically holds that $\mathcal{N}_+(\lambda^{-1}K) = O(\lambda^{-2p})$ with $p = d/4\beta$, and $\mathcal{S}(\lambda^{-1}K) = O(\lambda^{2r})$ with $r = \alpha/2\beta$, which can be used to prove Theorem 1 by tuning $\lambda = n^{-2\beta/(2\alpha+d)}$.

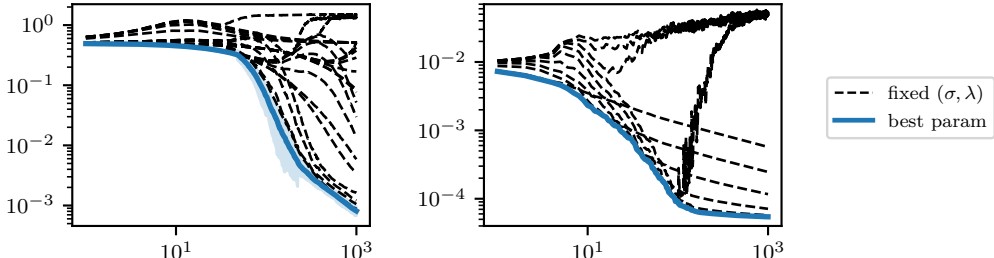

**Figure 4:** Composite convergence rates profile. The $x$-axis corresponds to the number of samples, while the $y$-axis corresponds to the excess risk. From the analysis in Section 3.3, one can build different convergence rates profiles. For example, regular functions with relatively high-frequencies are going to be hard to learn with few samples but really easy after a certain number of samples, roughly equals to the number of harmonics with lower-frequencies); while regular low-frequency functions with singularity are going to show convergence rates where the coarse details of the functions are learned with few samples, but the reconstructions of fine-grained details will required much more samples. The former profile is illustrated with the left figure, and the latter on the right figure. For any sample sizes, excess risk is reported for the best hyperparameters, found with cross-validation. More details are provided in Appendix C.3.

In particular, with $\sigma^2 \log(\lambda^{-1}\sigma^d)^{-1} = n^{-r}$, we get the following convergence rate profile, with $c_{\mathcal{N}}$ the constant in front of the characterization of $\mathcal{N}(\sigma, \lambda)$ in Table 1,

$$\mathbb{E}_{\mathcal{D}_n}[\mathcal{E}(f_n)] \simeq \inf_{r \in \mathbb{R}} \left( \varepsilon^2 c_{\mathcal{N}} n^{-1+rd/2} + \int_\alpha^\infty n^{-\gamma r} \mu(\mathrm{d}\gamma) \right) (1 + O(a_n)). \tag{16}$$

This characterization enables us to easily create target functions exhibiting different convergence profiles, as long as we stay in the high-sample regimes where our bounds are meaningful (i.e. when the factor in $1 + O(a_n)$ is relatively constant).

- *Fast then slow profile.* The first type of profile is built from $\mu$ that charges most of its mass on fast decays, but also puts some small mass on slow decays. It corresponds to target functions that are roughly well approximated by highly smooth functions, but whose exact reconstruction needs to incorporate less regular functions. The smooth part of the function will be learned quickly, yet the non-smooth part will be learned slowly. Typical examples of such a profile are provided by non-smooth functions that can be turned into infinitely differentiable ones after introducing infinitesimal perturbations, such as $f^*(x) = \exp(-\min(|x|^2, M))$, which is only $C^0$, but where one can expect to learn fast before stalling to estimate the $C^1$-singularity. Another example is given by a function made of a sum of one low-frequency cosine with large amplitude easy to learn together with one high-frequency cosine with small amplitude much harder to learn. We illustrate these profiles on Figure 4.

- *Slow then fast profile.* The second type of profile that can be created is for functions that are supported on a few eigenfunctions of $K$ associated with small eigenvalues. A typical example of this profile in one dimension would be $f^*(x) = \cos(\omega x)$ for a high frequency $\omega$. For this target function, no meaningful learning can be done when the search space is too small, because small search spaces do not contain high-frequency functions. On the other hand, when the search space is big enough, the bias quickly goes to zero, allowing for fast learning as long as one controls the estimation error. When provided with few samples, one would prefer a small search space to avoid blowing up of the estimation error, and learning will stall until enough samples are collected to explore bigger search spaces, where $f^*$ could be learned quickly. We illustrate this profile on Figure 4.

Those examples illustrate how, given a target function and a range on the number of available samples, convergence behaviors might fall in regimes that do not correspond to the steady convergence rate in $O(n^{-2\alpha/2\alpha+d})$ predicted by Theorem 1. The intuition beyond those examples is not specific to translation-invariant kernels in harmonics settings, but holds more generically for abstract RKHS.

**Empirical study of low-sample regimes.** In this work, we have focused on "under-parameterized" situations where the parameters were set to have more samples than the effective dimension of the

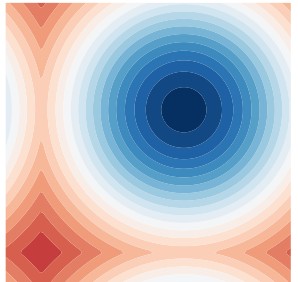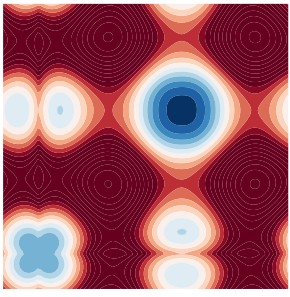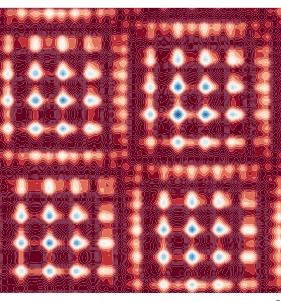

**Figure 5:** Level lines of the weights $x \to \alpha_x(x_0)$ (17) for a given $x_0 \in \mathcal{X}$, when $\mathcal{X}$ is the torus $\mathbb{R}^2/\mathbb{Z}^2$ and the kernel is taken as the Gaussian kernel with the Riemannian metric on the torus (think of an unrolled donut). Parameters are taken as $\sigma = 1$ together with $\lambda = 10^6$ (left), $\lambda = 10^2$ (middle) or $\lambda = 1$ (right). From this picture, one can build examples of non-smooth functions where the kernel inductive bias will have adversarial effects.

resulting functional space $\mathcal{F}_{t(n)}$. In a deep learning world, where many phenomena are understood as taking place in the "over-parameterized" regime, it is of interest to compare our perspective with the double descent phenomenon. Figure 1 shows the excess risk as a function of two of the three parameters $(n, \sigma, \lambda)$, as well as the graph defined by $\{(n, \sigma, \lambda) \,|\, \mathcal{N}_1(\lambda, \sigma) = n\}$. It illustrates a double descent phenomena with "phase" transition governed by the passage from the low-sample to the high-sample regime. To delve into this, we investigate into regression weight learned by kernel ridge regression. When given access to the knowledge of the full distribution $\rho$, the estimator in (8) can be rewritten as

$$f_{\infty, \lambda} = \mathbb{E}[Y \alpha_X], \qquad \alpha_X : x \to (K + \lambda I)^{-1} k(X, x). \qquad (17)$$

As such, kernel ridge regression can be seen as learning in an unsupervised fashion the weights $\alpha : \mathcal{X} \to L^2(\rho)$, which then indicate how to fold the input space to use information provided by the labels. At a high level, one can think of a scheme, given some input points, to perform finite differences and leverage the result to build an estimate of the target function from Taylor expansions, whatsoever would be the label observations. Figure 5 shows how, when $\lambda$ is not too big, the reconstruction $f_{\infty, \lambda}(x_0)$ ($x_0$ being the same point at the bluest center on the different pictures on this Figure) depends on observations made far away from $x_0$ according to some periodic pattern, implicitly assuming that the target function should be regular when looked at in the Fourier domain. Similarly, one can look at the weights $\widehat{\alpha}_X$ satisfying $\mathbb{E}_{\mathcal{D}_n}[f_n(x)] = \mathbb{E}_{(X,Y)}[\widehat{\alpha}_X(x)Y]$, and whose closed form is given in Appendix C.4. Those weights are shown on Figures 6. They present weird behaviors when the number of data $n$ is closed to the search space size $\mathcal{N}_2(K)$. Note that this double descent phenomenon actually takes place in the regularized setup, and not in the interpolation regime, contrarily to prior works on the matter [e.g. 28, 21].

## 4 Conclusion

In this paper, we have shown how subtle is the saying that smoothness allows to break the curse of dimensionality. In essence, without implicit bias and in presence of noise, one needs to be in the high-sample regime where the size of the search space $\mathcal{F}$ is smaller than the number of samples to avoid overfitting. As the input dimension grows, many more smooth functions can be defined. This constrains the diversity of functions within $\mathcal{F}$, which will typically be devoid of fine-grained details (linked with high-order, eventually trigonometric, polynomials), hence unable to harness high-order smoothness without accessing a large number of samples $n$.

**Future work.** Since we have shown that smoothness alone is not a strong enough prior to build efficient learning algorithms in high-dimensions, other priors could be investigated. As such, sparsity assumptions, multi-index models, feature learning or multi-scale behaviors might offer more realistic models to break the curse of dimensionality. How deep learning models exploit such priors has been an active line of research, although linking theoretical results with "interpretable" observations in neural networks remains challenging, and theory has not yet provided that many meaningful insights for practitioners.

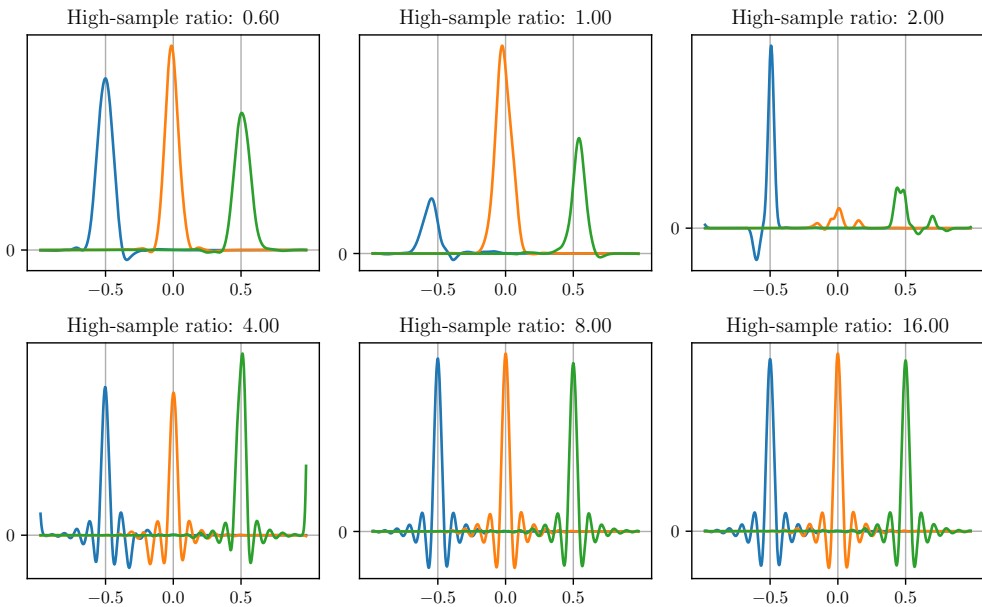

**Figure 6:** Weights $x \mapsto \widehat{\alpha}_x(x_0)$ such that $\mathbb{E}_{\mathcal{D}_n}[f_n(x)] = \mathbb{E}_{(X,Y)}[\hat{\alpha}_X(x)Y]$, for $x_0 = -1/2$ (blue), $x_0 = 0$ (orange) and $x_0 = 1/2$ (green), when $\mathcal{X} = [-1,1]$ and $\rho_{\mathcal{X}}$ is uniform. The weight $\alpha_x(x_0)$ quantifies how much the prediction at $x_0$ is inferred from the output observed at the point $x$. The weights are computed with the Gaussian kernel with bandwidth $\sigma = .1$ and $\lambda = 10^{-5}$, which yields an effective dimension $\mathcal{N}_2(K) = 45$, and for $n \in \{27, 45, 90, 180, 360, 720\}$, which explains the high-sample ratio $n/\mathcal{N}_2(K)$ seen on the title of the different plots. When this ratio is close to one, the weights present weird behaviors, which could explain the bad performance when transitioning from low-sample to high-sample regimes.

Furthermore, going beyond the sole selection of a few hyperparameters, it would be interesting to understand more aggressive model selection. In particular, given some observations $(X_i, Y_i)$ and some hypothesis classes $(\mathcal{F}_t)_t$, it seems natural to trade a term that fits the data as per (7), together with a regularization term $\min_t \|f\|_{\mathcal{F}_t}$ that selects $\mathcal{F}_t$ so that $f_n$ has a small $\mathcal{F}_t$ norm. We understand this as a *lex parsimoniae*, where each $\mathcal{F}_t$ encodes different notions of simplicity (e.g. different priors) while $f_n$ only needs to satisfy one of them.

Finally, while this work heavily relies on the least-square loss, practitioners tend to favor other losses such as the cross-entropy. How losses deform and modify the size of the search space $\mathcal{F}$ and its adherence properties to some target functions $f^*$ is an open-question –not to mention its adherence properties when the final predictor is built as a decoding $y(x) = \arg\max_y f(y \mid x)$ in order to learn a discrete $y$ from a score $f(y \mid x)$ that relates to $\mathbb{P}_{(X,Y)\sim\rho}(Y = x \mid X = x)$.

**Experiments reproduction.** All the code to run figures is available at `https://github.com/facebookresearch/rates`.

**Acknowledgements.** VC would like to thank Alberto Bietti, Jaouad Mourtada and Francis Bach for useful discussions. SV was partially supported by the MUR Excellence Department Project MatMod@TOV awarded to the Department of Mathematics, University of Rome Tor Vergata.

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

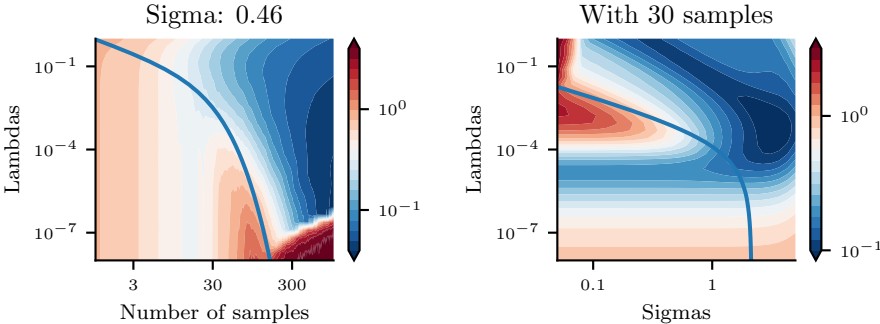

**Figure 7:** Two other cuts of Figure 1.

## A Generic Proofs and Discussions

### A.1 What do we mean by Transitory Regimes?

In essence, by transitory regimes we mean any finite-time behavior that does not match an expected long-time horizon "stationary" behavior. More precisely, let $\Gamma = \{(n, \mathbb{E}_{\mathcal{D}_n}[\mathcal{E}(f_n)]) \mid n \in \mathbb{N}\}$ be the graph of the expected excess risk. Theorem 4 provides a lower-upper bound of the form $\Gamma \subset \{(n, cn^{-\gamma}(1 + ah(n))) \mid n \in \mathbb{N}, a \in [-1, 1]\}$ with $c, \gamma$ two constants and $h$ a function that goes to zero when its argument goes to infinity. This shows that, as $n$ grows large, $\mathbb{E}_{\mathcal{D}_n}[\mathcal{E}(f_n)]$ will behave as $cn^{-\gamma}$. However, this stationary behavior in $cn^{-\gamma}$ might take time to kick in, and when only accessing a small number of samples $n$, our bound does not lead to strong constraints on $\mathbb{E}_{\mathcal{D}_n}[\mathcal{E}(f_n)]$, which might arguably exhibit a very different profile. We illustrate this idea on Figure 3.

### A.2 Generic Lower Bounds

Let us recall a relatively standard lower bound for linear regression.

**Theorem 5** (Linear regression minimax rates, [17])**.** *Let $\rho_\theta \in \Delta_{\mathcal{X} \times \mathcal{Y}}$ with $\mathcal{X} = \mathbb{R}^d$ and $\mathcal{Y} = \mathbb{R}$ be defined with its marginal over $\mathcal{X}$ being $\rho_{\mathcal{X}}$, and its conditionals being $(Y \mid X) = \theta^\top X + \varepsilon^2 \sim \mathcal{N}(0, 1)$ with $\mathcal{N}$ the Gaussian distribution. Let $\mathcal{D}_n \sim \rho_\theta^{\otimes n}$ be a dataset of $n$ samples $(X_i, Y_i)$, and $\mathcal{A}$ be an algorithm that map $\mathcal{D}_n$ to a guess for $\theta$. Under the assumption that $n > d$ and that $\sum X_i X_i^\top$ is almost always invertible, we have*

$$\sup_\theta \mathbb{E}_{\mathcal{D}_n \sim \rho_\theta^{\otimes n}} \left[ \|\theta - \mathcal{A}(\mathcal{D}_n)\|^2 \right] \geq \frac{\varepsilon^2 d}{n - d + 1}.$$

Replacing $X$ by $\varphi(X)$, we get the following corollary.

**Corollary 6.** *For any learning algorithm $\mathcal{A}$ targeting a function $f^*(X) = \varphi(X)^\top \theta$ with $\varphi : \mathcal{X} \to \mathbb{R}^D$ and $\theta \in \mathbb{R}^D$, there exists a $\theta$ and a distribution $\rho_{\mathcal{X}}$ on $X$ such that under Assumption 2,*

$$\mathbb{E}_{\mathcal{D}_n \sim \rho} \left[ \|\theta - \mathcal{A}(\mathcal{D}_n)\|^2 \right] \geq \frac{\varepsilon^2 \dim \operatorname{Span} \varphi(\mathcal{X})}{n}.$$

The proof of Theorem 2 (resp. Theorem 3) follows by considering $\varphi$ to be the concatenation of all the $\binom{d+\alpha}{\alpha}$ polynomials of degree at most $\alpha$ with $d$ variables (resp. all the trigonometric polynomials $x \mapsto \exp(-im^\top x)$ with $\|m\|_\infty \leq \omega$). To be totally rigorous, one should incorporate the assumptions of Theorem 5, yet if we remove the extra assumption, one can actually show that the worse excess risk could be infinite [17, Proposition 1], so this does not cast shadow on our results, but would only make them stronger. We choose to present weaker results that are easier to understand and parse for the reader.

Those theorems could be proven from scratch by considering a well-thought Bayesian prior on $\theta$ that has generated the dataset $\mathcal{D}_n$, before lower bounding

$$\inf_{\mathcal{A}} \sup_{\theta} \mathbb{E}_{\mathcal{D}_n} \left[ \|\mathcal{A}(\mathcal{D}_n) - \theta\|^2 \,\Big|\, \theta \right] \geq \inf_{\mathcal{A}} \mathbb{E}_{\theta} \mathbb{E}_{\mathcal{D}_n} \left[ \|\mathcal{A}(\mathcal{D}_n) - \theta\|^2 \,\Big|\, \theta \right]$$

$$= \inf_{\mathcal{A}} \mathbb{E}_{(\theta, \mathcal{D}_n)} \left[ \|\mathcal{A}(\mathcal{D}_n) - \theta\|^2 \right]$$

$$= \mathbb{E}_{(\theta, \mathcal{D}_n)} \left[ \|\mathbb{E}\left[ \theta \,|\, \mathcal{D}_n \right] - \theta\|^2 \right],$$

and computing the last term explicitly. We conjecture that it should be equally possible to compute with this technique a precise "exponentially bad" lower bound on the optimal minimax constants appearing in front of $n^{-2\alpha/(2\alpha+d)}$) in Theorem 1.

It is equally possible to retake the proof techniques of Theorems 2 and 3 to prove lower bound that are targeted to match the upper bound in $O(n^{-2\alpha/(2\alpha+d)})$.

**Theorem 7.** *For any RKHS $\mathcal{F}$ and distribution $\rho_{\mathcal{X}}$ such that the spectrum of the integral operator verifies* $\operatorname{spec} K = \left\{ n^{-2\alpha/d} \,\middle|\, n \in \mathbb{N} \right\}$, *for any algorithm $\mathcal{A}$ and value $D > 0$, there exists a function satisfying $\|f^*\|_{\mathcal{F}} \leq D$ such that under Assumption 2,*

$$\mathbb{E}_{\mathcal{D}_n} \left[ \mathcal{E}(\mathcal{A}(\mathcal{D}_n)) \right] \geq \frac{1}{6} \frac{\varepsilon^{4\alpha/(2\alpha+d)} D^{2d/(2\alpha+d)}}{n^{2\alpha/(2\alpha+d)}}.$$

*Proof.* The proof consists in finding a subproblem that reduces to linear regression. Under the assumptions of Theorem 5 and the additional assumption that $\|\theta^*\|_2 \leq D$, it is possible to show the minimax lower bound

$$\mathbb{E}_{\theta} \mathbb{E}_{\mathcal{D}_n \sim \rho_{\theta}^{\otimes n}} \left[ \|\theta - \mathcal{A}(\mathcal{D}_n)\|^2 \right] \geq \frac{1}{6} \min \left\{ \frac{\varepsilon^2 d}{n}, D^2 \right\},$$

for some Bayesian prior on $\theta$ [see 4, Chapter on "lower bound", section on Bayesian analysis].

Let us now consider the case where $\|\theta\| \leq D$, and $\theta$ is supported on the top-$k$ eigenvectors of $K$, we got $\|f^*\| = \|\Sigma^{-1/2}\theta\| \leq D\lambda_k^{-1/2}$ where $\lambda_k$ is the $k$-th eigenvalue of $T$. Hence, considering an RKHS $\mathcal{F}$, under Assumption 2, for any algorithm $\mathcal{A}$, there exists a function such that $\|f^*\|_{\mathcal{F}} \leq D$, and

$$\mathbb{E}_{\mathcal{D}_n} \left[ \mathcal{E}(\mathcal{A}(\mathcal{D}_n)) \right] \geq \frac{1}{6} \max_{k \in \mathbb{N}} \min \left\{ \frac{\varepsilon^2 k}{n}, D^2 \lambda_k \right\},$$

Considering $\lambda_k = k^{-2\alpha/d}$ and $k = (nD^2/\varepsilon^2)^{2\alpha/(2\alpha+d)}$ leads to the result. $\qquad\square$

In terms of implications of Theorem 7, in harmonics settings, it is possible to build an integral operator such that the eigenfunctions of $K$ are known to be regular smooth functions, e.g., the trigonometric polynomials $f_m : x \to \exp(-im^\top x)$ for $m \in \mathbb{Z}^d$ ordered according to $\|m\| = \omega$. As such, for $m \in \mathbb{Z}^d$ with $\|m\| = \omega$, $f_m$ is expected to be ordered around the $\omega^d$-th eigenfunctions of $K$, typically associated with the eigenvalue $\lambda_{\omega^d} \approx (\omega^d)^{-2\alpha/d} = \omega^{-2\alpha}$ if $\mathcal{F}$ corresponds to $H^\alpha$. Using that when $f^*$ is the $k$-th eigenfunctions of $K$, we have $\|f^*\| = \lambda_k^{-1}$, Theorem 3 would become that for any algorithm $\mathcal{A}$, under Assumption 2, there exists a function $f^*$ such that $\|f^*\|_{\mathcal{F}} \leq \|f_m\|_{\mathcal{F}}$ with $\|m\| = \omega$ and,

$$\mathbb{E}_{\mathcal{D}_n} \left[ \mathcal{E}(\mathcal{A}(\mathcal{D}_n)) \right] \gtrsim \frac{\varepsilon^{4\alpha/(2\alpha+d)} \omega^{2d\alpha/(2\alpha+d)}}{n^{2\alpha/(2\alpha+d)}}.$$

When $\alpha$ scales linearly with $d$ (so to beat the curse of dimensionality), this lower bound presents a constant that is exponential with $d$.

## A.3 Precise Excess Risk Bound

This subsection is devoted to the proof of the bound in Theorem 4.

For ease of notation, we will use the finite-dimensional notation $u^\top w$ to denote the inner product $\langle u, v \rangle$ in (infinite-dimensional) Hilbert spaces. Moreover, we will simply write $\|\cdot\|$ for both $\|\cdot\|_{\mathcal{H}}$

and the operator norm on $\mathcal{H}$ (depending on context), $L^2$ for $L^2(\rho_{\mathcal{X}})$, and $\|\cdot\|_2$ for both $\|\cdot\|_{L^2}$ and the operator norm on $L^2$.

For the sake of clarity, we have expressed all our statements in terms of operators on $L^2$. For the proofs, it is more convenient to work in $\mathcal{H}$. Let us introduce the embedding

$$S : \mathcal{H} \to L^2, \qquad \theta \mapsto (x \mapsto \theta^\top \varphi(x)).$$

From $S$, one can take its adjoint $S^*$ and check that $K = SS^*$. $K$ is isometric to the (non-centered) covariance operator

$$\Sigma = S^*S = \mathbb{E}[\varphi(X) \otimes \varphi(X)].$$

Note that

$$\|S\theta\|_2^2 = \mathbb{E}[(\varphi(X)^\top \theta)^2] \leq \mathbb{E}[\|\varphi(X)\|^2 \|\theta\|^2] = \|\varphi\|_2^2 \|\theta\|^2,$$

which implies that $K$ is a continuous operator as soon as $\varphi \in L^2$. The kernel ridge regression estimator (8) is characterized as

$$f_n = S(\Sigma_n + 1)^{-1}S_n^* \mathbb{Y},$$

where

$$S_n : \mathcal{H} \to \mathbb{R}^n, \ \theta \mapsto (\theta^\top \varphi(X_i))_{i \in [n]}, \qquad \Sigma_n = S_n^* S_n, \qquad \mathbb{Y} = (Y_i)_{i \in [n]} \in \mathbb{R}^n.$$

Endowing $\mathbb{R}^n$ with the scalar product $\langle a, b \rangle = \frac{1}{n} \sum_{i \in [n]} a_i b_i$, we have

$$S_n^* \mathbb{Y} = \frac{1}{n} \sum_{i \in [n]} Y_i \varphi(X_i), \qquad \Sigma_n = \frac{1}{n} \sum_{i=1}^n \varphi(X_i) \otimes \varphi(X_i).$$

It is useful to define $\varepsilon_i$ as the difference between $Y_i$ and $f^*(X_i) = \mathbb{E}[Y_i \,|\, X = X_i]$, which can be seen as the labeling noise and average to zero. For simplicity, we will first assume that our model is well-specified, i.e. $f^* = S\theta_*$ for some $\theta_*$. We have, with $E = (\varepsilon_i)_{i \in [n]} \in \mathbb{R}^n$,

$$Y_i = f^*(X_i) + \varepsilon_i = \varphi(X_i)^\top \theta_* + \varepsilon_i, \qquad \mathbb{Y} = S_n \theta_* + E.$$

As a consequence,

$$f_n = S(\Sigma_n + 1)^{-1}\Sigma_n \theta_* + S(\Sigma_n + 1)^{-1}S_n^* E.$$

Let $\mathbb{X} = (X_i)_{i \in [n]}$. we have

$$\begin{aligned}
\mathbb{E}_{\mathcal{D}_n}\left[\mathcal{E}(f_n) \,|\, \mathbb{X}\right] = \mathbb{E}_{\mathcal{D}_n}&\left[\|f_n - f^*\|_2^2 \,|\, \mathbb{X}\right] \\
= &\left\|S(\Sigma_n + 1)^{-1}\Sigma_n \theta_* - S\theta_*\right\|_2^2 \\
&+ \mathbb{E}_{\mathcal{D}_n}\left[\left\|S(\Sigma_n + 1)^{-1}S_n^* E\right\|_2^2 \,|\, \mathbb{X}\right] \\
&+ 2\mathbb{E}_{\mathcal{D}_n}\left[\left(S(\Sigma_n + 1)^{-1}\Sigma_n \theta_* - S\theta_*\right)^\top S(\Sigma_n + 1)^{-1}S_n^* E \,|\, \mathbb{X}\right] \\
= &\left\|S(\Sigma_n + 1)^{-1}\theta_*\right\|_2^2 \\
&+ \mathbb{E}_{\mathcal{D}_n}\left[\left\|S(\Sigma_n + 1)^{-1}S_n^* E\right\|_2^2 \,|\, \mathbb{X}\right] \\
&+ 2\left(S_n(\Sigma_n + 1)^{-1}\Sigma(\Sigma_n + 1)^{-1}\theta_*\right)^\top \mathbb{E}_{\mathcal{D}_n}[E \,|\, \mathbb{X}] \\
= &\left\|S(\Sigma_n + 1)^{-1}\theta_*\right\|_2^2 + \mathbb{E}_{\mathcal{D}_n}\left[\left\|S(\Sigma_n + 1)^{-1}S_n^* E\right\|_2^2 \,|\, \mathbb{X}\right],
\end{aligned}$$

where in the third equality we used $I - (\Sigma_n + 1)^{-1}\Sigma_n = (\Sigma_n + 1)^{-1}$, and in the last one that $\mathbb{E}_{\mathcal{D}_n}[E \,|\, \mathbb{X}] = 0$. Assuming for simplicity that the noise is homoscedastic so that $\mathbb{E}[EE^\top] = \varepsilon^2 I$ for $\varepsilon > 0$, we obtain

$$\begin{aligned}
\mathbb{E}_{\mathcal{D}_n}\left[\left\|S(\Sigma_n + 1)^{-1}S_n^* E\right\|_2^2 \,|\, \mathbb{X}\right] &= \frac{1}{n}\, \mathrm{Tr}\left(S_n(\Sigma_n + 1)^{-1}\Sigma(\Sigma_n + 1)^{-1}S_n^* \mathbb{E}_{\mathcal{D}_n}\left[EE^\top \,|\, \mathbb{X}\right]\right) \\
&= \frac{\varepsilon^2}{n}\, \mathrm{Tr}\left(\Sigma(\Sigma_n + 1)^{-2}\Sigma_n\right),
\end{aligned}$$

where the $1/n$ factor arises from the fact that $E^* = E^\top/n$ in the geometry we have considered on $\mathbb{R}^n$. Finally we have retrieved the following standard bias-variance decomposition result.

**Lemma 2** (Bias-Variance decomposition). *When our model is well-specified so that $f^* = S\theta_*$ with $\theta_* \in \mathcal{H}$, and when the noise in the label is homoscedastic with variance $\varepsilon^2$, the estimator (8) verifies*

$$\mathbb{E}_{\mathcal{D}_n}\left[\mathcal{E}(f_n) \,|\, \mathbb{X}\right] = \underbrace{\left\|S(\Sigma_n + 1)^{-1}\theta_*\right\|_2^2}_{\mathcal{B}_n} + \frac{\varepsilon^2}{n}\underbrace{\mathrm{Tr}\left(\Sigma(\Sigma_n + 1)^{-2}\Sigma_n\right)}_{\mathcal{V}_n}. \tag{18}$$

We would like to get the limit when $n$ goes to infinity in equation (18). We expect the first term to concentrate towards $\left\|S(\Sigma + 1)^{-1}\theta_*\right\|_2^2$, and the second term to $\mathrm{Tr}(\Sigma^2(\Sigma + 1)^{-2})$.

### A.3.1 Bounding the bias term

Let us begin by working out the term $\mathcal{B}_n = \left\|S(\Sigma_n + 1)^{-1}\theta_*\right\|_2^2$. We first introduce some notation to make derivations shorter. Let $E_n = \Sigma_n - \Sigma$, $\Sigma_1 = \Sigma + 1$ and $F_n = -\Sigma_1^{-1/2}E_n\Sigma_1^{-1/2}$. As long as $\|F_n\| < 1$, we have

$$\begin{aligned}
\mathcal{B}_n &= \theta_*^\top(\Sigma_n + 1)^{-1}\Sigma(\Sigma_n + 1)^{-1}\theta_* \\
&= \theta_*^\top(\Sigma_1 + E_n)^{-1}\Sigma(\Sigma_1 + E_n)^{-1}\theta_* \\
&= \theta_*^\top\Sigma_1^{-1/2}(I - F_n)^{-1}\Sigma_1^{-1}\Sigma(I - F_n)^{-1}\Sigma_1^{-1/2}\theta_* \\
&= \sum_{i,j\in\mathbb{N}}\theta_*^\top\Sigma_1^{-1/2}F_n^i\Sigma_1^{-1}\Sigma F_n^j\Sigma_1^{-1/2}\theta_*.
\end{aligned}$$

Let us assume for a moment that

$$\left\|(\Sigma\Sigma_1^{-1})^{-1/2}F_n(\Sigma\Sigma_1^{-1})^{1/2}\right\| \leq \|F_n\|. \tag{19}$$

If equation (19) holds, then

$$\begin{aligned}
\left|\mathcal{B}_n - \theta_*^\top\Sigma_1^{-2}\Sigma\theta_*\right| &= \left|\sum_{i,j\in\mathbb{N};i+j\neq 0}\theta_*^\top\Sigma_1^{-1/2}F_n^i\Sigma_1^{-1}\Sigma F_n^j\Sigma_1^{-1/2}\theta_*\right| \\
&\leq \sum_{i+j\neq 0}\left|\theta_*^\top\Sigma_1^{-1/2}F_n^i\Sigma_1^{-1}\Sigma F_n^j\Sigma_1^{-1/2}\theta_*\right| \\
&= \sum_{i+j\neq 0}\left|\left\langle\Sigma_1^{-1/2}(\Sigma_1^{-1}\Sigma)^{1/2}\theta_*, \left((\Sigma_1^{-1}\Sigma)^{-1/2}F_n^i\Sigma_1^{-1}\Sigma F_n^j(\Sigma_1^{-1}\Sigma)^{-1/2}\right)(\Sigma_1^{-1}\Sigma)^{1/2}\Sigma_1^{-1/2}\theta_*\right\rangle\right| \\
&\leq \sum_{i+j\neq 0}\left\|\Sigma_1^{-1/2}(\Sigma_1^{-1}\Sigma)^{1/2}\theta_*\right\|^2\left\|(\Sigma_1^{-1}\Sigma)^{-1/2}F_n^i\Sigma_1^{-1}\Sigma F_n^j(\Sigma_1^{-1}\Sigma)^{-1/2}\right\| \\
&= \theta_*^\top\Sigma_1^{-2}\Sigma\theta_*\sum_{i+j\neq 0}\left\|((\Sigma\Sigma_1^{-1})^{-1/2}F_n(\Sigma\Sigma_1^{-1})^{1/2})^{i+j}\right\| \\
&\leq \theta_*^\top\Sigma_1^{-2}\Sigma\theta_*\sum_{i,j\in\mathbb{N};i+j\neq 0}\left\|(\Sigma\Sigma_1^{-1})^{-1/2}F_n(\Sigma\Sigma_1^{-1})^{1/2}\right\|^{i+j} \\
&= \theta_*^\top\Sigma_1^{-2}\Sigma\theta_*\sum_{i\in\mathbb{N}}(i+2)\left\|(\Sigma\Sigma_1^{-1})^{-1/2}F_n(\Sigma\Sigma_1^{-1})^{1/2}\right\|^{i+1} \\
&\leq \theta_*^\top\Sigma_1^{-2}\Sigma\theta_*\sum_{i\in\mathbb{N}}(i+2)\|F_n\|^{i+1} = \theta_*^\top\Sigma_1^{-2}\Sigma\theta_*\int_0^\infty(\lfloor x\rfloor + 2)\|F_n\|^{\lceil x\rceil}\,\mathrm{d}x \\
&\leq \theta_*^\top\Sigma_1^{-2}\Sigma\theta_*\int_0^\infty(x+2)\|F_n\|^x\,\mathrm{d}x = \theta_*^\top\Sigma_1^{-2}\Sigma\theta_*\frac{1 - 2\log(\|F_n\|)}{\log^2(\|F_n\|)}.
\end{aligned}$$

This inequality is useful as long as $\|F_n\|$ is small enough, which is not always true. When $\|F_n\|$ is large, we can instead proceed with the simpler bound

$$\left|\mathcal{B}_n - \theta_*^\top\Sigma_1^{-2}\Sigma\theta_*\right| \leq \mathcal{B}_n + \theta_*^\top\Sigma_1^{-2}\Sigma\theta_* \leq 2\theta_*^\top\Sigma\theta_* = 2\|f^*\|_2^2.$$

Therefore, rewriting the limit as

$$\theta_*^\top (\Sigma + 1)^{-2} \Sigma \theta_* = (\Sigma^{1/2}\theta_*)^\top (\Sigma + 1)^{-2} \Sigma^{1/2}\theta_* = (S\theta_*)^\top (K+1)^{-2} S\theta_*$$
$$= (f^*)^\top (K+1)^{-2} f^* = \left\| (K+1)^{-1} f^* \right\|_2^2 = \mathcal{S}(K),$$

we split the bias error as

$$\left| \mathcal{B}_n - \theta_*^\top \Sigma_1^{-2} \Sigma \theta_* \right| \leq 2 \left\| f^* \right\|_2^2 \mathbf{1}_{\|F_n\|>1/2} + \mathcal{S}(K) \frac{1 - 2\log(\|F_n\|)}{\log^2(\|F_n\|)} \mathbf{1}_{\|F_n\|\leq 1/2}$$

$$\leq 2 \left\| f^* \right\|_2^2 \mathbf{1}_{\|F_n\|>1/2} - \frac{3\mathcal{S}(K)}{\log(\|F_n\|)} \mathbf{1}_{\|F_n\|\leq 1/2}.$$

Taking the expectation and using the convexity of the absolute value, we obtain

$$\left| \mathbb{E}_{\mathcal{D}_n}[\mathcal{B}_n] - \theta_*^\top \Sigma_1^{-2} \Sigma \theta_* \right| \leq 2 \left\| f^* \right\|_2^2 \mathbb{P}(\|F_n\| > 1/2) - \mathcal{S}(K) \int_0^{1/2} \frac{3\mathbb{P}\left(\|F_n\| > x\right)}{\log(x)} \, \mathrm{d}x.$$

We now proceed with an exponential concentration inequality on $\|F_n\|$. We will use the one of Cabannes et al. [5], Eq. (25). As long as $1 \leq \|K\|$, we have

$$\mathbb{P}(\|F_n\| > t) \leq 28\mathcal{N}_1(K) \exp\left( -\frac{nt^2}{\mathcal{N}_+(K)(1+t)} \right).$$

This inequality shows the restrictive notion of effective dimension, which is useful to ensure the good conditioning of the linear system implicitly encoded in (8), and, in essence, bound all moments of $\varphi(X)$. As long as $\mathcal{N}_+(K) \leq 3n/2$, we can compute the integral as

$$-\int_0^{1/2} \frac{\mathbb{P}\left(\|F_n\| > x\right)}{\log(x)} \, \mathrm{d}x \leq -28\mathcal{N}_1(K) \int_0^{1/2} \frac{\exp(-3nx^2/2\mathcal{N}_+(K))}{\log(x)} \, \mathrm{d}x$$

$$= -28\mathcal{N}_1(K) \frac{\mathcal{N}_+^{1/2}(K)}{1.5^{1/2}n^{1/2}} \int_0^{1/2} \frac{\exp(-u^2)}{\log(u) + \log(3n/2\mathcal{N}_\infty^2(K))} \, \mathrm{d}u$$

$$\leq -28 \frac{\mathcal{N}_1(K)\mathcal{N}_+^{1/2}(K)}{1.5^{1/2}n^{1/2}} \int_0^{1/2} \frac{\exp(-u^2)}{\log(u)} \, \mathrm{d}u \leq \frac{8\mathcal{N}_1(K)\mathcal{N}_+^{1/2}(K)}{n^{1/2}}.$$

We recall that the bounds above were derived under condition (19), which is rather strong. However, the attentive reader would remark that a much laxer assumption is sufficient, which we introduce thereafter.

**Assumption 3.** *There exists a constant $c$ such that, for all $i, j \in \mathbb{N}$,*

$$\mathbb{E}\left[ \left\| (\Sigma\Sigma_1^{-1})^{-.5} F_n^i \Sigma\Sigma_1^{-1} F_n^j (\Sigma\Sigma_1^{-1})^{-.5} \right\| \, \Big| \, \|F_n\| \leq 1/2 \right] \leq c^2 \mathbb{E}\left[ \|F_n\|^{i+j} \, \Big| \, \|F_n\| \leq 1/2 \right]. \quad (20)$$

Assumption 3 notably holds when $\mathcal{F}$ is finite dimensional with $c^2 = \left\| K^{-1} \right\|_2^{-1} (\|K\|_2 + 1)$. As such all our lower-bound results can be cast with finite-dimensional approximation of infinite-dimensional RKHS. Under Assumption 3, we get

$$\mathbb{E}\left[ \left| \mathcal{B}_n - \theta_*^\top \Sigma_1^{-2} \Sigma \theta_* \right| \, \Big| \, \|F_n\| \leq 1/2 \right]$$

$$\leq \sum_{i+j\neq 0} \left\| \Sigma_1^{-1/2} (\Sigma_1^{-1}\Sigma)^{1/2} \theta_* \right\|^2 \mathbb{E}\left[ \left\| (\Sigma_1^{-1}\Sigma)^{-1/2} F_n^i \Sigma_1^{-1} \Sigma F_n^j (\Sigma_1^{-1}\Sigma)^{-1/2} \right\| \, \Big| \, \|F_n\| \leq 1/2 \right]$$

$$\leq c^2 \sum_{i,j\in\mathbb{N}; i+j\neq 0} \left\| \Sigma_1^{-1/2} (\Sigma_1^{-1}\Sigma)^{1/2} \theta_* \right\|^2 \mathbb{E}\left[ \|F_n\|^{i+j} \, \Big| \, \|F_n\| \leq 1/2 \right],$$

which allows us to proceed with the precedent derivations without assuming that (19) holds.

While the previous results were achieved for $f^* \in \mathcal{F}$, they can be extended by density to any $f^*$ in the closure of $\mathcal{F}$ in $L^2(\rho_{\mathcal{X}})$, i.e. $f \in (\ker K)^\perp$, leading to the following result.

**Proposition 3.** *When $f^* \in (\ker K)^\perp$ and $1 \leq \|\Sigma\|$, under the technical Assumption 3, the bias term can be bounded from above and below by*

$$\left| \mathbb{E}_{\mathcal{D}_n}[\mathcal{B}_n] - \mathcal{S}(K) \right| \leq c^2 \mathcal{N}_1(K) \left( 56 \left\| f^* \right\|_2^2 \exp\left( -\frac{n}{6\mathcal{N}_+(K)} \right) + \frac{8\mathcal{S}(K)\mathcal{N}_+^{1/2}(K)}{n^{1/2}} \right). \quad (21)$$

### A.3.2 Discussion on the bias bound

**More direct upper bound.** The precedent derivations can be made more direct with the following series of implications, with $A \preceq B$ meaning that $x^\top A x \le x^\top B x$ for every $x$:

$$\|F_n\| \le 1/2 \quad \Rightarrow \quad -1/2I \preceq F_n \preceq 1/2I$$
$$\Rightarrow \quad 1/2I \preceq I - F_n \preceq 3/2I$$
$$\Rightarrow \quad 4/9I \preceq (I - F_n)^{-2} \preceq 4I$$
$$\Rightarrow \quad 4/9\mathcal{S}(K) \preceq \theta_*^\top \Sigma^{1/2}\Sigma_1^{-1}(I - F_n)^{-2}\Sigma_1^{-1}\Sigma^{1/2}\theta_* \preceq 4\mathcal{S}(K).$$

Let us assume that

$$\mathbb{E}\left[(I - F_n)^{-1}\Sigma\Sigma_1^{-1}(I - F_n)^{-1} \mid \|F_n\| \le 1/2\right]$$
$$\preceq c^2\mathbb{E}\left[\Sigma^{1/2}\Sigma_1^{-1/2}(I - F_n)^{-2}\Sigma^{1/2}\Sigma_1^{-1/2} \mid \|F_n\| \le 1/2\right], \tag{22}$$

This leads to the simple upper bound

$$\mathcal{B}_n = \theta_*^\top \Sigma_1^{-1/2}(I - F_n)^{-1}\Sigma_1^{-1}\Sigma(I - F_n)^{-1}\Sigma_1^{-1/2}\theta_*$$
$$\le c^2\theta_*^\top \Sigma_1^{-1/2}(\Sigma_1^{-1})^{1/2}(I - F_n)^{-2}(\Sigma_1^{-1})^{1/2}\Sigma_1^{-1/2}\theta_* + \|f^*\|_{L^2}^2 \, \mathbb{P}(\|F_n\| > 1/2)$$
$$\le 4c^2\mathcal{S}(K) + \|f^*\|_2^2 \mathcal{N}_1(K)\exp\left(-\frac{n}{6\mathcal{N}_+(K)}\right).$$

**Improvement directions.** In essence, we expect the bias upper-lower bound to behave as

$$\mathcal{S}(K)(I - F_n)^{-2} - I \simeq \mathcal{S}(K)F_n.$$

Getting this linear dependency in $F_n$ explicitly would allow to improve the bound since

$$\mathbb{E}_{\mathcal{D}_n}\left[\|F_n\| \mid \|F_n\| \le 1/2\right] \lesssim \mathcal{N}(K)\mathcal{N}_+(K)n^{-1}.$$

Moreover, going back to the definition of $F_n$,

$$\mathbb{E}_{\mathcal{D}_n}\theta_*^\top \Sigma\Sigma_1^{-1}F_n\Sigma\Sigma_1^{-1}\theta_* = \mathbb{E}_{\mathcal{D}_n}\left[\left(\frac{1}{n}\sum_{i=1}^n \theta_*^\top \Sigma\Sigma_1^{-3/2}\varphi(X_i)\right)^2\right] - \mathbb{E}_X\left[\theta_*^\top \Sigma\Sigma_1^{-3/2}\varphi(X)\right]^2$$
$$= \mathbb{E}_{\mathcal{D}_n}\left[\left(\frac{1}{n}\sum_{i=1}^n \theta_*^\top \Sigma\Sigma_1^{-3/2}\varphi(X_i) - \mathbb{E}_X\left[\theta_*^\top \Sigma\Sigma_1^{-3/2}\varphi(X)\right]\right)^2\right],$$

which suggests possible improvements of the bound in $\mathcal{S}(K)n^{-1}$.

### A.3.3 Bounding the variance term

Let us now work on the term $\mathcal{V}_n = \text{Tr}\,\Sigma(\Sigma_n + 1)^{-2}\Sigma_n$. It works similarly to the bias term, concentrating towards $\mathcal{N}_2(K)$. With $\Sigma_{n,1} = \Sigma_n + 1$, we have

$$\text{Tr}\left(\Sigma\Sigma_n\Sigma_{n,1}^{-2} - \Sigma^2\Sigma_1^{-2}\right) = \text{Tr}\left(\Sigma\Sigma_1^{-1}\Sigma_n\Sigma_{n,1}^{-1}(\Sigma_{n,1}^{-1}\Sigma_1 - I) + \Sigma\Sigma_1^{-1}(\Sigma_n\Sigma_{n,1}^{-1} - \Sigma\Sigma_1^{-1})\right).$$

Using that, for any $A$ positive semi-definite and any $B$, $\text{Tr}(AB) \le \|B\|\,\text{Tr}(A)$, it follows that

$$|\mathcal{V}_n - \mathcal{N}_2(K)| \le \mathcal{N}_1(K)\left\|\Sigma_n\Sigma_{n,1}^{-1}\right\|\left\|\Sigma_{n,1}^{-1}\Sigma_1 - I\right\| + \mathcal{N}_1(K)\left\|\Sigma_n\Sigma_{n,1}^{-1} - \Sigma\Sigma_1^{-1}\right\|$$
$$\le \mathcal{N}_1(K)\left\|\Sigma_{n,1}^{-1}\Sigma_1 - I\right\| + \mathcal{N}_1(K)\left\|\Sigma_n\Sigma_{n,1}^{-1} - \Sigma\Sigma_1^{-1}\right\|.$$

Let us focus on the first term. Using $a^{-1} - b^{-1} = a^{-1}(b - a)b^{-1}$, we get

$$\left\|\Sigma_1\Sigma_{n,1}^{-1} - I\right\| = \left\|\Sigma_1\Sigma_{n,1}^{-1}(\Sigma - \Sigma_n)\Sigma_1^{-1}\right\|$$
$$\le \left\|\Sigma_1\Sigma_{n,1}^{-1}\right\|\left\|\Sigma_1^{1/2}F_n\Sigma_1^{-1/2}\right\|$$
$$\le \left(\|I\| + \left\|\Sigma_1\Sigma_{n,1}^{-1} - I\right\|\right)\left\|\Sigma_1^{1/2}F_n\Sigma_1^{-1/2}\right\|$$
$$\le \sum_{i>0}\left\|\Sigma_1^{1/2}F_n\Sigma_1^{-1/2}\right\|^i.$$

For the second term,

$$\left\|\Sigma_n \Sigma_{n,1}^{-1} - \Sigma \Sigma_1^{-1}\right\| \le \left\|\Sigma_n (\Sigma_{n,1}^{-1} - \Sigma_1^{-1}\right\| + \left\|(\Sigma_n - \Sigma)\Sigma_1^{-1}\right\|$$
$$\le \left\|\Sigma_n \Sigma_{n,1}^{-1}(\Sigma - \Sigma_n)\Sigma_1^{-1}\right\| + \left\|(\Sigma_n - \Sigma)\Sigma_1^{-1}\right\|$$
$$\le 2\left\|(\Sigma_n - \Sigma)\Sigma_1^{-1}\right\|$$
$$= 2\left\|\Sigma_1^{1/2} F_n \Sigma_1^{-1/2}\right\|.$$

Similarly as for (19), if we assume that

$$\left\|\Sigma_1^{1/2} F_n \Sigma_1^{-1/2}\right\| \le c\,\|F_n\|, \tag{23}$$

then we have, as long as $\|F_n\| \le 1/2c$,

$$|\mathcal{V}_n - \mathcal{N}_2(K)| \le \sum_{i>0} \left\|\Sigma_1^{1/2} F_n \Sigma_1^{-1/2}\right\|^i + 2\left\|\Sigma_1^{1/2} F_n \Sigma_1^{-1/2}\right\|$$
$$\le \frac{c\,\|F_n\|}{1 - c\,\|F_n\|} + 2c\,\|F_n\| \le 4c\,\|F_n\|.$$

For the case when $\|F_n\|$ is large, we can again proceed with a simple bound:

$$|\mathcal{V}_n - \mathcal{N}_2(K)| \le \left|\mathrm{Tr}\left(\Sigma \Sigma_n (\Sigma_n + 1)^{-2}\right)\right| + \mathrm{Tr}\left(\Sigma^2 \Sigma_1^{-2}\right) \le 2\,\mathrm{Tr}(\Sigma).$$

Splitting the full expectation as we did for the bias we thus obtain

$$|\mathbb{E}_{\mathcal{D}_n}[\mathcal{V}_n] - \mathcal{N}_2(K)| \le \mathbb{E}_{\mathcal{D}_n}[|\mathcal{V}_n - \mathcal{N}_2(K)|]$$
$$\le 2\,\mathrm{Tr}(\Sigma)\mathbb{P}(\|F_n\| > 1/2c) + 4c\mathbb{E}\left[\|F_n\|\,\big|\,\|F_n\| \le 1/2c\right]\mathbb{P}(\|F_n\| \le 1/2c).$$

We are left with the computation of two integrals. As long as $1 \le \|\Sigma\|$, with $a$ the coefficient appearing in the exponential

$$\mathbb{P}(\|F_n\| \le 1/2c)\mathbb{E}\left[\|F_n\|\,\big|\,\|F_n\| \le 1/2c\right] \le 28\mathcal{N}_1(K)\int_0^{1/2c} x\exp(-3nx^2/2\mathcal{N}_+(K))\,\mathrm{d}x$$
$$= 28\mathcal{N}_1(K)a^{-1}\int_0^{a^{1/2c}/2} x\exp(-x^2)\,\mathrm{d}x \le \frac{10\mathcal{N}_1(K)\mathcal{N}_+(K)}{n}.$$

Once again, the condition (23) can be relaxed with the following assumption.

**Assumption 4.** *There exists a constant c such that, for all $i \in \mathbb{N}$,*

$$\mathbb{E}\left[\left\|\Sigma_1^{1/2} F_n \Sigma_1^{-1/2}\right\|^i \,\bigg|\, \|F_n\| \le 1/2c\right] \le c^i \mathbb{E}\left[\|F_n\|^i \,\big|\, \|F_n\| \le 1/2c\right]. \tag{24}$$

As for Assumption 3, Assumption 4 holds when $\mathcal{F}$ is finite-dimensional with $c^2 = \|K^{-1}\|\,(\|K\|+1)$. It holds in general with $c^2 = \|\Sigma\| + 1$, although this would deteriorate the bound by a factor of $\lambda$ when considering $\Sigma$ to be $\lambda^{-1}\Sigma$.

We are finally ready to collect the different pieces.

**Proposition 4.** *When $f^* \in (\ker K)^\perp$ and $1 \le \|\Sigma\|$, under the technical Assumption 4, the variance term can be bounded from above and below by*

$$\frac{\varepsilon^2}{n}\,|\mathbb{E}_{\mathcal{D}_n}[\mathcal{V}_n] - \mathcal{N}_2(K)| \le \varepsilon^2 \mathcal{N}_1^2(K)\left(\frac{28\,\mathrm{Tr}(\Sigma)}{n}\exp\left(-\frac{c^2 n}{(4 + 2c)\mathcal{N}_+(K)}\right) + \frac{40c\mathcal{N}_+(K)}{n^2}\right). \tag{25}$$

### A.3.4   Discussion to the variance bound

Once again, the bound presented here is somewhat unsatisfying, as it will not necessarily decrease faster than $\mathcal{N}(K)/n$. If one considers target functions that are far away from $\mathcal{F}$ and requires a large search space (i.e. $\mathcal{S}(\lambda^{-1}K)$ decreases slowly with $\lambda$, and given a fixed number of samples $n$ the

optimal $\lambda_n$ is found for $\mathcal{N}(\lambda_n K)$ quite large compared to $n$), so that $\mathcal{N}_+(\lambda^{-1}K)\mathcal{N}(\lambda^{-1}K)/n$ does not go to zero. Several directions can be taken to improve the bound. For example, when $Y$ is bounded by $M$, the noise $\varepsilon^2$ can be replaced by $M^2$, and it is possible to get an upper bound of the form

$$\mathbb{E}_{\mathcal{D}_n}[\mathcal{E}(f_n^{(\text{thres.})})] \leq \frac{8M^2}{n}\mathcal{N}_1(K) + \inf_{f \in \mathcal{F}}\left(\|f - f^*\|_{L^2}^2 + \|f\|_{\mathcal{F}}^2\right),$$

for a truncated version $f_n^{(\text{thres.})}$ of the estimator (8) as proved by Mourtada et al. [19] for ridge-less regression and extended to ridge regression in Mourtada et al. [20]. Moreover, retaking the analysis of Mourtada and Rosasco [18], one can get a lower bound of the form

$$\mathbb{E}_{\mathcal{D}_n}\mathcal{V}_n \geq \frac{n}{n+1}\mathbb{E}_{\mathcal{D}_{n+1}}\operatorname{Tr}\left(\Sigma_n\Sigma_{n,1}^{-1}\Sigma_{n+1}\Sigma_{n+1,(n+1)/n}^{-1}\right)$$

$$\geq \frac{n}{n+1}\mathbb{E}_{\mathcal{D}_n}\operatorname{Tr}\left(\Sigma_n\Sigma_{n,1}^{-1}\right) - \mathbb{E}_{\mathcal{D}_{n+1}}\operatorname{Tr}\left(\Sigma_n\Sigma_{n,1}^{-1}\Sigma_{n+1,(n+1)/n}^{-1}\right),$$

which might lead to some lower bound with

$$\mathbb{E}_{\mathcal{D}_{n+1}}\operatorname{Tr}\left(\Sigma_n\Sigma_{n,1}^{-1}\Sigma_{n+1,(n+1)/n}^{-1}\right) = \frac{n+1}{n}\mathbb{E}_{\mathcal{D}_n,X}\operatorname{Tr}\left(\Sigma_n\Sigma_{n,1}^{-1}(\Sigma_{n,1} + \varphi(X) \otimes \varphi(X))^{-1}\right)$$

$$\leq \frac{n+1}{n}\mathbb{E}_{\mathcal{D}_n}\operatorname{Tr}\left(\Sigma_n\Sigma_{n,1}^{-1}\right)\mathbb{E}_X\left[\|(\Sigma_{n,1} + \varphi(X) \otimes \varphi(X))^{-1}\|\right].$$

We also note that it should not be too hard to replace $F_n$ by $\Sigma^{1/2}\Sigma_1^{-1}E_n\Sigma^{1/2}\Sigma_1^{-1}$, which would lead to $\varepsilon^2\mathcal{N}_2(K)/n$ instead of $\varepsilon^2\mathcal{N}_1(K)/n$ in the right-hand side of (11).

### A.3.5 Full theorem

Collecting the precedent results leads to the following theorem.

**Theorem 8.** *Under the technical Assumptions 3 and 4, as long as $1 \leq \|\Sigma\|$, when $f^*$ belongs to the closure of $\mathcal{F}$ in $L^2(\rho_\mathcal{X})$, the estimator (8) verifies*

$$\left|\mathbb{E}_{\mathcal{D}_n}[\mathcal{E}(f_n)] - \frac{\varepsilon^2\mathcal{N}_2(K)}{n} - \mathcal{S}(K)\right| \leq 40\mathcal{N}_1(K)\left(a_n \cdot \frac{\varepsilon^2\mathcal{N}_1(K)}{n} + a_n^{1/2}\mathcal{S}(K)\right)$$

$$+ 56\mathcal{N}_1(K)\left(\frac{\operatorname{Tr}(\Sigma)\varepsilon^2}{n} + \|f^*\|_{L^2}^2\right)\exp(-ca_n), \tag{26}$$

*where $a_n = \mathcal{N}_+(K)/n$ and some constant $c$.*

As long as its right-hand side decreases faster then $\mathbb{E}_{\mathcal{D}_n}[\mathcal{E}(f_n)]$, Theorem 8 states that $\mathbb{E}_{\mathcal{D}_n}[\mathcal{E}(f_n)]$ behaves like

$$\mathbb{E}_{\mathcal{D}_n}[\mathcal{E}(f_n)] \simeq \frac{\varepsilon^2\mathcal{N}_2(K)}{n} + \mathcal{S}(K).$$

When optimizing for $\lambda$ when $K$ is actually $\lambda^{-1}K$, assuming that $\mathcal{N}_1 \simeq \mathcal{N}_2$, the right-hand side of Theorem 8 decreases faster than $\mathbb{E}_{\mathcal{D}_n}[\mathcal{E}(f_n)]$ if and only if $\mathcal{N}_1(K)a_n^{-1/2}$ goes to zero with $n$. This implies $\mathcal{N}(K)^3 \leq n$, which is a much stronger condition than the high-sample regime condition $\mathcal{N}(K) \leq n$.

### A.4 Interpolation spaces, capacity and source conditions

This section discusses the values of $\mathcal{N}(K)$ and $\mathcal{S}(K)$, and prove the last statements of Theorem 4.

### A.4.1 Variances

We begin with simple facts about the variance term.

**Proposition 5** (Relation between variances). *For any kernel $k$, we have the relation*

$$\mathcal{N}_2(K) \leq \mathcal{N}_1(K) \leq \mathcal{N}_+(K). \tag{27}$$

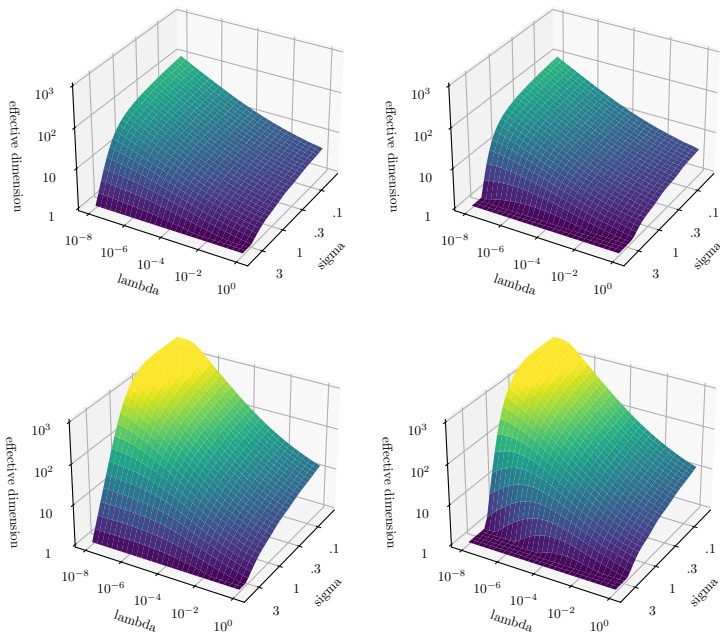

**Figure 8:** Effective dimensions $\mathcal{N}_1$ (left) and $\mathcal{N}_2$ (right) as a function of $(\lambda, \sigma)$ in one dimension (top) and two dimension (bottom) when $\rho_{\mathcal{X}}$ is uniform on $\mathcal{X} = [-1, 1]^d$, and $k$ is the Gaussian kernel.

*Proof.* Once again, the precise study of the variance is easier in $\mathcal{H}$ with the operator $\Sigma$ rather than in $L^2$. First of all, notice that $K = SS^*$ has the same spectrum as $\Sigma = S^*S$, so that, for $a \in [1, 2]$,

$$\mathcal{N}_a(K) = \mathrm{Tr}((K+1)^{-a}K^a) = \mathrm{Tr}((\Sigma+1)^{-a}\Sigma^a) = \sum_{\mu \in \mathrm{spec}(\Sigma)} \frac{\mu^a}{(1+\mu)^a}.$$

This shows the first part of the inequality (27):

$$\left(0 \le \frac{x}{x+1} \le 1 \quad \Rightarrow \quad \frac{x}{x+1} \le \frac{x^a}{(x+1)^a}\right) \quad \Rightarrow \quad \mathcal{N}_2(K) \le \mathcal{N}_1(K).$$

For the second part of the inequality, we need to reformulate $\mathcal{N}_+(K)$, which is the object of the next Lemma. In view of Lemma 6, the last inequality in (27) follows from

$$\mathcal{N}_2(K) = \mathbb{E}_X \left[\mathrm{Tr}((\Sigma+1)^{-1}\varphi(X) \otimes \varphi(X)\right] \le \mathrm{ess\,sup}_X \mathrm{Tr}((\Sigma+1)^{-1}\varphi(X) \otimes \varphi(X)) = \mathcal{N}_+(K).$$

The ends the proof of the proposition. $\qquad\square$

**Lemma 6.** $\mathcal{N}_+(K)$ *can be expressed in $\mathcal{H}$ as*

$$\mathcal{N}_+(K) = \mathrm{ess\,sup}_{x \sim \rho_{\mathcal{X}}} \left\|(\Sigma+1)^{-1/2}\varphi(x)\right\|^2.$$

*Proof.* Observe that, for $x \in \mathcal{X}$,

$$\left\|(\Sigma+1)^{-1/2}\varphi(x)\right\|^2 = \varphi(x)^\top(\Sigma+1)^{-1}\varphi(x) = \mathrm{Tr}\left((\Sigma+1)^{-1}\varphi(x) \otimes \varphi(x)\right).$$

Let us introduce the operator

$$S_x : \mathcal{H} \to L^2(\rho_{\mathcal{X}}), \quad \theta \mapsto (x' \mapsto \varphi(x)^\top\theta).$$

From

$$\langle S_x\theta, g \rangle = \mathbb{E}[g(X)\varphi(x)^\top\theta] = \langle \theta, \mathbb{E}_X[g(X)\varphi(x)]\rangle$$

we get $S_x^* g = \mathbb{E}[g(X)\varphi(x)]$. Similarly, one can check that

$$K_x(g)(x') = (S_x S_x^* g)(x') = (S_x(\mathbb{E}_X[g(X)]\varphi(x)))(x') = \varphi(x')^\top \varphi(x)\mathbb{E}_X[g(X)] = \mathbb{E}[g]k(x,x),$$

and that

$$\Sigma_x \theta = S_x^* S_x \theta = \mathbb{E}_X[\varphi(x)^\top \theta]\varphi(x) = (\varphi(x) \otimes \varphi(x))\theta,$$

from which we deduce that there exists $\varepsilon \in \{-1, 1\}$ such that

$$\varepsilon \left\| (K+1)^{-1}K_x \right\| = \operatorname{Tr}(K+1)^{-1}K_x) = \operatorname{Tr}(\Sigma+1)^{-1}\varphi(x) \otimes \varphi(x)) = \left\| (\Sigma+1)^{-1/2}\varphi(x) \right\|.$$

Necessarily $\varepsilon = 1$ since the right term is positive. Taking the essential supremum ends the proof. $\square$

The following is a reinterpretation of Proposition 29 of Cabannes et al. [5].

**Proposition 7** (Capacity condition under interpolation inequalities)**.** *When $K^p(L^2(\rho_\mathcal{X}))$ is continuously embedded in $L^\infty(\rho_\mathcal{X})$ with $p \leq 1/2$, there exists a constant $c$ such that*

$$\mathcal{N}_+(\lambda^{-1}K) \leq c\lambda^{-2p}. \tag{28}$$

*When the function $x \to k(x,x)$ is bounded, the RKHS associated with $k$ verifies*

$$\mathcal{N}_+(\lambda^{-1}K) = O(\lambda^{-1}).$$

*Proof.* The continuous embedding means that there exists a constant $c$ such that, for any $\lambda \geq 0$,

$$\|K^p f\|_\infty \leq c \|f\|_2.$$

Stated in $\mathcal{H}$, we get

$$\|S\theta\|_\infty \leq c \left\| K^{-p}S\theta \right\|_2 = c \left\| \Sigma^{1/2-p}\theta \right\|$$

for every $\theta \in \mathcal{H}$. In other terms,

$$\operatorname*{ess\,sup}_x \left| \varphi(x)^\top \theta \right| \leq c \left\| \Sigma^{1/2-p}\theta \right\|.$$

Let us denote by $(\lambda_i, \theta_i)$ the eigenvalue decomposition of $\Sigma$. Then

$$\operatorname*{ess\,sup}_x (\varphi(x)^\top \theta)^2 \leq c^2 \left\| \Sigma^{1/2-p}\theta \right\|^2 = c^2 \sum_{i\in\mathbb{N}} \lambda_i^{1-2p}(\theta_i^\top \theta)^2.$$

When considering $\theta = \theta_i$, this leads to

$$\left| \theta^\top \varphi(x) \right| \leq c\lambda_i^{1/2-p}.$$

Therefore,

$$\mathcal{N}_\infty^{1/2}(\lambda^{-1}K) = \sup_x \left\| (\Sigma+\lambda)^{-1/2}\varphi(x) \right\| = \sup_x \sup_{\theta; \|\theta\|\leq 1} \theta^\top \Sigma_\lambda^{-1/2}\varphi(x)$$

$$= \sup_x \sup_{\theta; \|\theta\|\leq 1} \sum_{i\in\mathbb{N}} \frac{\theta^\top \theta_i \theta_i^\top \varphi(x)}{(\lambda+\lambda_i)^{1/2}} \leq c \sup_{a; \sum a_i^2 \leq 1} \sum_{i\in\mathbb{N}} \frac{a_i \lambda_i^{1/2-p}}{(\lambda+\lambda_i)^{1/2}}$$

$$= c \sup_{i\in\mathbb{N}} \frac{\lambda_i^{1/2-p}}{(\lambda+\lambda_i)^{1/2}} = c \sup_{t\in\operatorname{spec}(K)} \frac{t^{1/2-p}}{(\lambda+t)^{1/2}} \leq c \sup_{t\geq 0} \frac{t^{1/2-p}}{(\lambda+t)^{1/2}}$$

$$= c(2p)^{-p}(1-2p)^{1/2-p}\lambda^{-p},$$

where the last equality follows from basic calculus.

The second inequality follows from the fact that $K^{1/2}(L^2) \hookrightarrow L^\infty$ as soon as $\varphi$ is bounded since, for any $f = \varphi(\cdot)^\top \theta \in \mathcal{F} = S\mathcal{H} = K^{1/2}(L^2)$,

$$|f(x)| = \left| \varphi(x)^\top \theta \right| \leq \|\varphi\|_\infty \|\theta\| = \|\varphi\|_\infty \|f\|_\mathcal{F} = \|\varphi\|_\infty \left\| K^{-1/2}f \right\|_2.$$

The previous characterization of $\mathcal{N}_+$ leads to the claim. $\square$

### A.4.2 Bias

We now focus our attention on the bias term.

**Proposition 8** (Source condition). *When $f^* \in K^q(L^2(\rho_\mathcal{X})$ with $q \le 1$, there exists a constant $c$ such that, for any $\lambda \ge 0$,*

$$\mathcal{S}(\lambda^{-1}K) \le c\lambda^{2q}. \tag{29}$$

*Moreover,*

$$\mathcal{S}(\lambda^{-1}K) \le 2 \inf_{q \in [0,1], f \in K^q(L^2)} \lambda^{2q} \left\| K^{-q}f \right\|_2^2 + \left\| f - f^* \right\|_2^2. \tag{30}$$

*Proof.* The proof is straight-forward. If $f^* = K^q g$ with $g \in L^2$, then

$$
\begin{aligned}
\lambda^{-2}\mathcal{S}(\lambda^{-1}K)^{1/2} &= \left\| (K+\lambda)^{-1}f^* \right\|_2 = \left\| (K+\lambda)^{-1}K^q g \right\|_2 \\
&\le \left\| (K+\lambda)^{q-1} \right\|_2 \left\| (K+\lambda)^{-q}K^q \right\|_2 \|g\|_2 \le \lambda^{q-1} \left\| K^{-p}f^* \right\|_2.
\end{aligned}
$$

Squaring this term and multiplying it by $\lambda^2$ leads to the result.

The second equation is due to the decomposition

$$
\begin{aligned}
\lambda^{-2}\mathcal{S}(\lambda^{-1}K) &= \left\| (K+\lambda)^{-1}f^* \right\|_2^2 \le 2 \left\| (K+\lambda)^{-1}f \right\|_2^2 + 2 \left\| (K+\lambda)^{-1}(f^* - f) \right\|_2^2 \\
&\le 2 \left\| (K+\lambda)^{-1}f \right\|_2^2 + 2\lambda^{-2} \left\| f^* - f \right\|_2^2.
\end{aligned}
$$

The previous derivations explain the final result. $\qquad\square$

### A.5 Application of Theorem 4 to Taylor expansions

A natural idea to estimate a target function $f^* \in C^\alpha$, with $C^\alpha$ the space of $\alpha$-Hölder functions, is to estimate its Taylor expansion. This is done with local polynomials which consists in concatenating into $\varphi(x)$ a finite number of features of the form $x \to \mathbf{1}_{x \in A}x^k$ for $k \in [\beta]$ for some $\beta \in \mathbb{N}$ and $A$ in a partition of $\mathcal{X}$; together with the ridgeless estimator of (7). In this setting $\mathcal{N}_2(K, 0)$ is equal to the number of polynomials of degree at most $\beta$ with $d$ variables times the size of the partition considered. The number of such polynomials is $h_\beta(1_d)$, where $h_{\lfloor \beta \rfloor}$ is the complete homogeneous symmetric polynomial of degree $\beta$ in $d$ variables and $1_d$ is the vector of all ones. Thus,

$$\mathcal{N}_2(K, 0) = h_\beta(1_d) = \binom{d+\beta}{\beta} = \frac{(d+\beta)!}{d!\beta!} \in \left(1 + \frac{\beta}{d}\right)^d \cdot \left[1, e^d\right],$$

where the last inequalities is due to the fact that $\binom{n}{k} \in \left[(n/k)^k, (ne/k)^k\right]$.

When $\mathcal{X} = [0,1)$ with uniform distribution, and $f = f^*$ is assumed to be $(\alpha, L_\alpha)$-Hölder, i.e.,

$$\left| f^{(\lfloor \alpha \rfloor)}(x) - f^{(\lfloor \alpha \rfloor)}(y) \right| \le L_\alpha \left| x - y \right|^{\alpha - \lfloor \alpha \rfloor},$$

by fitting Taylor expansions on intervals $[(i-1)/m, i/m)$ for $i \in [m]$ and $m \in \mathbb{Z}_+$ with polynomials

$$\varphi(x) = \left( \left( x - \frac{2i-1}{2m} \right)^j \cdot \mathbf{1}_{x \in [\frac{i-1}{m}, \frac{i}{m})} \right)_{i \in [m], j \in [0, \lfloor \alpha \rfloor]},$$

one can ensure that [see 12, Lemma 11.1]

$$\left\| \Pi_\mathcal{F}f^* - f^* \right\|_2 \le \left\| \Pi_\mathcal{F}f^* - f^* \right\|_\infty \le \frac{L}{2^\alpha \lfloor \alpha \rfloor! \, m^\alpha},$$

where $\Pi_\mathcal{F}$ denotes the orthogonal projection from $L^2$ onto $\mathcal{F}$. The same type of result also holds for $\mathcal{X} = [0,1]^d$ when using $m^d$ multivariate polynomials of degree less then $\lfloor \alpha \rfloor$. Balancing the bias and the variance term, we get an excess risk that behaves as (up to higher order term)

$$\mathbb{E}\left[ \|f_n - f^*\|^2 \right] \lesssim \inf_{m \in \mathbb{Z}_+; \beta \in [\alpha]} \left( \frac{\varepsilon^2 (me)^d (1+\beta/d)^d}{n} + \frac{L_\beta^2}{2^{2\beta}\beta!^2 m^{2\beta}} \right) = \inf_{\beta \in [\alpha]} c_\beta n^{-2\beta/(d+2\beta)},$$

for some constant $c_\beta$ that depends on $\beta$, $d$ and grows with $L_\beta$, $\sigma^2$, the infimum being found for $m \propto (L_\beta^2 n)^{1/(d+2\beta)}$. This argument of the minimizer illustrates how the size of the window should depend on how smooth $f^*$ is expected to be among the functions in $C^\alpha$. Interestingly, it equally shows that the partition size does not deteriorate with the dimension of the input space. Nor will deteriorate the percentage of the total volume contained in each region of the partition. However, the radius of those regions will scale as $r = v^{1/d}$ for $v$ the volume of those regions, meaning that when this volume will shrink to zero, the radius will shrink slower as the dimension grows, which will lead to a slower minimization of the approximation error. Ultimately as $n$ grows, the last upper bound will end up behaving according to the "stationary behavior" in $c_\alpha n^{-2\alpha/(2\alpha+d)}$.

## A.6  Application of Theorem 4 to Fourier expansions

Rather than local properties, one could leverage global smoothness properties, such as fast decays of Fourier coefficients, e.g. $f^* \in H^\alpha$ with $H^\alpha$ the Sobolev space of functions that are $\alpha$-times differentiable with derivatives in $L^2(\rho_\mathcal{X})$.

Fourier coefficient estimators can be built implicitly built from translation-invariant kernels, i.e. $k(x, x') = \lambda^{-1} q((x - x')/\sigma)$ for $q : \mathbb{R}^d \to \mathbb{R}$, $\sigma$ a bandwidth parameter and $\lambda > 0$ a regularization parameter. Examples are provided by the Matérn kernels, defined from $\hat{q}(\omega) = (1 + \|\omega\|^2)^{-\beta}$, the exponential kernel, which corresponds to a Matérn kernel of low smoothness $\beta = (d + 1)/2$, and the Gaussian kernel, which can be seen as the limit of a Matérn kernel to infinite smoothness, defined as $q(x) = \exp(-\|x\|^2)$. The corresponding estimators 8 can be proven to guarantee the convergence rates as Theorem 1, namely

$$\mathbb{E}\left[\|f_n - f^*\|^2\right] \leq \inf_{\beta < \alpha} c_\beta n^{-2\beta/(2\beta+d)},$$

where $c_\beta$ notably relates to the norm of $f^*$ in $H^\beta$.

*Proof.* For the Matérn kernels, the generalization error reads, with $\tau = 2\beta - d$, up to constants and higher order terms, according to Table 1,

$$\mathbb{E}\left[\|f_n - f^*\|^2\right] \lesssim \frac{(\sigma^\tau \lambda)^{-d/2\beta}}{n} + (\sigma^\tau \lambda)^{\alpha/\beta}.$$

This is optimized for

$$\sigma^\tau \lambda = n^{-2\beta/(2\alpha+d)},$$

leading to minimax convergence rates in $O(n^{-2\alpha/(2\alpha+d)})$.

For the Gaussian kernel, we get

$$\mathbb{E}\left[\|f_n - f^*\|^2\right] \lesssim \frac{\sigma^{-d} \log(\lambda^{-1}\sigma^d)^{d/2}}{n} + \sigma^{2\alpha} \log(\lambda^{-1}\sigma^d)^{-\alpha},$$

which is optimized for

$$\sigma^{-2} \log(\lambda^{-1}\sigma^d) = n^{2/(2\alpha+d)},$$

leading to the same minimax convergence rate. In particular, when $\sigma$ is fixed, this leads to

$$\lambda = \lambda_n = \sigma^d \exp(-\sigma^2 n^{2/(2\alpha+d)}).$$

Based on Theorem 4, this is true as long as $\mathcal{N}(\lambda)\mathcal{N}_\infty^{1/2}(\lambda)/n^{1/2}$ goes to zero with $n$, which imposes some constraints on $\alpha$ when assuming $f^* \in H^\alpha$. However, considering a generalization of Mourtada et al. [20] from linear regression to RKHS, the upper bound is actually true without this constraint, which allows to prove the convergence rates in $O(n^{2\alpha/(2\alpha+d)})$ for any $\alpha$. □

## A.7  Application of Theorem 4 to Sobolev spaces

We now turn ourselves to an informal more generic reformulation of the previous facts on Fourier expansions estimation based on well-known facts in approximation theory [30, 10].

**Proposition 9** (Informal source condition). *When $\mathcal{F} = H^\beta$ and $f^* \in H^\alpha$, it holds*
$$f^* \in K^{\alpha/2\beta}(L^2(\rho_\mathcal{X})).$$

*Proof.* In essence, as explained in Appendix B, $K$ takes a function in $L^2(\rho_\mathcal{X})$ and multiply its Fourier transform by $\hat{q}(\omega)^{-1} = (1 + \|\omega\|^2)^\beta$, with $q$ defining the Matérn kernel, making it $2\beta$-smooth in the Sobolev sense. In harmonic settings where the Fourier functions diagonalize $K$ and $\hat{q}(\omega)$ parameterizes the spectrum of $K$, the fractional operator $K^p$ can be seen as multiplying the Fourier transform of $f$ by $\hat{q}(\omega)^{-p}$, making it $2p\beta$-smooth. This fact can be extended beyond those harmonic settings, notably with interpolation inequalities. On the opposite direction, any $\alpha$-smooth function can be multiplied by $q(\omega)^{\alpha/2\beta}$ in Fourier while staying in $L^2(\rho_\mathcal{X})$, so that, if $f^*$ is $\alpha$-smooth, it belongs to $K^{\alpha/2\beta}$. $\qquad\square$

**Proposition 10** (Informal interpolation inequality). *When $\mathcal{F} = H^\beta$ is the space of $\alpha$-Sobolev functions,*
$$K^{d/2\beta}(L^2) \hookrightarrow L^\infty.$$

*Proof.* Note that $\mathcal{F} = S\mathcal{H} = K^{1/2}(L^2)$. We have seen informally in the proof of the previous lemma how $K^p(L^2) \subset H^{2p\beta}$. Now, let us recall the Sobolev embedding theorems [1]. Under mild assumptions on $\rho_\mathcal{X}$, for $k, r, l, s > 0$
$$W^{k,r}(\rho_\mathcal{X}) \hookrightarrow W^{l,s}(\rho_\mathcal{X}), \qquad \text{as long as} \qquad \frac{1}{r} - \frac{k}{d} \leq \frac{1}{s} - \frac{l}{d}.$$
We want to use it with $k = 2p\beta$, $r = 2$, $l = 0$ and $s = +\infty$, which leads to $p = 4\beta/d$. $\qquad\square$

These results could be used to explain the scaling with respect to $\lambda$ in Table 1, which we will derive formally in Appendix B. They can also be used to show convergence rates in $O(n^{-2\alpha/(2\alpha+d)})$ for any target functions $f^* \in H^\alpha$ when utilizing a kernel such that $\mathcal{F} = H^\beta$ (in terms of sets equally, $\mathcal{F}$ being eventually endowed with an other norm than Sobolev norms).

## B  Translation-invariant kernels and Fourier analysis

This section recalls basic facts about kernel methods and Fourier analysis, before proving Table 1.

### B.1  Stylized analysis on the torus

When $k$ is a translation-invariant kernel, i.e. $k(x, x') = q(x - x')$, the integral operator $K$ is a convolution against $q$. Let us expand on the friendly case provided by the torus $\mathcal{X} = \mathbb{T}^d := \mathbb{R}^d/\mathbb{Z}^d = [0,1]^d/\sim$, where $\sim$ is the relation identifying opposite faces of the hypercube, and $\rho_\mathcal{X}$ the uniform distribution. On the torus, a translation invariant kernel is defined through $q$ being a one-periodic function on $\mathbb{R}^d$. Let $\mathrm{d}x$ denote the Lebesgue measure on $\mathbb{R}^d$. The integral operator $K : L^2(\mathcal{X}, \mathrm{d}x) \to L^2(\mathcal{X}, \mathrm{d}x)$ is the convolution
$$Kf(x) = \int_{[0,1]^d} k(x, x')f(x')\,\mathrm{d}x' = \int_{[0,1]^d} q(x' - x)f(x')\,\mathrm{d}x' = q * f(x).$$
For $m \in \mathbb{Z}^d$, define the Fourier function $f_m : x \mapsto \exp(2i\pi \langle m, x\rangle)$. One can check that the $f_m$'s form an orthonormal family that diagonalizes $K$ with[6]
$$Kf_m = \hat{q}_m f_m, \qquad \text{where} \qquad \hat{q}_m = \int_{[0,1]^d} q(x)\exp(2i\pi\langle x, m\rangle)\,\mathrm{d}x.$$
Hence, using Pythagoras theorem, we can define the norm on $\mathcal{F}$ through its action on Fourier coefficients as
$$\|f\|_\mathcal{F}^2 = \left\langle f, K^{-1}f\right\rangle_{L^2(\rho_\mathcal{X})} = \sum_{m \in \mathbb{Z}^d} \widehat{q_m^{-1}}|\hat{f}_m|^2 = \int_{\mathbb{R}^d} \hat{q}(\omega)^{-1}|\hat{f}(\omega)|^2 \#(\mathrm{d}\omega),$$
where $\hat{f}_m = \langle f, f_m\rangle_{L^2(\rho_\mathcal{X})}$, and $\#$ is the counting measure on $\mathbb{Z}^d \subset \mathbb{R}^d$.

---

[6]Indeed, the Fourier transform of a function $f \in L^2(\mathbb{T}^d)$ can be defined as the mapping from $\mathbb{N}$ to $(\langle f_i, f\rangle)_{i \in \mathbb{N}}$ where $(f_i)$ is a basis that diagonalizes all convolution operators (note that this definition is possible because $\rho_\mathcal{X}$ is uniform on the torus).

### B.1.1 First part of the proof of Proposition 1

Since $K$ is diagonalized in Fourier, we compute the size of $\mathcal{F}$ for $a \in \{1, 2\}$ with

$$\mathcal{N}_a(K) = \mathrm{Tr}\left(K^a(K+1)^{-a}\right) = \sum_{m \in \mathbb{Z}^d} \frac{\widehat{q}_m^a}{(\widehat{q}_m + 1)^a}.$$

For kernels whose scales are explicitly defined through $q_\sigma = q(x/\sigma)$, we have $\widehat{q}_\sigma(\omega) = \sigma^d \widehat{q}(\sigma\omega)$, which leads to (14).

Similarly, the bound on the bias term follows from Fourier analysis by

$$\mathcal{S}(K) = \left\|K(K+1)^{-1}f\right\|_{L^2(\rho_\mathcal{X})}^2 = \sum_{m \in \mathbb{Z}^d} \left|\widehat{f}_m\right| \left(\frac{1}{\widehat{q}_m + 1}\right)^2,$$

which provides (15).

### B.1.2 Second part of the proof of Proposition 1

When $\rho$ is a distribution that is absolutely continuous with respect to the Lebesgue measure and whose density is bounded from above, we get

$$K \preceq \rho_\infty K_{\mathrm{d}x}, \qquad \text{with} \qquad \rho_\infty = \left\|\frac{\mathrm{d}\rho_\mathcal{X}}{\mathrm{d}x}\right\|_{L^\infty(\rho_\mathcal{X})},$$

where $K_{\mathrm{d}x}$ is the integral operator associated to the kernel $k$ on $L^2(\mathrm{d}x)$. Using the fact the effective dimension is an increasing function of the eigenvalues (since $x \mapsto x/(x+1)$ is increasing), and that eigenvalues are increasing with the Loewner order, this leads to

$$\mathcal{N}(K) \leq \rho_\infty \,\mathrm{Tr}\left((K_{\mathrm{d}x}+1)^{-1}K_{\mathrm{d}x}\right) = \rho_\infty \int_{\mathbb{R}^d} \frac{\widehat{q}(\omega)}{1 + \widehat{q}(\omega)}\,\mathrm{d}\omega.$$

Note that those derivations are written informally (since $K$ and $K_{\mathrm{d}x}$ do not act on the same space), but could be made formal with the isomorphic covariance operators on $\mathcal{H}$, plus some technicalities to make sure $\Sigma_{\mathrm{d}x}$ is well defined (assuming $\varphi(X)$ has a fourth-order moment against Lebesgue, or approaching $K_{\mathrm{d}x}$ within its action on compact subspaces of $\mathcal{X}$ where it is bounded, before taking the limit of $\mathcal{N}_{\mathrm{d}x}(K)$).

For the bias term, using the fact that $L^2(\rho_\mathcal{X})$ is continuously embedded in $L^2(\mathrm{d}x)$ and the isometry between the spatial and the Fourier domain, we get

$$\|f - f^*\|_{L^2(\rho_\mathcal{X})} \leq \rho_\infty^{1/2} \|f - f^*\|_{L^2(\mathrm{d}x)} = \rho_\infty^{1/2} \|\widehat{f} - \widehat{f^*}\|_{L^2(\mathrm{d}x)}.$$

Finally, it should be noted that the norm associated with $\mathcal{F}$ does not depend on the density of $X$, hence the formula can be written independently of $\rho_\mathcal{X}$. Indeed, under definition assumption, i.e. if $q \in L^1(\mathrm{d}x)$, this formula can even be written with a measure of infinite mass. For example, when $\mathcal{X} = \mathbb{R}^d$, one can consider the Fourier transform associated with $L^2(\mathrm{d}x)$, and get

$$\|f\|_\mathcal{F}^2 = \int_{\mathbb{R}^d} \widehat{q}(\omega)^{-1}|\widehat{f}(\omega)|^2\,\mathrm{d}\omega, \qquad \text{where} \qquad \widehat{q}(\omega) = \int_{\mathbb{R}^d} q(x)\exp(-2i\pi\langle x, \omega\rangle)\,\mathrm{d}x, \quad (31)$$

although some care is needed to deal with the continuous version of the spectral theorem (the set of eigenvalues being non-countable). From there the same derivations as for Proposition 1 lead to the desired result.

### B.2 Sobolev spaces

Recall the action of differentiation on the Fourier transform: for $m \in \mathbb{N}^d$, $|m| := \|m\|_1$, and $f \in L^2(\mathrm{d}x)$,

$$\frac{\widehat{\partial^{|m|}f}}{\prod_{i \in [d]} \partial^{m_i} x_i}(\omega) = (2i\pi)^{|m|} \prod_{i \in [d]} \omega_i^{m_i} \widehat{f}(\omega).$$

This characterizes the pseudo-norm

$$\|f\|_m^2 = \int_{\mathbb{R}^d} \left\| \frac{\partial^{|m|} f(x)}{\prod_{i \in [d]} \partial^{m_i} x_i} \right\|^2 \mathrm{d}x = (2\pi)^{2|m|} \int_{\mathbb{R}^d} \prod_{i \in [d]} \omega_i^{2m_i} \left| \widehat{f}(\omega) \right|^2 \mathrm{d}\omega.$$

This pseudo-norm is associated with the translation-invariant kernel such that $\widehat{q}(\omega) = \prod_{i \in [d]} \omega_i^{-2m_i}$ as per (31). Note that $q$ is well defined when $\widehat{q}$ belongs to $L^1(\mathrm{d}x)$ (by Bochner's theorem), that is $|m| > d$. Those observations are usual to deduce that the Matérn kernels, which are defined from $\widehat{q}(\omega) \propto (1 + \|\omega\|_2^2)^{-\beta}$, correspond to the Sobolev spaces $H^\beta(\mathrm{d}x)$ endowed with the norm

$$\|f\|_{H^\beta}^2 = \sum_{m; |m| \leq \beta} \|f\|_m^2.$$

It follows from Bochner's theorem that $H^\beta$ is a reproducing kernel Hilbert space if and only if $2\beta > d$. Remarkably, the exponential kernel corresponds to the Matérn kernel with $\beta = (d+1)/2$ [24]. For the Gaussian kernel, $\widehat{q}(\omega) = \pi^{-d/2} \exp(-\pi^2 \|\omega\|^2)$, and the associated function class $\mathcal{F}$ is analytic (by the Paley-Wiener theorem).

### B.2.1  Functional sizes

Let us now express the capacity and bias bound within Sobolev spaces.

**Proposition 11** (Sobolev capacity). *When $\widehat{q}(\omega) = \lambda^{-1}(1 + \|\omega\|^2)^{-\beta}$ for $\beta > d$, $\lambda\sigma^{-d}$ is bounded and $\rho$ has a bounded density, we have*

$$\mathcal{N}_1(\sigma, \lambda) \leq \frac{2\beta \rho_\infty \pi^{(d+1)/2}}{\Gamma((d-1)/2)} \lambda^{-d/2\beta} \sigma^{-d(2\beta-d)/2\beta}.$$

*Moreover, when $\mathcal{X} = \mathbb{T}^d$ and $\rho_{\mathcal{X}}$ is uniform, we get*

$$\mathcal{N}_2(\sigma = 1, \lambda) \geq \max_{l \in [d]} \frac{l\pi^{(l+1)/2}}{2^{l+1}\Gamma((l-1)/2)} \lambda^{-l/2\beta}.$$

*Proof.* In this setting, Proposition 1 leads to

$$\int_{\mathbb{R}^d} \frac{1}{1 + \lambda\widehat{q_\sigma}(\omega)^{-1}} \mathrm{d}\omega = \int_{\mathbb{R}^d} \frac{1}{1 + \lambda\sigma^{-d}\widehat{q}(\sigma\omega)^{-1}} \mathrm{d}\omega = \int_{\mathbb{R}^d} \frac{1}{1 + \lambda\sigma^{-d}(1 + \sigma^2\|\omega\|^2)^\beta} \mathrm{d}\omega$$

$$= \mathrm{surf}(\mathcal{S}^{d+1}) \int_{\mathbb{R}_+} \frac{r^{d-1}\,\mathrm{d}r}{1 + \lambda\sigma^{-d}(1 + \sigma^2 r^2)^\beta}$$

$$= 2\pi\,\mathrm{vol}(\mathcal{S}^d) \int_{\lambda^{1/\beta}\sigma^{-d/\beta}}^\infty \frac{(u - \lambda^{1/\beta}\sigma^{-d/\beta})^{d/2-1}\,\mathrm{d}u}{\lambda^{d/2\beta}\sigma^{d-d^2/2\beta}(1 + u^\beta)}$$

$$= 2\pi\,\mathrm{vol}(\mathcal{S}^d)\lambda^{-d/2\beta}\sigma^{d(d-2\beta)/2\beta} \int_{\mathbb{R}_+} \frac{x^{d/2-1}\,\mathrm{d}x}{1 + (x + \lambda^{1/\beta}\sigma^{-d/\beta})^\beta}$$

$$\leq 2\pi\beta\,\mathrm{vol}(\mathcal{S}^d)\lambda^{-d/2\beta}\sigma^{d(d-2\beta)/2\beta},$$

where we used the fact that

$$\int_0^\infty \frac{x^{d/2-1}\,\mathrm{d}x}{1 + (x + \lambda^{1/\beta}\sigma^{-d/\beta})^\beta} \leq \int_0^\infty \frac{x^{d/2-1}\,\mathrm{d}x}{\max(1, \max(x^\beta, \lambda\sigma^{-d}))}$$

$$\leq \int_0^1 x^{d/2-1}\,\mathrm{d}x + \int_1^\infty \frac{x^{d/2-1}\,\mathrm{d}x}{x^\beta} = d/2 - (d/2 - \beta) = \beta,$$

which is true as long as $\beta > d/2$ to ensure proper convergence of the last integral.

For the part on the torus, in order to get a sharp learning limit, we need to be slightly more precise. In particular, we want to relate the discrete Fourier transform integral of Proposition 1 with the continuous one through series-integral comparison, and get a lower bound on the last integral. We

will fix $\sigma = 1$ for simplicity. A simple cut of $\mathbb{R}^d$ into unit cubes, together with the fact that our integrand is decreasing, leads to

$$\sum_{m \in \mathcal{Z}^d} \mathbf{1}_{0 \notin m} \frac{\widehat{q}_m^2}{(\widehat{q}_m + \lambda)^2} \leq \int \frac{\widehat{q}(\omega)^2}{(\widehat{q}(\omega) + \lambda)^2} \, d\omega \leq \sum_{m \in \mathcal{Z}^d} 2^{\#\{i \in [d] \,|\, m_i = 0\}} \frac{\widehat{q}_m^2}{(\widehat{q}_m + \lambda)^2}.$$

We simplify it as

$$\mathcal{N}_2(\sigma, \lambda) \geq 2^{-d} \int \frac{\widehat{q}(\omega)^2}{(\widehat{q}(\omega) + \lambda)^2} \, d\omega.$$

We now compute the integral with the same techniques as before:

$$\int \frac{\widehat{q}(\omega)^2}{(\widehat{q}(\omega) + \lambda)^2} \, d\omega = \int \frac{1}{(1 + \widehat{q}(\omega)^{-1}\lambda)^2} \, d\omega = \int \frac{1}{(1 + (1 + \|\omega\|^2)^\beta \lambda)^2} \, d\omega$$

$$= 2\pi \, \mathrm{vol}(\mathcal{S}^d) \int \frac{x^{d-1}}{(1 + (1 + x^2)^\beta \lambda)^2} \, dx$$

$$= 2\pi \, \mathrm{vol}(\mathcal{S}^d) \lambda^{-d/2\beta} \int \frac{x^{d-1}}{(1 + (\lambda^{1/\beta} + x^2)^\beta)^2} \, dx$$

$$\geq 2\pi \, \mathrm{vol}(\mathcal{S}^d) \lambda^{-d/2\beta} \int \frac{x^{d-1}}{4 \max(1, 4^\beta \max(\lambda^2, x^{4\beta}))} \, dx$$

$$= 2^{-1} \pi \, \mathrm{vol}(\mathcal{S}^d) \lambda^{-d/2\beta} \left( \int_0^1 x^{d-1} \, dx + 4^{-\beta} \int_1^\infty \frac{x^{d-1}}{x^{4\beta}} \, dx \right)$$

$$= 2^{-1} \pi \, \mathrm{vol}(\mathcal{S}^d) \lambda^{-d/2\beta} (d + 4^{-\beta}(4\beta - d))$$

$$\geq 2^{-1} \pi \, \mathrm{vol}(\mathcal{S}^d) \lambda^{-d/2\beta} d.$$

It should be noted that this last bound is somewhat too lax, as it tends to zero when the dimension increases. Since $\sum_{m \in \mathbb{Z}^d} a(\|m\|)$ for $a > 0$ is strictly increasing with $d$, we deduce that this lower bound holds for any $k \leq d$. $\qquad\square$

**Proposition 12** (Gaussian capacity)**.** *When $\widehat{q}(\omega) = \lambda^{-1} \exp(-\|\omega\|^2)$ and $\rho$ has a bounded density, we have*

$$\mathcal{N}_1(\lambda, \sigma) \leq \frac{\rho_\infty \pi^{(d-1)/2} d}{2\sigma^d} L(\lambda^{-1}\sigma^d),$$

*where $L$ is defined by Eq. (32). In particular, $L(x) \leq x$ when $x < 1$, and $L(x) \lesssim \log(x)^{d/2}$ when $x$ gets large. Moreover when $\mathcal{X} = \mathbb{T}^d$ and $\rho_{\mathcal{X}}$ is uniform, we get*

$$\mathcal{N}_2(\lambda, \sigma) \geq \frac{\pi^{(d-1)/2}}{2^{d+1}\sigma^d} L(\lambda^{-1}\sigma^d).$$

*Proof.* With the Gaussian kernel, Proposition 1 leads to

$$\int_{\mathbb{R}^d} \frac{1}{1 + \lambda\sigma^{-d} \exp(\sigma^2 \|\omega\|^2)} \, d\omega = \mathrm{vol}(\mathbb{S}^d) \int_{\mathbb{R}_+} \frac{2x^{d-1}}{1 + \lambda\sigma^{-d} \exp(\sigma^2 x^2)} \, dx$$

$$= \mathrm{vol}(\mathbb{S}^d)\sigma^{-d} \int_{\mathbb{R}_+} \frac{u^{d/2-1}}{1 + \lambda\sigma^{-d} \exp(u)} \, du$$

$$= \mathrm{vol}(\mathbb{S}^d)\Gamma(d/2) \frac{-\mathrm{Li}_{d/2}\left(-\sigma^d/\lambda\right)}{\sigma^d},$$

where $\mathrm{Li}_{d/2}$ is the polylogarithm function, hence the definition of $L$ as

$$L(x) = -\mathrm{Li}_{d/2}(-x) = \sum_{k=1}^\infty \frac{(-1)^{k+1}x^k}{k^{d/2}}. \tag{32}$$

We recognize an alternating sequence, whose term amplitudes are decreasing as a function of $k \in \mathbb{N}$ when $x \leq 1$, which explains that $L(x)$ is smaller than the first term in this case. The expansion of the polylogarithm function at infinity leads to the upper bound when $x$ goes to infinity.

When it comes to a lower bound, we can proceed as the precedent lemma with

$$\mathcal{N}_2(\sigma, \lambda) \geq 2^{-d} \int \frac{\mathrm{d}\omega}{(1 + \lambda \widehat{q}(\omega)^{-1})^2} \geq 2^{-d} \int \frac{\mathrm{d}\omega}{1 + \lambda \widehat{q}(\omega)^{-1}},$$

which corresponds to the integral computed for the upper bound. Once again, this also holds when $d$ is replaced by any $l \in [d]$. $\square$

### B.2.2 Adherence

Let us now turn our attention to the bias, i.e. the adherence of functions in those spaces. For proof readability, we will assume that $\rho_{\mathcal{X}}$ has compact support. In this setting, $W^{\alpha,2}$ is continuously embedded in $C^{\alpha+d/2} = W^{\alpha+d/2,\infty}$, which can be defined as

$$C^{\alpha+d/2} = \left\{ f : \mathbb{R}^d \to R \ \middle| \ \|f\|_\alpha := \operatorname*{ess\,sup}_{\omega \in \mathbb{R}^d} \left| \widehat{f}(\omega) \right| (1 + \|\omega\|)^{\alpha+d/2} < +\infty \right\}.$$

**Proposition 13** (Adherence of $H^\alpha$ in $H^\beta$). *When $\widehat{q}(\omega) = \lambda^{-1}(1 + \|\omega\|^2)^{-\beta}$, hence $\mathcal{F} = H^\beta$, if $\alpha > 2\beta$, for any $f^* \in H^\alpha(\rho_{\mathcal{X}})$, and $\lambda$ small enough, we have*

$$\mathcal{B}(\sigma, \lambda) \leq \lambda^2 \left\| K^{-1} f \right\|_{L^2(\rho_{\mathcal{X}})}^2.$$

*If $\alpha < \beta$ and $\rho_{\mathcal{X}}$ has a bounded density, then for any function $f^* \in C^{\alpha+d/2}$*

$$\mathcal{B}(\sigma, \lambda) \leq \rho_\infty \frac{4\beta \pi^{(d+1)/2} \rho_\infty \|f\|_\alpha^2 \beta}{(\beta^2 - \alpha^2)\Gamma((d-1)/2)} \lambda^{\alpha/\beta} \sigma^{(2\beta-d)\alpha/\beta}. \tag{33}$$

*Proof.* The first part results from previous considerations on the source condition since we have the inclusion $f^* \in H^\alpha \subset H^{2\beta} = K(L^2(\rho_{\mathcal{X}}))$. The second part follows from an $L^1 - L^\infty$ Hölder inequality:

$$\mathcal{B}(\sigma, \lambda) \leq \rho_\infty \int_{\mathbb{R}^d} \frac{\left| \widehat{f}(\omega) \right|^2}{(\lambda^{-1}\sigma^d \widehat{q}(\sigma\omega) + 1)^2} \, \mathrm{d}\omega \leq \rho_\infty \|f\|_\alpha^2 \int_{\mathbb{R}^d} \frac{(1 + \|\omega\|^2)^{-(d/2+\alpha)}}{(\lambda^{-1}\sigma^d(1 + \sigma^2\|\omega\|^2)^{-\beta} + 1)^2} \, \mathrm{d}\omega$$

$$= 2\pi \operatorname{vol}(\mathbb{S}^d) \rho_\infty \|f\|_\alpha^2 \int_{\mathbb{R}_+} \frac{2(1 + x^2)^{-(d/2+\alpha)} x^{d-1}}{(\lambda^{-1}\sigma^d(1 + \sigma^2 x^2)^{-\beta} + 1)^2} \, \mathrm{d}x$$

$$= 2\pi \operatorname{vol}(\mathbb{S}^d) \rho_\infty \|f\|_\alpha^2 a^\alpha \sigma^{2\alpha} \int_a^\infty \frac{(u - a + a\sigma^2)^{-(d/2+\alpha)}(u - a)^{d/2-1}}{(u^{-\beta} + 1)^2} \, \mathrm{d}u$$

$$\leq \frac{4\beta\pi \operatorname{vol}(\mathbb{S}^d) \rho_\infty \|f\|_\alpha^2 \beta}{\beta^2 - \alpha^2} \lambda^{\alpha/\beta} \sigma^{(2\beta-d)\alpha/\beta},$$

where $a$ was set to $\lambda^{1/\beta}\sigma^{-d/\beta}$, and the last integral can be bounded by

$$\int_a^\infty \frac{(u - a + a\sigma^2)^{-(d/2+\alpha)}(u - a)^{d/2-1}}{(u^{-\beta} + 1)^2} \, \mathrm{d}u \leq \int_0^\infty \frac{u^{d/2+\beta-1}}{(u + a(\sigma^2 - 1))^{d/2+\alpha}(1 + u^\beta)^2} \, \mathrm{d}u$$

$$\leq \int_0^\infty \frac{u^{d/2+\beta-1}}{(u + a(\sigma^2 - 1))^{d/2+\alpha}(1 + u^\beta)^2} \, \mathrm{d}u \leq \int_0^1 \frac{u^{d/2+\beta-1}}{u^{d/2+\alpha}} \, \mathrm{d}u + \int_1^{+\infty} \frac{u^{d/2+\beta-1}}{u^{d/2+\alpha}u^{2\beta}} \, \mathrm{d}u$$

$$= \frac{1}{\beta - \alpha} + \frac{1}{\beta + \alpha} = \frac{\beta}{\alpha(\beta - \alpha)}.$$

Recalling the volume of the sphere completes the proof. $\square$

**Proposition 14** (Adherence of $H^\alpha$ in the Gaussian RKHS). *When $\mathcal{F}$ is associated with the Gaussian kernel and $\rho_{\mathcal{X}}$ has a bounded density, for any $f^* \in C^{\alpha+d/2}$ we have*

$$\mathcal{B}(\sigma, \lambda) \leq \frac{2\pi^{(d+1)/2} \rho_\infty \|f\|_\alpha^2}{\Gamma((d-1)/2)} \left( \frac{1}{\sigma^{2d+4\alpha}} + \frac{2\log(2)^{-(d/2+2\alpha)}}{d + 2\alpha} \right) \sigma^{2\alpha} \log(\lambda^{-1}\sigma^d)^{-\alpha}.$$

*Proof.* We follow the same path as for the adherence of $H^\alpha$ in $H^\beta$:

$$\mathcal{B}(\sigma, \lambda) \leq \rho_\infty \|f\|_\alpha^2 \int_{\mathbb{R}^d} \frac{(1 + \|\omega\|^2)^{-(d/2+\alpha)}}{(\lambda^{-1}\sigma^d \widehat{q}(\sigma\omega) + 1)^2} \, d\omega$$

$$= \rho_\infty \|f\|_\alpha^2 \int_{\mathbb{R}^d} \frac{(1 + \|\omega\|^2)^{-(d/2+\alpha)}}{(\lambda^{-1}\sigma^d \exp(-\sigma^2 \|\omega\|^2) + 1)^2} \, d\omega$$

$$= 2\pi \operatorname{vol}(\mathbb{S}^d)\rho_\infty \|f\|_\alpha^2 \int_{\mathbb{R}_+} \frac{(1 + x^2)^{-(d/2+\alpha)} x^{d-1}}{(\lambda^{-1}\sigma^d \exp(-\sigma^2 x^2) + 1)^2} \, dx$$

$$= 2\pi \operatorname{vol}(\mathbb{S}^d)\rho_\infty \|f\|_\alpha^2 \sigma^{2\alpha} \int_1^{+\infty} \frac{\log(x)^{d/2-1}}{(\sigma^2 + \log(x))^{d+2\alpha}(a + x)} \, dx$$

$$\leq 2\pi \operatorname{vol}(\mathbb{S}^d)\rho_\infty \|f\|_\alpha^2 \sigma^{2\alpha} \log(\lambda^{-1}\sigma^d)^{-\alpha} \left( \frac{1}{\sigma^{2d+4\alpha}} + \frac{\log(2)^{-\alpha}}{\alpha} \right),$$

where the integral can be bounded by

$$\int_1^{+\infty} \frac{\log(x)^{d/2-1}}{(\sigma^2 + \log(x))^{d+2\alpha}(a + x)} \, dx \leq \int_1^e \frac{\log(x)}{\sigma^{2d+4\alpha}} \, dx + \int_e^{+\infty} \frac{1}{(\log(x))^{d/2+2\alpha+1}x} \, dx$$

$$= \frac{1}{\sigma^{2d+4\alpha}} + \frac{\log(2)^{-(d/2+2\alpha)}}{d/2 + 2\alpha}.$$

The same type of derivations can be made for the bias in the lower bound. $\square$

Note that the proofs also work when the $L^1 - L^\infty$ Hölder inequality is replaced with $L^\infty - L^1$, showcasing the norm of $f$ in $H^\alpha$ instead of in $C^{\alpha+d/2}$. We refer to Bach [4] for details.

**Remark 15** (Blessing of dimensionality). *It should be noted that all our integral calculations show a constant $2\pi\rho_\infty \operatorname{vol}(\mathbb{S}^d)$ which will be present in front of the excess risk. As $d$ increases, this constant goes to zero faster than any exponential profile. To see that, note how the volume of the $d$-sphere is always smaller than twice the one of its inscribed hypercube, whose volume is $d^{-d/2}$. We do not have clear intuition to understand this behavior at the time of writing.*

The previous derivations could be used to derive lower bound under highly specific priors on the structure of $f^*$. However, this proposition is somewhat deceptive compared to Theorems 2 and 3 that are more generic, easier to parse, and show more clearly the curse of dimensionality.

**Proposition 16** (Lower-bound application example). *For the Matérn kernel corresponding to $\mathcal{F} = H^\beta$, on the torus $\mathbb{T}^d = \mathbb{R}^d/\mathbb{Z}^d$ with uniform measure $\rho_{\mathcal{X}}$, when $f^* : x \mapsto \cos(2\pi m^\top x)$ is a function with a single frequency $m \in \mathbb{Z}^d$,*

$$\inf_{\lambda > 0} \frac{\varepsilon^2 \mathcal{N}_2(\lambda^{-1} K)}{n} + \mathcal{S}(\lambda^{-1} K) \geq \max_{l \in [d]} \left( \frac{l\pi^{(l-1)/2}\varepsilon^2}{2^{l-1}\Gamma((l-1)/2)n} \right)^\gamma \left( 1 + \|m\|^2 \right)^{\gamma l/2},$$

*where $\gamma = 4\beta/(4\beta + l)$ goes to one as $\beta$ goes to infinity.*

*Proof.* Using the bias and variance lower bounds decomposition and Proposition 11, we get, when $f^* = f_m$ is a single frequency function $\hat{f}(\omega) = \delta_m(\omega)$ with $m \in \mathbb{Z}^d$,

$$\frac{\varepsilon^2 \mathcal{N}_2(\lambda^{-1} K)}{n} + \mathcal{S}(\lambda^{-1} K) \geq \left( \frac{d\pi^{(d-1)/2}}{2^{d+2}\Gamma((d-1)/2)n}\lambda^{-d/2\beta} + \frac{\lambda^2}{((1 + \|m\|^2)^{-\beta} + \lambda)^2} \right)$$

$$\geq \frac{1}{2} \left( \frac{d\pi^{(d-1)/2}}{2^{d+1}\Gamma((d-1)/2)n}\lambda^{-d/2\beta} + \min(\lambda^2(1 + \|m\|^2)^{2\beta}, 1) \right).$$

From the fact that

$$\inf_{x \in \mathbb{R}} ax^{-\alpha} + bx^2 = \left( \alpha^{2/(2+\alpha)} + \alpha^{-\alpha/(2+\alpha)} \right) b^{\alpha/(2+\alpha)}a^{2/(2+\alpha)} \geq b^{\alpha/(2+\alpha)}a^{2/(2+\alpha)},$$

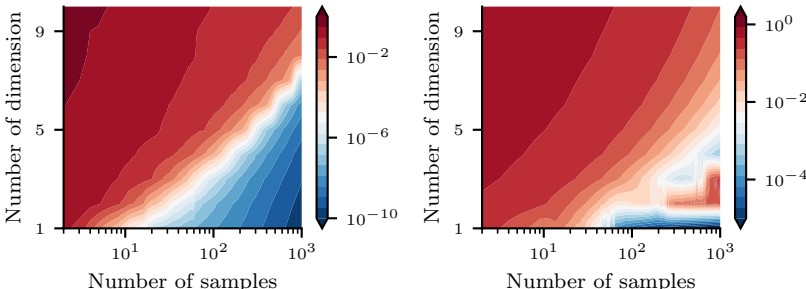

**Figure 9:** (Right) Noise-free convergence rates for $f^*(x) = x_1^5$ with $k(x,y) \propto (1 + x^\top y)^5$. We observed similar deterioration of convergence rates as a function of the dimension $d$ as on Figure 2. The fact that the error is not exactly zero when $d = 1$ and $n \geq 5$ is due to a small regularization added in our algorithm to avoid running into computational issues when inverting a matrix online. (Left) Convergence rates for $f^*(x) = \cos(4\pi x_1)$ on the torus $\mathcal{X} = \mathbb{T}^d$ with the (periodic) exponential kernel $k(x,y) = q(-100 \|x - y\|^2 /d)$. We observe similar behavior as on Figure 2, the picture being worse because the kernel weights all frequencies in the Fourier domain, and not only the first $\binom{d+5}{5}$ ones.

we have that, whatsoever $\lambda$ is (if it is fixed for all $n$ independently of the realization $\mathcal{D}_n$).

$$\frac{d\pi^{(d-1)/2}}{2^{d+1}\Gamma((d-1)/2)n}\lambda^{-d/2\beta} + \lambda^2(1 + \|m\|^2)^{2\beta}$$

$$\geq \left(\frac{d\pi^{(d-1)/2}}{n2^{d-1}\Gamma((d-1)/2)}\right)^{4\beta/(4\beta+d)} \left(1 + \|m\|^2\right)^{2\beta d/(4\beta+d)} .$$

Once again, this lower bound is also true when $d$ is replaced by any $l \in [d]$. $\qquad\square$

## C   Experimental details

### C.1   Online solving of problems of increasing size

In order to solve a big number of least-squares problems with increasing numbers of samples, one can use recursive matrix inversion. When $A \in \mathbb{R}^{n \times n}$, $x \in \mathbb{R}^n$ and $b \in \mathbb{R}$, one can check that

$$\begin{pmatrix} A & x \\ x^\top & b \end{pmatrix}^{-1} = \begin{pmatrix} A^{-1} + cyy^\top & -cy \\ -cy^\top & c \end{pmatrix},$$

where

$$c = \frac{1}{b - x^\top y}, \qquad \text{and} \qquad y = A^{-1}x.$$

This allows efficient computation of matrix inversion online as the number of samples increases.

### C.2   Example of convergence rates for "atomic" functions

**Polynomial estimation.**   Consider the target function $f^*(x) = 63x_1^5 - 70x_1^3 + 15x_1$, learned with the polynomial kernel $k(x,y) = (1 + x^\top y)^p$ with $p = 5$ on $\mathcal{X} = [0,1]^d$ and $\rho_{\mathcal{X}}$ uniform, in the interpolation regime $\lambda = 0$. In this setting, $\mathcal{F}$ is exactly the space of polynomials of degree no larger than 5, which follows from the fact that $k$ can be rewritten through a vector $\varphi$ that enumerates monomials:

$$(1 + x^\top y)^p = \sum_{i=0}^p \binom{p}{i} \sum_{(i_j)_j : \sum_j i_j = i} \binom{i}{(i_j)_j} \prod x_j^{i_j} y_j^{i_j} = \varphi(x)^\top \varphi(y).$$

As a consequence, $f^*$ belongs to $\mathcal{F}$, and we expect the bias term to be zero. Yet, we expect the variance, hence the generalization error, to behave as $\varepsilon^2 \dim \mathcal{F}/n$, where $\varepsilon$ corresponds to some notion of variability between labels. The dimension of the class of polynomials in dimension $d$ of degree no larger than $p$ is $\binom{p+d}{d}$. In particular, in dimension $d = 100$, a polynomial of degree at

most $p = 5$ can have up to one hundred million coefficients, so that one needs about one hundred million observations to enter the high-sample regime and expect an excess risk $\mathcal{E}(f_n)$ of order $\varepsilon^2$ as per Theorem 4. While one could fit a polynomial of lower maximum degree, e.g. 4 instead of 5, since the $f^*$ considered here is actually orthogonal to all polynomials of lower degree, there is no hope to obtain better rates. On Figure 2, the target function is $f^*(x) = x_1^5$, and the polynomial kernel is normalized as

$$k(x, y) = \left( \frac{1 + x^\top y}{1 + d} \right)^5$$

to avoid computational issues. The noise level $\varepsilon$ is set to $10^{-2}$, and the lower bound in $\varepsilon^2 \dim \mathcal{F}/n$ is plotted on Figure 2. In practice, this lower bound describes well the learning dynamic when the number of samples is high compared to the effective dimension of the space of functions considered.

**Same examples in Fourier.** To transpose the previous example in the Fourier domain, one can consider $f^*(x) = \cos(\omega_0 x_1)$ together with some translation-invariant kernel. We illustrate this case on Figure 9. The deterioration of the rates with respect to dimension can be understood precisely. In harmonic settings, such as on the torus with uniform measure, one can consider $f^*$ as an eigenfunction of the integral operator $K$ associated with the eigenvalue $\lambda_{\omega_0}$. The lower bound on the bias is given by

$$\mathcal{S}(\lambda^{-1} K) = \frac{\lambda^2}{(\lambda + \lambda_{\omega_0})^2}.$$

When $k$ is translation-invariant, $k(x, y) = q(x - y)$, we get that

$$\mathrm{Tr}(K) = \sum_{\omega \in \mathbb{Z}^d} \lambda_\omega = \mathrm{Tr}\left( \mathbb{E}[\varphi(X)\varphi(X)^\top] \right) = \mathbb{E}[\varphi(X)^\top \varphi(X)] = \mathbb{E}[k(X, X)] = q(0).$$

When $q(0)$ does not depend on $d$, this quantity is constant. On the other hand, we expect $\lambda_\omega$ to decrease with $\|\omega\|$. But since the number of frequencies below $\|\omega_0\|$ grows exponentially with the dimension, in order to keep this sum constant $\lambda_{\omega_0}$ has to decrease exponentially fast with the dimension, hence the bias will increase exponentially fast with the dimension.

**Example of "wrongfully" arbitrarily fast convergence rates.** To further emphasize the importance of constants and transitory regimes, let us discuss an even simpler example. Assume that one wants to learn a polynomial of a unknown degree $s \in \mathbb{N}$ in a noiseless setting; or equivalently, learn an analytical function such that $f^{(s+1)} = 0$ for an unknown $s \in \mathbb{N}$. This polynomial can be learned exactly when provided with as many points as the unknown coefficients in the polynomial, meaning that the generalization error will almost surely goes to zero when provided enough points. As a consequence,

$$\forall\, h : \mathbb{N} \to \mathbb{R}, \qquad \mathcal{E}(f_n) \leq O(h(n)),$$

where $f_n$ is defined in (7) with $\mathcal{F}$ the space of polynomials of any degree. In other terms, we are able to prove *arbitrarily fast convergence rates*. Yet, such convergence rates hide constants that are the real quantities governing convergence behaviors of any learning procedure. Figure 9 shows how the number of coefficients in a Taylor expansion of order $s$ is once again the right quantity to look at.

### C.3 Different convergence rates profiles

Figure 4 was computed with the Gaussian kernel on either the one-dimensional torus or $\mathbb{R}$. One hundred runs were launched and averaged to get a meaningful estimate of the expected excess risk. Convergence rates were computed for different hyperparameters, and the best set of hyperparameters (changing with respect to the number of samples but constant over run) was taken to show the best achievable convergence rate.

**Fast then slow profile.** Let us focus on the example provided by $f^* : x \mapsto \exp(-\max(x^2, M)) - \exp(-M)$ (note that we substract $\exp(-M)$ so that the function goes to zero at infinity, which remove the burden of learning a constant offset with the Gaussian kernel). Note that, for $q_\sigma = \exp(-x^2/\sigma)$, the convolution $q_\sigma * f$ for a large $\sigma$ will not modify $f$ much, while making it analytical. This follows from Fourier analysis: if $f$ is integrable, its Fourier transform is bounded; since a convolution corresponds to a product in Fourier, and since the Fourier transform of $q_\sigma$ decays exponentially fast, so does $f * q_\sigma$, which implies its analytical property. As a consequence, all functions are close to

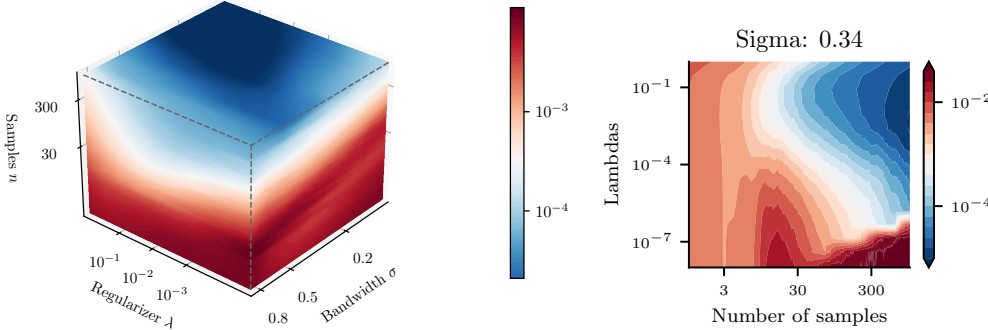

**Figure 10:** Excess risk when the target at the top is taken as $f^*(x) = \exp(-\max(x^2, M)) - \exp(M)$ with $M = 1/4$, and $x \in \mathbb{R}$ with unit Gaussian distribution. Note that the learning of the smooth part is more efficient when the regularizer is big, which forces the reconstruction to be smooth.

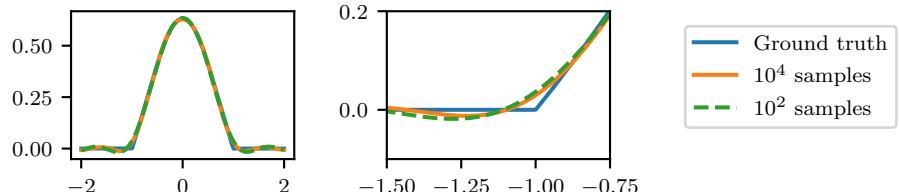

**Figure 11:** Example of a smooth target function with a $C^1$-singularity whose estimation is expected to showcase convergence rates that decrease first fast and then slowly. The $x$-axis represents the input space $\mathcal{X} = [-2, 2]$, the $y$-axis represent the output space $f^*(x)$ and $f_{n,\lambda}$ for $n = 10^2$ and $n = 10^4$. The first fast decrease of excess risk corresponds to the easy estimation of the coarse details of the function, while the slow decrease thereafter corresponds to the precise estimation of the $C^1$-singularity. The target function $x \mapsto \exp(-\max(x^2, 1)) - \exp(-1)$ is represented in blue, the estimation with $10^4$ samples is represented in orange, and the one with $10^2$ samples is represented in dashed green. The left picture zooms in on the estimation of the singularity. We see that the increase from $10^2$ to $10^4$ does not lead to a much better estimate.

analytical functions, whose approximation should exhibit convergence rates in $O(n^{-1})$. In particular, for $f^*$ defined as before for a some $M$ large enough, without enough observations one will not be able to distinguish between $f^*$ and $q_\sigma * f^*$, and as the number of sample first increases, one will learn quite fast a smooth version of $f^*$. After a certain number of samples, the learning will stall until enough points are provided to distinguish between $f^*$ and its smoothing, and learn the $C^1$-singularity of the former. Figures 10 and 11 illustrate this observation. Note that similar reasoning could be made for any RKHS that is dense in $L^2(\rho_{\mathcal{X}})$.

**Slow then fast profile.** The slow then fast profile was computed with $\mathcal{X} = \mathbb{S}^1$, $\rho_{\mathcal{X}}$ being uniform and $f^* : x \mapsto \cos(2\pi\omega x)$ with $\omega = 20$. One hundred runs were launched and averaged to get an estimate of the excess risk of the estimator in (8) for different values of $\sigma$ and $\lambda$. Again, the best results for different sample sizes were reported to get an estimate of convergence rates on Figure 4. A log-log-log-log plot of the results is provided on Figure 12.

### C.4 Exploring the low-sample regime

**Looking at the empirical weights.** While the previous paragraph discusses the weights $\alpha_X(x)$ when given access to the full distribution, similar derivations can be made when accessing a finite number of samples. Indeed, kernel ridge regression reads as

$$f_{\lambda,n}(x) = \sum_{i \in [n]} Y_i \widehat{\alpha}_i(x), \qquad \widehat{\alpha}(x) = (\widehat{K} + n\lambda)^{-1} \widehat{K}_x \in \mathbb{R}^n,$$

where

$$\widehat{K} = (k(X_i, X_j))_{i,j \in [n]} \in \mathbb{R}^{n \times n}, \qquad \widehat{K}_x = (k(X_i, x))_{i \in [n]} \in \mathbb{R}^n.$$



**Figure 12:** Excess of risk when the target at the top is taken as $f^*(x) = \cos(2\pi\omega x)$ with $\omega = 20$, and $x \in \mathbb{S}^1 = \mathbb{R}/\mathbb{Z}$ uniform on the circle. Observe how the risk first stalls, before learning the function quite fast. The two graphs $\{(n, \mathcal{N}_a(\lambda, \sigma))\}$ for $a \in \{1, 2\}$ are plotted with the blue lines.

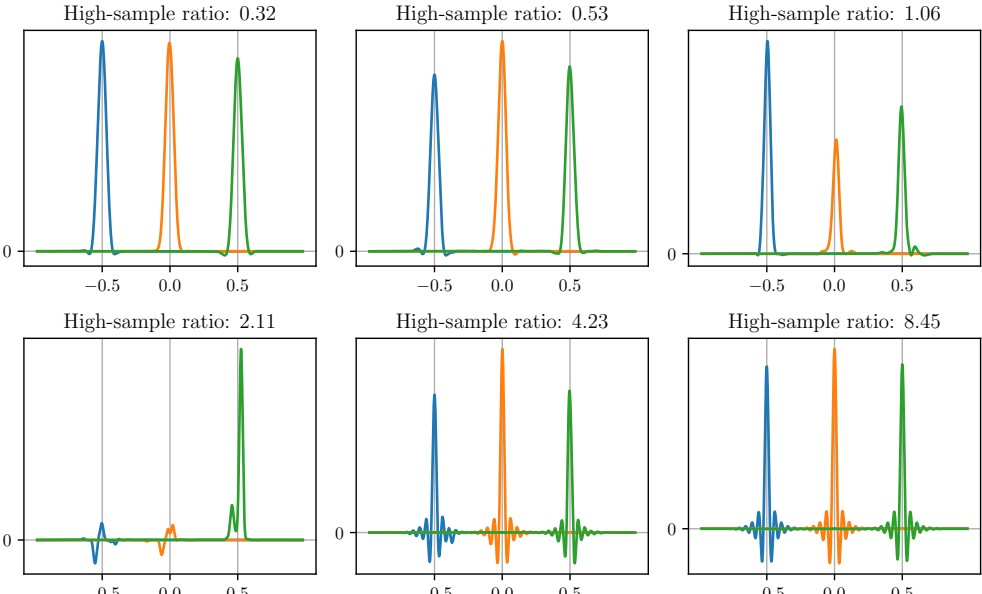

**Figure 13:** Same picture as Figure 6 yet with $\sigma = .05$, which leads to an effective dimension $\mathcal{N} = 85$.

Note that $\widehat{\alpha}_i(x) = \widehat{\alpha}_{X_i|\mathbb{X}}(x)$ where $\mathbb{X} = (X_1, \cdots, X_n)$ is the input dataset. As a consequence,

$$\mathbb{E}_{\mathcal{D}_n}[f_n(x)] = \sum_{i \in [n]} \mathbb{E}_{\mathcal{D}_n}[Y_i \widehat{\alpha}_{X_i|\mathbb{X}}(x)] = n \cdots \mathbb{E}_{\mathcal{D}_n}[Y_1 \widehat{\alpha}_{X_1|\mathbb{X}}(x)].$$

In other terms, $f_n$ is a bias estimator whose average is defined as

$$\mathbb{E}_{\mathcal{D}_n}[f_n] = \mathbb{E}_{(X,Y)}[Y \widehat{\alpha}_X], \qquad \widehat{\alpha}_X = n \cdot \mathbb{E}_{\mathbb{X}}\left[\widehat{\alpha}_{X_1|\mathbb{X}} \,\big|\, X_1 = X\right]. \tag{34}$$

These are the weights plotted on Figures 6 and 13. In order to compute those weights efficiently, one can use the block matrix inversion. Using the sliced indices matrix notations, we have

$$\begin{aligned}
\widehat{\alpha}_{X_n|\mathcal{D}_n}(x) &= [(\widehat{K} + n\lambda)^{-1}\widehat{K}_X]_n = [(\widehat{K} + n\lambda)^{-1}]_{n,:} \times \widehat{K}_x \\
&= [(\widehat{K} + n\lambda)^{-1}]_{n,:n-1} \times [\widehat{K}_x]_{:n-1} + [(\widehat{K} + n\lambda)^{-1}]_{n,n} \times [\widehat{K}_x]_n \\
&= -(b - x^\top A^{-1} x)^{-1}([\widehat{K}_x]_n - x^\top A^{-1} \times [\widehat{K}_x]_{:n-1}).
\end{aligned}$$

where

$$\widehat{K} + n\lambda I = \begin{pmatrix} A & x \\ x^\top & b \end{pmatrix} = \begin{pmatrix} [\widehat{K}]_{:n-1,:n-1} + n\lambda & [\widehat{K}]_{n,:n-1} \\ [\widehat{K}]_{n,:n-1}^\top & [\widehat{K}]_{n,n} + n\lambda \end{pmatrix}.$$

Denoting

$$\tilde{K} = (k(X_i, X_j))_{i,j \in [n-1]} \in \mathbb{R}^{n-1 \times n-1} \tilde{K}_x = (k(X_i, x))_{i \in [n-1]} \in \mathbb{R}^{n-1},$$

we get

$$\widehat{\alpha}_{X|\mathcal{D}_n}(x) = \left(k(X,X) - Z_X^\top Z_X + n\lambda\right)^{-1}\left(k(X,x) - Z_X^\top Z_x\right), \qquad Z_x = (\tilde{K} + n\lambda)^{-1/2}\tilde{K}_x.$$

