# OpenReview forum: "How many samples are needed to leverage smoothness?"
_NeurIPS.cc/2023/Conference — NeurIPS 2023 poster_

### Official Review · Reviewer_jRbT · 2023-06-29

**Soundness:** 4 excellent
**Presentation:** 2 fair
**Contribution:** 3 good
**Rating:** 7
**Confidence:** 3

**Summary:**

This paper first states and proves Theorem 2, which bounds the generalisation error from above and below. The authors then use this result in Section 3 to find lower bounds on this generalisation error, quantifying the curse of dimensionality with Taylor expansion in the case of uniform measure on the unit cube and with Fourier expansion in the case of uniform measure on the torus. The final section of the paper investigates the transitory regime, theoretically in the high-sample regime and empirically in the low-sample regime.

**Strengths:**

1. This paper considers a problem which I find very important and interesting, i.e. the sample complexity required for smoothness of target functions to start overcoming the curse of dimensionality.

2. The mathematics seem mostly solid, albeit some notational inconsistencies and presentational issues (see Weaknesses and Limitations), and the authors clearly put in a lot of effort to make the proofs solid (although I didn't have time to go through all of the proofs).

3. For a theory paper, the authors valued and put in a lot of effort into experimental findings where theory could not extend to, e.g. in the low-sample transitory regime. This I find very valuable.

**Weaknesses:**

1. There are a few places in the paper that leave seemingly important results as conjectures. For example, on L147, the authors state "We expect the right-hand side to be improvable with the removal of N_1(lambda) in front of the rates (which is due to our usage of concentration inequalities on operators rather than on scalar values)". If the improvement is needed for the subsequent statement within the paragraph and the authors have an idea of how to go about it, I wonder why they didn't do it? This comes up again on L418.

2. The mathematical presentation could be more solid. For example, there are a few instances of notations being used without being introduced, and the same notation being used for different things - see Minor Comments below. Moreover, Theorem 2, which is the main result of the paper, is presented with some issues with regards to clarity - the assumptions under which the result holds can only be found scattered among the proofs in the Appendix. These assumptions do not seem as mild as the authors claim. For example, on L436, it is stated that for Assumption 2 to hold, F has to be finite-dimensional, which is very restricting, or the bound stated in the theorem has to be loosened.

3. Despite the fact that the paper is in general written in a way to suggest very wide generality, the set-up is restricted in Section 2.1 to RKHSs, and the results do not seem to extend to other hypothesis spaces. Of course, RKHSs can be and indeed are large classes of functions, but do exclude e.g. neural networks and other interesting learning situations, and perhaps this restriction should be stated more clearly throughout the paper.

Minor comments

L16: by hands -> by hand

L53: four-folds -> fourfold

L70: Possibly worth defining the notation [n]={1,2,...,n}.

L86: Possibly worth defining the notation for the floor function.

L125-126, L159-160, L201: The notation q is used for different purposes. Perhaps worth changing one of them for consistency?

L223: The same goes for the notation x used here, as opposed to the use of x in other parts of the paper.

L238: In the denominator, 2alpha+d is missing the +.

L368, displayed equation right below: to be consistent with (6), f_n should be f_{n,lambda}. Same is true for every f_n that comes afterwards.

**Questions:**

L372, displayed equation: I don't recall theta_* being introduced? I assume it's the theta in H such that <phi(x),theta_*>=f*(x)? But then this assumes that f* is in F? In fact this assumption only gets introduced on L374, so is it correct to write the displayed equation after L372?

**Limitations:**

The authors adequately discussed the limitations in the conclusion section, as future work. These include the need to investigate other priors than smoothness, how neural networks exploit such priors, and other loss functions.

---

> ### Author Rebuttal · Authors · 2023-08-07
>
> Dear Reviewer,
>
> Thank you very much for your appreciation of our work, your highly valuable feedback, and for spotting typos.
> We address here your main concerns.
>
> 1. Thank you for your question.
>
> a) The right-hand side in (10) is enough for subsequent paragraphs.
>
> b) While we expect the possibility of refining the bounds, we were unable to prove it. We will clarify that our expectation takes the form of a "conjecture."
>
> To provide more details:
>
> b) Concentration inequalities are primarily known for empirical averages, and convergence proofs for kernel methods often involve expressing differences between distinct empirical averages.
> Current "state-of-the-art" proofs notably involve rewriting a difference between empirical and real matrix inversions as a difference between matrix averages, which is bounded using recent concentration inequalities on operators (mainly due to the works of Stanislav Minsker and Joel Tropp).
> These inequalities result in an additional factor of $N(\lambda)$ in front of the rates (compared to scalar concentration).
>
> a) The additional term means that to ensure that the right-hand side of (10) is of higher-order compared to the left-hand side, we need $a_n^{1/2} {\cal N}(\lambda)$ to go to zero as $n$ goes to infinity.
> This implies that we cannot assume overly weak source conditions on $f^*$ (i.e., the largest $q$ such that $f^*$ belongs to $K^q L^2$ should not be too small), otherwise one would need to consider a big search space ${\cal F}$ to approach $f^*$, which would lead to a too big value of ${\cal N}(\lambda)$ compared to $a_n$.
> This is a typical caveat with weak source conditions (also known as "hard learning problems").
> Nevertheless, throughout the rest of the paper, we only consider $f^*$ (e.g., Proposition 2) that indeed belongs to $K^q L^2$ for any $q$, ensuring there are no issues in that regard.
>
> 2. We thank you for helping us strengthen our draft, particularly regarding notations.
> For Assumption 2, one can use $c^2 \simeq \lambda^{-1}$ which primarily results in the condition that $\lambda^{-1} {\cal N}(\lambda) a_n^{-1/2}$ must approach zero.
> This restriction further narrows down the required source condition, similar to point 1. However, this adjustment does not impact the rest of the discussion.
> Moreover, we have added a generic lower bound that does not depend on any assumption as detailed in the generic rebuttal answer.
>
> 3. We agree with your concern, we have addressed it with points 1 and 2 in the generic rebuttal answer.
>
> You are right that the Assumption on line 374 and $\theta_*$ should have been introduced on Line 372. Otherwise the equation afterwards is meaningless.
> Indeed, the proof assumes that $f^*$ belongs to the RKHS, before extending the results by density.
> We will clarify it.
>
> Thank you for your careful reading, and the many valuable suggestions to improve our paper.

---

> > ### Comment · Reviewer_jRbT · 2023-08-13
> > **Thank you**
> >
> > Thank you for taking time to answer my questions. After reading the authors' answers and the other reviews, I maintain my score, and would like the authors to improve the presentation quite a bit for the camera-ready version (if accepted), as they propose to do so.

---

### Official Review · Reviewer_n2Cg · 2023-07-06

**Soundness:** 2 fair
**Presentation:** 1 poor
**Contribution:** 2 fair
**Rating:** 6
**Confidence:** 3

**Summary:**

This paper studies transitory, non-asymptotic regimes in the generalization error of kernel methods.
It focuses on the use of smoothness to break the curse of dimensionality, and the sample complexity needed to achieve the fast asymptotic rates.
In several settings, the authors show that the classical analysis overlooks constants that can be exponential in the dimension.

EDIT: I have read the author's rebuttal, which partially addressed my concerns (this depends on the changes made by the authors to the writing).

**Strengths:**

The paper makes a very important point that is often overlooked.
My lack of expertise in the area, as well as the technicality of the writing, prevent me from judging of the novelty or significance of the analysis.

**Weaknesses:**

The paper is difficult to follow. Part of it is due to the fact that I am not an expert in this area at all, and this paper seems to be written for people with a significant deal of intuition about statistical learning analyses. I think it would be very useful for the authors to explain this intuition to the reader, rather than assuming they have it. Part of it is also due to the setting of the paper, which seems to constantly switch: in some cases the authors consider kernel ridge regression with a fixed kernel, in other cases an abstract family $\mathcal F_t$, in yet other cases polynomials of fixed degree. In the end, I have no overview of the precise results of the paper, and it reads to me like a series of disconnected illustrations.

Minor remark: I would suggest to change the notation $\mathcal N_\infty(\lambda)$, as one would expect it to be equal to $\Vert K (K + \lambda)^{-1} \Vert_\infty$.

Typos:
- line 16: hands -> hand
- line 53: four-folds -> four-fold
- line 97: a -> an
- line 113: space -> spaces
- line : $2\alpha d$ -> $2\alpha + d$

**Questions:**

What do the authors mean by "so to be able to consider function spaces $\mathcal F_t$ containing interesting-enough functions with still having more samples than the effective dimension of $\mathcal F_t$"? (lines 173-174)?

What exactly is the relationship between the $\mathcal N_a$ quantities? In the same paragraph (lines 150-160), the authors state that they all behave similarly, while making the distinction between the regimes when $\mathcal N_\infty(\lambda) \leq n$ and $\mathcal N_2(\lambda) \leq n$.

Which kernel is being used in Proposition 1 to define the quantities $\mathcal N_2$ and $\mathcal B$?

Could the authors summarize their results? I do not see the link from the propositions to the claims of the paper.

**Limitations:**

The authors acknowledge several limitations, among which being limited to the square loss.

From my own (limited) understanding,the main limitation of the paper is that does not really answer the question in the title. The authors make various disconnected cases that quantities that are exponential in the dimension might appear for some learning algorithms in some settings.

---

> ### Author Rebuttal · Authors · 2023-08-08
>
> Dear Reviewer,
>
> Thank you very much for your review, and for spotting numerous typos.
> Your perspective was highly valuable for us to enhance the readability of our paper, which we discussed in the general rebuttal answer.
> Our immersion in technical details has diverted us from a simple presentation of the overarching narrative.
>
> There are three levels of abstraction:
> 1. Abstract function classes ${\cal F}$ that could encompass any sets of (measurable) functions from ${\cal X}$ to ${\cal Y}$.
> 2. Reproducing Kernel Hilbert Spaces (RKHS), a specific type of function class with an inherent inner Hilbertian structure.
> 3. Polynomials, a specialized type of RKHS, endowed with a Euclidean structure on polynomial coefficients.
>
> Throughout the paper, we transition between these different levels of generality to provide specific results while demonstrating their broader applicability.
> We have revised our draft to ensure a smooth progression from one level to the next, as outlined in point 2 of the general rebuttal answer.
>
> In reference to Lines 173-174, our intention was to convey a trade-off between the scope of the function class ${\cal F}$ – which impacts the proximity of $f^*$ to ${\cal F}$ (represented by ${\cal S}$) – and the risk of overfitting, influenced by the size of the function class (represented by ${\cal N}$).
> More precisely, without implicit bias, one needs to be in the high-sample regime (${\cal N} < n$) to avoid overfitting.
> This constrains the diversity of functions within ${\cal F}$, which will typically be devoid of fine-grained details (linked with high-order derivatives, polynomials, and Fourier coefficients), hence unable to harness high-order smoothness without accessing a large number of samples $n$.
> We will remove this sentence and provide the intuition more sharply as detailed in point 4 of our generic rebuttal answer.
>
> It is true that the ${\cal N}$ notations are not ideal; we used some convention in the recent literature.
> We will change the notations to make it clearer with $N_{\infty}$ becoming $N_{+}$.
> We have that $N_2 \leq N_1 \leq N_\infty$ (Proposition 7 in submission's appendix).
> Those quantities behave similarly when there is no heavy-tail phenomenon in the input data (although to be rigorously general, one should distinguish the "meso"-sample regime where $N_2 \leq n \leq N_\infty$).
>
> Proposition 1 uses the kernel $k(x, x') = \phi(x)^\top \phi(x')$ where $\phi$ is defined by concatenating monomials. We will make it clearer.
>
> The crux of this paper is to show that leveraging smoothness is not enough for truly overcoming the curse of dimensionality.
> We explain that the curse of dimensionality should be read in the constants in front of convergence rates instead of read in exponents (where it does not show up).
> Our new lower bounds, established through Theorem 2, along with Propositions 1, 2, and Section 3.3, substantiate this claim.
> In particular, Theorem 2 elucidates how the leading order in convergence rates aligns with the left-hand side of equation (10), while Propositions 1 and 2 show that this quantity grows exponentially with the dimension of the input space.
> It highlights the insufficiency of smoothness as a robust prior for efficient high-dimensional learning.
> We agree that there was some extra work needed from the reader to understand the full picture: notably we should have mentioned clearly that the discussed methods are optimal to leverage smoothness.
> We have detailed this fact in our general rebuttal answer.
>
> Lastly, because our RKHS analysis is accurate up to higher-order terms, that are insignificant in the "underparameterized" regime, but could change the picture in the "overparameterized" regime (when one has much fewer samples than dimensions), we turned to experiments to shed light on those regimes, where no magic happens (mainly the algorithm focuses on a coarse reconstruction of $f^*$, which could be, or not be, beneficial).
>
> We hope that the rebuttal will help you to better appreciate our work.
> We stay at your disposal for any additional questions.

---

> > ### Comment · Reviewer_n2Cg · 2023-08-11
> >
> > I thank the authors for their detailed answer. I appreciate the efforts to explain the paper more clearly, which has helped me further my understanding. This discussion has already been useful to me, thank you!
> >
> > As far as I understand, here is the revised summary of the paper, with some additional questions:
> > - Traditional analyses only focus on rates and hide constants which may be exponential in the dimension.
> > - As an illustration, polynomials of bounded degree and band-limited functions can be learned with a linear rate (by leveraging the finite-dimensionality of the corresponding function space) but with an exponential constant (due to the exponential dimensionality of the space). For polynomials, I'm now confused regarding the binomial coefficient ${\alpha + d \choose d} = O(\max(\alpha,d)^{\min(\alpha,d)})$: it is polynomial in both $d$ and $\alpha$ separately but exponential when $\alpha \sim d$. As such, I'm not sure how to interpret Theorem 1 in the revised outline.
> > - In any case, Theorems 1 and 2 hint that exponential constants might also be present in the traditional upper bound, but in this general setting the analysis is not sharp and thus inconclusive.
> > - A sharper analysis can be made in the kernel setting, where one obtains matching lower and upper bounds if the training set size is not too small. The rate is then $\min_\lambda A(\lambda, n)$. Doesn't this already allow for arbitrary convergence profiles?
> > - The authors then go to the meta-algorithm setting where one has a family of kernels indexed by $t = (\sigma, \lambda)$ and the rate becomes $\min_{\lambda,\sigma} A_\sigma(\lambda, n)$. However the authors consider a specific choice of $t$ (line 265) which is not necessarily optimal? In any case, arbitrary convergence profiles can be obtained in this more general case.
> > - If I understand correctly, the Sobolev setting of the submission's Section 4 is only useful to provide an explicit characterization of the kernel eigenfunctions and eigenvalues, but the results could be stated for an arbitrary kernel.
> >
> > Please correct me if I'm wrong. I have raised my score to a 5, as I feel like the proposed changes are in the right direction for improved clarity. I'm still hesitant to recommend acceptance, as the big picture remains a bit elusive to me despite investing again a significant amount of time reading the reviews, author rebuttal, the original submission, and the revised outline. In particular:
> > - I do not understand the relationship between the revised sections 2.1 and 2.2: as the lower and upper bounds have different rates in $n$ I do not see how they complement themselves, and the presence of constants exponential in $d$ in the lower bound do not necessarily imply that there are such constants in the upper bound. Also, the hypotheses on $f^*$ are different in both sections.
> > - Couldn't one compute explicitly $\min_\lambda A_(\lambda, n)$ in some cases to show that it is of the form e.g. $c^d n^{-\gamma}$, using a setting similar to the submission's Section 4? This would seem to be more direct in making the paper's point.
> > - Is the kernel selection setting (meta-algorithm) really necessary to showcase the presence of arbitrary convergence profiles?

---

> > > ### Author Response · Authors · 2023-08-11
> > > **Discussion with Reviewer n2Cg**
> > >
> > > Thank you very much for starting the discussion phase.
> > > We are happy to read your highly relevant questions, it seems that you understand our work really well now.
> > > It shows how valuable the reviewers feedback for improving the readability of our draft.
> > >
> > > > I'm now confused regarding the binomial coefficient : it is polynomial...
> > >
> > > You are right, this is not an exponential constant, but it is still a constant that is quite big, for example when $d=100$, if one wants to leverage fourth order smoothness $\alpha = 4$, one needs at least $4,598,126 = {104 \choose 4}$ points to be in the high-sample regime, and get meaningful estimate of $f^*$.
> > > We have removed the word "exponential" there.
> > >
> > > > In any case, Theorems 1 and 2 hint that exponential constants might also be present in the traditional upper bound, but in this general setting the analysis is not sharp and thus inconclusive.
> > >
> > > This is true, Theorems 1 and 2 illustrates that "smoothness is not enough to break the curse of dimensionality", but it does not states exactly which are the minimax optimal constant in front of $O(n^{-2\alpha / 2\alpha + d})$.
> > >
> > > > Doesn't this already allow for arbitrary convergence profiles?
> > >
> > > Yes it does, Section 4 is only here to show how we can derive them explicitly -- since the $A(\lambda, n)$ are highly related for different $n$ and $\lambda$, it is not clear *a priori* that any convergence profile could be derived from $n \to \min_\lambda A(n, \lambda)$.
> > >
> > > > However the authors consider a specific choice of $t$ (line 265) which is not necessarily optimal?
> > >
> > > Yes, we have remove the notations with $t$, we have simply rewritten ${\cal N}$ as a function of ${\cal N}(K)$ and said that $K$ depends on both $\phi$ (or $k$), and the hyperparameters ($\lambda$, $\sigma$ or the degree and partition for local polynomials).
> > > This would make it extra clear that all algorithms consist in cross-validating the operator $K$ by modifying a few hyperparameters (before we had this intricate way to first introduce $t$ then to explain $t$ would be cross-validated).
> > >
> > > > If I understand correctly, the Sobolev setting of the submission's Section 4 is only useful to provide an explicit characterization of the kernel eigenfunctions and eigenvalues, but the results could be stated for an arbitrary kernel.
> > >
> > > Yes, this is true. We will make it clearer.
> > >
> > > > I do not understand the relationship between the revised sections 2.1 and 2.2: as the lower and upper bounds have different rates in $n$
> > >
> > > The goal is to show that at first glance it seems that "smoothness breaks the curse of dimensionality" (section 2.1), but that a closer look shows that it is still here but hidden in constant (section 2.2).
> > > The theorems in Section 2.2 have much stronger assumptions on the target function, which is what we want (we want strong lower bound even under really strong smoothness assumptions).
> > >
> > > > Couldn't one compute explicitly in some cases to show that it is of the form.
> > >
> > > Yes, this is exactly what we did in the submission to arrive at Propositions 1 and 2.
> > >
> > > > Is the meta-algorithm really necessary to showcase the presence of arbitrary convergence profiles?
> > >
> > > We understand that the presentation of the meta-algorithm causes extra work.
> > > We have rewritten ${\cal N}$ as a function of $K$ alone, with $\lambda$ integrated in the kernel, to make sure that the meta structure appears clearly (in contrast with the submission where it appears unclear).
> > >
> > > Thank you again for your highly valuable feedback. We are really grateful for the time you devoted to our paper, it drastically helps the clarity of our exposition.

---

> > > > ### Comment · Reviewer_n2Cg · 2023-08-12
> > > >
> > > > Thank you again for your answers.
> > > >
> > > > Please allow me to insist regarding the logical relationship between sections 2.1 and 2.2.
> > > > Section 2.1 shows a rate in $O(n^{-2\alpha/2\alpha + d})$. Under stronger assumptions on $f^*$ (that it belongs to a finite-dimensional space), section 2.2 shows a minimax-optimal error bound in $c(\alpha,d) /n$ where the constant $c(\alpha,d)$ is very large and can be exponential in $d$. To me, this shows that the finite-dimensional rate in $O(1/n)$ is cursed by dimensionality, *but this does not say anything about the constant in front of the $O(n^{-2\alpha/2\alpha + d})$ upper bound which still applies*. The convergence profile for one of the settings considered in section 2.2 could thus be
> > > > $${\\cal E}(f_n) = \\begin{cases} n^{-2\\alpha/2\\alpha + d} & \\text{for } n \\ll c(\\alpha, d) \\\\ c(\\alpha,d)/n & \\text{for } n \\gg c(\\alpha, d) \\end{cases}$$
> > > > and in this case I wouldn't say that there is curse of dimensionality, as taking the limit $n \propto \alpha \propto d \to \infty$ leads to ${\cal E}(f_n) \sim n^{-\gamma}$ without any exponential constant.
> > > >
> > > > Unless I'm mistaken, section 2 thus doesn't *show* that "smoothness is not enough to break the curse of dimensionality", though it provides some evidence for this conjecture.

---

> > > > > ### Author Response · Authors · 2023-08-12
> > > > > **Logical relationship between Sections 2.1 and 2.2**
> > > > >
> > > > > Thank you for expressing your concerns.
> > > > >
> > > > > > this does not say anything about the constant in front of the $O(n^{-2\alpha/2\alpha + d})$ upper bound which still applies.
> > > > >
> > > > > Since the assumption in Section 2.1 is weaker than the assumptions in Section 2.2, the upper bound in $c_{2.1} n^{-\gamma}$ has to be lower bounded by $c_{2.2} / n$ (where $c_{2.2}$ grows fast with dimension).
> > > > > Taking $n=1$, it implies that $c_{2.1} \geq c_{2.2}$.
> > > > > Now if we want to be really sharp, the picture could be slightly more subtle, since we have seen how Theorem 1 in the submission should rather be written as $c_{2.1} n^{-\gamma} (1 + O({\cal N} /  n))$ with a smaller $c_{2.1}$ (at the price of adding the $O({\cal N} / n)$).
> > > > > Anyways, one can not prove rates faster than $c_{2.2} / n$, which means that when $n$ is not "indecently large", one can not guarantee a small excess risk when solely leveraging smoothness.
> > > > >
> > > > > To discuss your example (there might have been a typo in your answer), Sections 2.1 and 2.2 shows that one can consider the profile
> > > > > $${\cal E}(f_n) = \begin{cases} c(\alpha,d)/n & \text{for } n \ll c(\alpha, d); \\ n^{-2\alpha/2\alpha + d} & \text{for } n \gg c(\alpha, d). \end{cases}$$
> > > > > What we argue if that this type of guarantee are not really meaningful in modern machine learning, where typically $n \ll c(\alpha, d)$ and $c(\alpha, d)/n$ is large (think of images where $n$ could be several billions ($n=10^{11}$ is the order of magnitude of the number of images of the internet), but $d$ is about a million ($10^6$ is the order of magnitude of the number of pixels in a good resolution image)).
> > > > >
> > > > > Thanks again for asking for clarification.

---

> > > > > > ### Comment · Reviewer_n2Cg · 2023-08-12
> > > > > >
> > > > > > Thank you for addressing my concerns. I have no further questions. I find the technical results of the paper very interesting and valuable to the community, when presented clearly.
> > > > > >
> > > > > > I have raised my score to a 6, and my confidence score to 3.

---

### Official Review · Reviewer_7mBD · 2023-07-07

**Soundness:** 4 excellent
**Presentation:** 4 excellent
**Contribution:** 2 fair
**Rating:** 6
**Confidence:** 3

**Summary:**

 This paper aims to understand the transitory regimes where the data is not enough to enter the asymptotic regime. When the smoothness is leveraged through Taylor expansions, the number of samples needed to enter the aysmptotic regime is the number of polynomial in d-dimension.  When smoothness is leveraged through Fourier expansions, the picture will depend on how the Fourier coefficinet decays.

**Strengths:**

The paper investigated an important problem of how the constant looked like before the polynomial rate. This makes a contribution to understand what's the prior needed to break the curse of dimensionality.

**Weaknesses:**

This paper is conceptually very interesting. However the reviewer is very suspicious about the exact theorem proved.  The reviewer is open to raise the score if the author can hlep me to understand the theorems and their technical contribution better.

From my understanding theorem 2 is standard.

If we characterize the smoothness using taylor expansion. The main conclusion is the sample complexity depends on the number of polynomials. I'm wonder is this conclusion not surprising? and my second question is the sample complexity not depend on the volume in high dimension?

For the kernel characterization,  from prop3 till (14) looks standard to me. Am I missing anything? From my viewpoint it may have a better characterization of the fast than slow/slow that fast regime using the representing kernel Sobolev norms. Different spectrums have the same kernel norm will have different Sobolev norm. Using this it's easy to have this characterization. (takeoff between norm and convergence rate respect to the number of samples)


**Questions:**

See above

**Limitations:**

I'm suspicious about the technical contribution made by the paper and how the theorem reveals the main information of the paper.

---

> ### Author Rebuttal · Authors · 2023-08-08
>
> Dear Reviewer,
>
> We are glad you find our paper conceptually very interesting.
> Thank you for raising constructive concerns about our contributions; we will try to clarify them here.
>
> Regarding Theorem 2, while similar upper bounds can be found in the literature (see e.g., Mourtada and Rosasco 2022), the main contribution of Theorem 2 is that it provides a matching lower bound.
> We are not aware of any such lower bound in the literature.
> In the narrative of our paper, this lower bound is crucial, as it constrains the learning curve in its transitory and asymptotic regimes both from above and from below.
>
> Regarding Proposition 1 on Taylor expansions, it is not really surprising that the sample complexity depends on the dimension of the space of polynomials with bounded degree, which corresponds to the number of unknown in one Taylor expansion.
>
> Regarding volume in Taylor expansion, as we argue in Appendix A.4.1, paragraph "Covering issues in high-dimension?", it may indeed be surprising that an explicit dependence on the volume does not show up in Proposition 1.
> In fact, one would expect needing exponentially more local neighborhoods to cover a $d$-dimensional domain as $d$ grows.
> However, by balancing bias and variance to the minimax rate $O(n^{-2\alpha/(2\alpha+d)})$, it appears explicitly that the number of local neighborhoods must be set to have cardinality $O(n^{d/(2\alpha+d)})$, which does not deteriorate with the dimension.
>
> Regarding volume in Fourier expansions, it should be noted that the volume of the sphere decreases exponentially fast with the dimension (the base object with unit volume being the unit hypercube).
> As a consequence, series-integral comparisons (i.e., replacing the counting measure by the Lebesgue measure in Proposition 3) are somewhat deceptive in high-dimension, which explains the intricate formulation of Proposition 2.
>
> Yes, Proposition 3 is somewhat standard.
> It is possible to get a characterization through Sobolev norms.
> In essence, we have an upper bound on rates of the form $U(n) = \min_{a < \alpha} c_a n^{-2a / (2a + d)}$ where $c_a$ relates to the norm of $f$ in the Sobolev space $H^a$.
> One might expect that to a given $n$ corresponds a certain $a$ which is the minimizer defining $U(n)$, and that the real error ${\cal E}(n)$ might locally decrease in $cn^{-a}$.
> As such, one could expect that $a$ would increase little by little as one is able to leverage more and more smoothness.
> However, 1. the real error does not have to behave as its upper bound, 2. the norms of $f^*$ in different Sobolev spaces are not independent, which makes it harder to create any type of given convergence rates profile (in contrast with our analysis that totally decorrelates the power laws), 3. it is not clear to us how this analysis allows to explain fast then slow profiles (such as the ones described by Song Mei et al. 2022).
> On the other hand, 1. our analysis is not really different from Sobolev, which are basically integration of the Fourier transform of $f$ against some power law, 2. it describes the behavior of the real error, and not of an upper bound. 3. it can be used to derive fast then slow profiles or slow then fast ones.
> We will clarify this.
>
> To conclude, it seems that you have a really good intuition on the described phenomenon.
> Our work consisted in working out the details, which actually reveal subtleties (e.g., the exact variance and bias terms are not exactly the ones usually proven in upper bounds).
> We hope that this is a good contribution in your eyes.

---

> > ### Comment · Reviewer_7mBD · 2023-08-17
> > **Thank you and I've increased the score**
> >
> > I find the problem quite interesting conceptually and understand the novelty of characterizing the problem precisely. I still believe that using Sobolev space can characterize the slow but maybe this is beyond the discussion in the paper (this paper already discussed a lot of things in my mind and sufficient to be a conference at Neurips)

---

### Official Review · Reviewer_yXo1 · 2023-07-08

**Soundness:** 3 good
**Presentation:** 3 good
**Contribution:** 3 good
**Rating:** 6
**Confidence:** 3

**Summary:**

This paper studies the problem of learning a smooth function from iid data under different formal definitions of smoothness (e.g. learning an $\alpha$-Holder function, learning a function in Sobolev space). The paper makes a distinction between two different regimes of sample complexities: the "very high sample complexity" regime and the "high sample complexity" regime, noting that prior work on learning functions generally proves that the excess risk decays at some rate $O(n^{-\gamma})$ where $\gamma$ depends on your particular smoothness assumption.

Such results have to hold for arbitrarily large $n$, meaning that these rates really describe the decay of excess risk in the "very high sample complexity" regime, which they argue is often only descriptive for an unreasonably large sample complexity which isn't possible in practice (i.e. the constant in the big-Oh is huge). They further argue that realistic sample complexities lie in the merely "high sample complexity" regime, which exhibits decay at some other rate, say $O(n^{-\gamma'})$ for some $\gamma' \neq \gamma$.

---

It's worth noting the proof technique at a high level. The theory of the paper is mainly split into two key flavors of results:
1. A tight characterization of the excess risk $\mathcal{E}$ of a function-learning algorithm, showing that the excess risk $\mathcal{E} \approx A(n,\lambda)$, where $A(n,\lambda)$ is the sum of a variance term depending on a technical notion of "$\lambda$-statistical dimension" and bias term induced by regularization parameter $\lambda$.
1. A series of lower bounds roughly showing that for learning $\alpha$-Holder functions, Sobolev functions, and wavelets, the function $A(n,\lambda)$ grows exponentially in the dimension of the data.

These problems are already known to admit excess risk decay rates of $\mathcal{E} \leq O(n^{-\gamma})$ in theory, where the big-Oh hides the dependence on dimension. The above summarized two results together imply that these big-Oh rates in the prior work hide constants that are exponentially large in the data dimension.
More broadly, this implies that the intuition that "assuming a function is smooth lets us avoid a sample complexity that's exponential in the data dimension" is an insufficient intuition to carry around in data science.
It's more accurate to note that the sample complexity is not $O(n^d)$, but nevertheless still likely exponential in the dimension $d$.

Some experiments compliment this theory.

**Strengths:**

The paper has an interesting narrative, trying to add nuance to the discussion of how assuming that a function is smooth should significantly reduce the sample complexity of learning that function in $\mathbb{R}^d$ from a $O(n^{O(d)})$ sample complexity down to a $O(n^{\gamma})$ sample complexity, where $\gamma$ depends on the smoothness assumption.
It's a nice story, showing that the big-Oh really hides a different exponential dependence on data dimension.

The presentation is usually pretty good. There were points where I did have to read sentences or paragraphs a couple times to fully understand the idea, but the relevant information is all there, and nice intuitions for the expressions are given. This needing to re-read lines could _in some but not all parts_ also be chalked up to me being not super used to reading papers in this area.

The quality of the results is also pretty strong.
The first core result [Theorem 2, Line 139] is a very tight characterization of how the excess risk grows, being well approximated by this $A(n,\lambda)$ function.
It's nicely interpreted and its lower bounds for specific problems are compelling shows of how the excess risk rate of $O(n^{-\gamma})$ can really hide huge exponential terms.
I really like this proof setup and style -- though it's not perfect and lacks some connective, which I'll elaborate on in the _weaknesses_ section.
This proof approach also strikes me a pretty novel, and I could imagine similar analysis for other statistical problems following a similar proof strategy as this paper, with the two pronged analysis I mentioned in my summary.

Overall, I like this paper.
Unfortunately, it has some clarity of writing errors and a lack of connective tissue in some parts, which makes me hesitant to recommend publishing the paper. But I do like it quite a bit!

**Weaknesses:**

The paper has some clarity issues and some obvious error that really harm legibility.

The most odd weakness is that many of the references to figures are incorrect. Like [Line 50] mentions Figure 4 when it should mention Figure 1. [Line 217] mentions Figure 5, when Figure 5 is clearly irrelevant to that discussion, but I'm not clear which figure is supposed to be mentioned here.
I think Figure 8 on [Line 300] has the same issue, and I completely do not understand the message of that paragraph.
It's an odd (and presumably very easy to fix) error, but it really did hurt my comprehension at some vital points of the paper.
Because it hurt my comprehension, it acts as a very real mark against the paper.

---

The paper also makes some leaps in logic that can be hard to follow.
For instance, Figure 2 [above Line 196] is supposed to visually convey that, when learning an $\alpha$-Holder function, the excess risk has an exponential dependence on dimension.
To my understanding, if there was no such dependence on the dimension $d$ (as suggested by the convergence rate $O(n^{-\frac{2\alpha}{2\alpha+d}})$) then the heatmaps in Figure 2 would not show any relationship between excess risk (the color of the heatmap) and the dimension.
But since the heatmaps do show such a relationship, we can empirically see that the big-Oh bound is hiding a painful dependence on $d$.

This story is not well conveyed by Figure 2's description.
There a basic description of what experiment generated the plots, but there's no connective tissue explaining how the plots relate back to the problem of this dimension dependence.
I also have exactly no understanding of why the third plot in Figure 2 is present. No clue what message it's trying to evoke, nor why the x-axis of the plot is nonlinear in such an odd way.
This sort of lack-of-connective tissue appears in a few places, and really harms my understanding of how various puzzle pieces fit together in this paper.

This writing issue is especially painful in Section 4, where a theoretically tighter but harder to interpret bound on excess risk is presented in equation (14) [Line 265]. This harder-to-interpret rate is not really explained and instead the writing shifts to discussing a very intuitive explanation of two synthetic experiments where the excess rate decays fast-then-slow and slow-then-fast. The tie between the equation (14) and these experiments is thoroughly unclear.

These issues of clarity are really my big qualms with this paper.
Having not fully understood some of this technical material, or rather how this technical material relates to the experiments and intuitions, I really hesitate to accept the paper, hence the borderline judgement.

**Questions:**

Does Prop 1 [Line 194] really show an exponential dependence on dimension, or is the dependence polynomial in dimension and exponential in the degree $\alpha$?
Specifically, I'm thinking of the approximation $(\frac{n}{k})^k \leq \binom{n}{k} \leq (\frac{ne}{k})^k$, which implies that
$$\textstyle{(1+\frac d\alpha)^\alpha = (\frac{d+\alpha}{\alpha})^\alpha \leq \binom{d+\alpha}{\alpha} \leq (\frac{e(d+\alpha)}{\alpha})^\alpha = e^\alpha (1+\frac{d}{\alpha})^\alpha}$$
and therefore overall that the term in proposition 1 grows polynomially in $d$ for fixed alpha, and grows exponentially in $\alpha$.
Am I missing something here?

---

Here's a list of typos / recommended edits. Feel free to use whatever edits you want to use, these are all recommendations.

1. [Line 10] "they" instead of "that"
1. [Line 17] "mapping's" instead of "mapping"
1. [Line 17] "can we" instead of "to"
1. [Line 20] end sentence with a question mark
1. [Line 23] add "us" after "allows"
1. [Line 35] "was" instead of "were"
1. [Figure 1] Define $\mathcal{N}_1(\sigma,\lambda)$ or don't use the symbol.
1. [Line 64] Sometime before this line (i.e. before Section 2), define the notion of a transitory regime. Perhaps use the sentence from Figure 1, but outside of a figure caption.
> Those transitory regimes, where the behavior of the excess risk can be quite different than its asymptotic “stationary” behavior, are usually not well described by theory, while they might be the dominating regimes in applied machine learning when the number of samples is relatively small compared to the input dimension
1. [Line 50] Wrong figure reference
1. [Line 51] "era" instead of "area"
1. [Line 135] Remove "$\varepsilon^2/n$"
1. [Line 173-174] I think the grammar of this sentence is completely borked? Or maybe there's just a word or two that's throwing me way off? I dunno what you're trying to say.
1. [Line 213] Digest the rate of Prop 2 for us. For example, lower bound the maximum by the choice of $\ell=d$, and show the rate of the two terms, so that we can easily tell that the second term dominates the first term when $\|\|\vec m\|\|_2$ is large enough. (If I did my math correctly, it's roughly when $\|\|\vec m\|\| \geq d$?). Take this as a moment to return to the broader story about the scale of the constants in the big-Oh notation and really pull everything back together.
1. [Line 224] Remove "a"
1. [Line 254] You mention inequalities, but it's not clear if they're $\geq$ or $\leq$
1. [Figure 3] Include labels for the axes and titles on the plots.
1. [Line 262] add "being" after "$c_\gamma$"

---

> ### Author Rebuttal · Authors · 2023-08-08
>
> Dear Reviewer,
>
> Thank you very much for the depth of your review.
> Your observations helped us a lot to improve our paper.
> We are also deeply grateful for the typos you spotted.
>
> It is true that we have focused too much on derivations and have missed to present the connections between our different results.
> We have modified our draft as explained in our general answer to remedy this limitation.
> We have equally modified the caption of Figure 2 following your suggestions, and integrated your many other suggestions, it will greatly help us to better convey our message.
>
> We are sorry for the figure misnumbering, especially since it has caused you some extra work.
> We changed the figure labels while polishing the draft and missed some cross-references.
> Line 50 should refer to Figure 1, line 217 should refer to Figure 7 (left), line 300 should refer to Figures 12 and 13.
>
> We are sorry for the lack of clarity of Section 4.
> Our motivation was to crisply capture how, as the number of samples increases, one is able to reconstruct finer and finer details, and to present a simple analysis of how this can be read mathematically.
> We will clarify this, and we will expand on the math to show how Eq. (14) can be utilized to derive formally slow/fast and fast/slow profiles.
>
> You are right for the polynomial dependency in $d$ (actually, since ${\alpha+ d \choose \alpha} = {\alpha + d \choose d}$, the constants has the same type of dependency in $d$ and $\alpha$, which is polynomial).
> However, to beat the curse of dimensionality, $\alpha$ has to scale linearly in $d$, which implies the exponential dependency of the constant with respect to $d$.
> Thank you for pointing out our lack of explanation there.
>
> The right of Figure 2 illustrates the constant ${\alpha + d \choose \alpha}$ as a function as $\alpha$ in log-log scale (using the analytical continuation of the factorial to consider non-integer value of $\alpha$), we will equally add a Figure to illustrate the constant that appear in Proposition 2 to better digest it as you suggested.
>
> Thank you again for your detailed comments.
> We hope that our revised revision will answer most of our concerns, and stay at your disposal for further questions and suggestions.

---

> > ### Comment · Reviewer_yXo1 · 2023-08-21
> > **Thanks for the response!**
> >
> > Sorry for the late reply here. I appreciate the response from the authors.
> >
> > The core of my concern lied in clarity of presentation, and the authors did a good job of resolving my concerns. In particular, they show a clear image of how they would clarify the language in several parts of their paper.
> >
> > Unfortunately, I can't see a complete draft of the revised writing (and I'm not 100% I'd have a lot of time to rereading everything from scratch anyways....). But, since the majority of the paper is already well written, I do have faith that these authors can clear my concerns about clarity in a camera-ready draft.
> >
> > In light of this, I raise my score to a 6 -- a paper which should be accepted but that I cannot label as 7 without seeing a full revision of those sections. Overall, I do think this paper should be accepted.

---

### Official Review · Reviewer_z9Dq · 2023-07-24

**Soundness:** 3 good
**Presentation:** 1 poor
**Contribution:** 3 good
**Rating:** 5
**Confidence:** 3

**Summary:**

Classical results in learning theory have shown that smoothness of the function class can be leveraged to "break" the curse of dimensionality in empirical risk minimization. However, this paper shows that in the specific case where a Taylor expansion or a Fourier expansion is used to leverage smoothness, the constant in the sample complexity can grow exponentially in $d$, which is the dimension of the function.

**Strengths:**

I think the theoretical results in this paper are interesting since it tackles a very fundamental question in learning theory: "is smoothness of the target function enough to guarantee good generalization?" Classical results in learning theory (e.g. Theorem 1) show that the excess risk for estimating $\alpha$-smooth functions are upper bounded as $c n^{-2 \alpha /(2 \alpha+d)}$, where $n$ is the sample size, $d$ is the dimension of the input, and $c$ is a constant. Thus, for a fixed $d$, as the sample size $n$ grows, the excess risk decays rapidly to zero.

 In this paper, the authors show that in specific cases, such as when a Taylor expansion or Fourier expansion is used to estimate $f^\star$, the constant $c$ can actually be exponential in $d$. Therefore, at least in these specific cases, smoothness alone is not enough to guarantee efficient learning algorithm in terms of the sample complexity. I think these negative results in terms of sample complexity are of interesting because it shows that smoothness might not be a strong enough prior to guarantee good generalization.

**Weaknesses:**

I think the writing of this paper could be significantly improved. In its current state, it takes a long time to see what are the "main results" and main contributions of this paper. It is also hard to which parts of the paper are restating standard results or standard proofs, and which parts are technically new. On line 132 the authors state that "the usual proof of Theorem 1 [which is a known result] relies on theorems such as Theorem 2 below". This is very confusing to me. Is theorem 2 a standard result that the authors restate? Or is it a new result proved in this paper. How it is different from previous theorems used to prove Theorem 1? There also parts of the paper that are just difficult to understand because of the English. For instance, the sentence on line 172 makes no grammatical sense and is hard to understand.

Overall I think both the organization and the writing of this paper needs to be significantly improved, otherwise it would be hard for readers to spot the main contributions of this paper. The organization and the writing is the main issue preventing me from giving this paper a higher score, as I think this paper could benefit greatly from a major revision.

**Update** -- After the rebuttal I have decided to increase the score by 1, because the revised version outlined in the rebuttal pdf is much clearer than the current version. I hope the authors can implement all these changes, as promised.









**Questions:**

I also have trouble understanding the role of the meta-algorithm described on line 161. How is it related to the main results regarding Taylor and Fourier expansions?

---

> ### Author Rebuttal · Authors · 2023-08-07
>
> Dear Reviewer,
>
> Thank you very much for your feedback.
> It greatly helped us in improving the presentation of our work.
> As detailed in our general rebuttal answer (point 2), our draft now begins with generic negative results right after the introduction, and subsequently delves into details in RKHS settings.
>
> In terms of novelties, Theorem 2 was necessary to establish a lower bound on convergence rates, which was previously unknown. Previous results solely provide upper bounds.
>
> Moreover, as Taylor and Fourier expansions can be shown to be Bayesian optimal, the fact that they suffer from unfavorable constants implies that any other algorithm will necessarily suffer from such constants (see the attached pdf in our general rebuttal answer for crisper details).
>
> The objective of the meta-algorithm is to explain that the discussed approaches can be viewed through a single meta-algorithm that seeks the optimal ${\cal F}$ to balance and minimize ${\cal N}$ and ${\cal S}$.
> Implementation involves "kernel cross-validation" when defining $f_n$ (ridgeless) or $f_{n,\lambda=1}$ (ridge).
> In practice, local polynomials estimators change the kernel by adding or removing features in $\phi$ (based on partition coarseness and polynomial degree), while Fourier-based estimators modify the hyperparameters $\lambda$ and $\sigma$ which can be seen as a change of kernel as explained in the paragraph on "regularization is a change of kernel".
>
> Thank you again for your suggestions that have led us to meaningfully revise the presentation of our work.
> We hope that it will help you to better appreciate this work.
> We stay at your disposal for any further questions or suggestions.

---

### Author Rebuttal · Authors · 2023-08-07

Dear Reviewers,

We are grateful for your careful assessment of our work.
We feel that our contributions have been well understood, and your feedback has helped us to improve the presentation of our results.
It seems that the reviewers all agree on the facts that:
- It is important to get a sense of the statistical guarantees that one can expect in practical machine learning setups where one has few samples compared to input dimension.
- It is important to understand if smoothness alone can alleviate the curse of dimensionality.

Two primary concerns were raised:

1. The assertion that "smoothness does not completely alleviate the curse of dimensionality" seems to be proven for specific counterexamples but not to hold universally.

2. Shortcomings of presentations necessitate readers to exert additional effort in weaving together the various components.

(3.) Additionally, questions have been raised regarding the novelty of Theorem 2.

(4.) Some sentences conveying intuitions on the trade-off between approximation and estimation errors were hard to parse (especially lines 172-174).

We agree with these observations. Our immersion in technical details has diverted us from a simple presentation of the overarching narrative.
We have taken the following actions to rectify these shortcomings:

1. The discussed Taylor and Fourier expansion can be proved to achieve "Bayesian optimality", thereby establishing unassailable constants.
Indeed, utilizing the fact that when the target function depends linearly on $D$ features, any algorithm has to incur an excess risk of $\epsilon^2 D / n$ for at least one of the "linear" target functions (which is easily proved based on the work of Pinsker on "Optimal Filtering of Square-Integrable Signals in Gaussian Noise" in 1980), we can establish the comprehensive applicability of our findings by demonstrating that:
- For any algorithm, there exists a function $f$ with $f^{(\alpha)} = 0$, where the excess risk for this specific target function exceeds $\epsilon^2 {\alpha + d \choose \alpha} / n$.
- For any algorithm, there exists a function $f$ with $\hat f(\omega') = 0$ for $|\omega'| \geq \omega$, leading to an excess risk greater than $\epsilon^2 \omega^d / n$.

2. To enhance the document's coherence, we will sequentially present generic negative outcomes (as elaborated in point 1 above), and subsequently transition to the RKHS framework.
Here, we will expose the trade-off between the size of the search space and the proximity of $f^*$ to this space, explicitly characterized by the quantities ${\cal N}$ and ${\cal S}$ in Theorem 2.
Our focus will later shift to harmonic settings (e.g., uniform distribution on the sphere), where. thanks to explicit computation of ${\cal N}$ and ${\cal S}$, one can crisply capture the fact that "as the number of sample increases, one is able to reconstruct finer and finer details".
Finally, we will add a discussion section where we will gather the intuition scattered here and there in the submission.
Without modifying the substance of our work, this revised structure will provide a cohesive thread uniting the distinct facets of our paper.
We have attached a pdf to detail this revised organization.

(3.) Concerning novelty, Theorem 2 implies two corollaries: an upper bound and a lower bound on the excess risk. While akin upper bounds have been established (and used to prove Theorem 1), the novelty lies in the introduced lower bound (whose proof approach, as pointed out by Reviewer yXo1, is novel).
Its inclusion was indispensable for effectively bounding the constants that accompany the rates.

(4.) We will remove obscure sentences scatter here and there in the draft such as lines 172-174, and will make a specific discussion section where we discuss the generic "take-home" intuition.

Thanks again to all the reviewers for deeply helping us to improve our paper's presentation.
Your feedback will greatly facilitate the understanding of our work by future readers.
We are hopeful that these adjustments will improve your evaluation, and stay at your disposal for additional discussions.

Best regards,

The Authors

---

> ### Author Response · Authors · 2023-08-21
> **Thank you**
>
> As the discussion period comes to an end, we would like to thank all the reviewers for their many valuable suggestions: we are deeply grateful for the time you took to help us improve our draft!

---

### Decision · Program_Chairs · 2023-09-21

**Decision:**

Accept (poster)

**Comment:**

This paper studies the role of smoothness in breaking the curse of dimensionality in statistical learning problems. In particular, the paper presents an interesting negative result: it shows that in the specific case where a Taylor expansion or a Fourier expansion is used to leverage smoothness, the constant in the sample complexity to leverage smoothness can grow exponentially in the dimension.

Five reviewers have reviewed the paper, and their overall assessment of the paper was positive. I agree with the reviewers and believe that the paper sheds light on an interesting and, to some extent, overlooked phenomenon in the connection between the sample size, smoothness, and the problem dimension.